



# How does the UKESM1 climate model produce its cloud-aerosol forcing in the North Atlantic?

Daniel  P.  Grosvenor[1] and  Kenneth S. Carslaw[2]

[1]National Centre for Atmospheric Sciences, School of Earth and Environment, University of Leeds, Leeds, LS2 9JT, UK
[2]Institute for Climate and Atmospheric Science, School of Earth and Environment, University of Leeds, Leeds, LS2 9JT, UK

**Correspondence:** D. P. Grosvenor
(daniel.p.grosvenor@gmail.com)

**Abstract.**

Climate variability in the North Atlantic influences processes such as hurricane activity and droughts. Global model simulations have identified aerosol-cloud interactions (ACIs) as an important driver of sea surface temperature variability via surface aerosol forcing. However, ACIs are a major cause of uncertainty in climate forcing, therefore caution is needed in interpreting
the results from coarse resolution, highly parameterized global models.

Here we separate and quantify the components of the surface shortwave effective radiative forcing (ERF) due to aerosol in the atmosphere-only version of the UK Earth System Model (UKESM1) and evaluate the cloud properties and their radiative effects against observations. We focus on a northern region of the North Atlantic (NA) where stratocumulus clouds dominate (denoted the northern NA region) and a southern region where trade cumulus and broken stratocumlus dominate (southern
NA region). Aerosol forcing was diagnosed using a pair of simulations in which the meteorology is approximately fixed via nudging to analysis; one simulation has pre-industrial (PI) and one has present-day (PD) aerosol emissions.

Contributions to the surface ERF from changes in cloud fraction ($f_c$), in-cloud liquid water path ($LWP_{ic}$) and droplet number concentration ($N_d$) were quantified. Over the northern NA region increases in $N_d$ and $LWP_{ic}$ dominate the forcing. This is likely because the high $f_c$ there precludes further large increases in $f_c$ and allows cloud brightening to act over a larger region.
Over the southern NA region increases in $f_c$ dominate due to the suppression of rain by the additional aerosols. Aerosol-driven increases in macrophysical cloud properties ($LWP_{ic}$ and $f_c$) will rely on the response of the boundary layer parameterization, along with input from the cloud microphysics scheme, which are highly uncertain processes.

Model gridboxes with low-altitude clouds present in both the PI and PD dominate the forcing in both regions. In the northern NA the brightening of completely overcast low cloud scenes (100% cloud cover, likely stratocumulus) contributes the most,
whereas in the southern NA the creation of clouds with $f_c$ of around 20% from clear skies in the PI was the largest single contributor, suggesting that trade cumulus clouds are created in response to increases in aerosol. The creation of near-overcast clouds was also important there.

The correct spatial pattern, coverage and properties of clouds are important for determining the magnitude of aerosol forcing so we also assess the realism of the modelled PD clouds against satellite observations. We find that the model reproduces the
spatial pattern of all the observed cloud variables well, but that there are biases. The shortwave top-of-the-atmosphere ($SW_{TOA}$)





flux is overestimated by 5.8% in the northern NA region and 1.7% in the southern NA, which we attribute mainly to positive biases in low-altitude $f_c$. $N_d$ is too low by -20.6% in the northern NA and too high by by 21.5% in the southern NA, but does not contribute greatly to the main $SW_{TOA}$ biases. Cloudy-sky liquid water path mainly shows biases north of Scandinavia that reach up to between 50 and 100% and dominate the $SW_{TOA}$ bias in that region.

The large contribution to aerosol forcing in the UKESM1 model from highly uncertain macrophysical adjustments suggests that further targeted observations are needed to assess rain formation processes, how they depend on aerosols and the model response to precipitation in order to reduce uncertainty in climate projections.

## 1  Introduction

Uncertainty in the radiative forcing (RF) from aerosols is the largest of the climate RF uncertainties over the industrial period
(Boucher et al., 2013). GCMs (General Circulation Models) that simulate a small magnitude of cooling from aerosols are able to reproduce the observed temperature record if they have a low climate sensitivity and vice versa, which results in large uncertainties in climate sensitivity and therefore also in temperature change predictions (Andreae et al., 2005; Golaz et al., 2013). Mülmenstädt and Feingold (2018) suggests that one reason for the lack of progress in reducing the uncertainty in aerosol forcing despite years of research is that there has been a lack of studies that target (via evaluation and improvement)
the individual processes that cause aerosol-cloud interactions within GCMs.

Aerosol effective radiative forcing (ERF; which differs from RF in that all physical variables are allowed to respond to perturbations except for those concerning the ocean and sea ice, e.g., see Myhre et al., 2013) can be separated into a component due to aerosol radiative interactions (ARIs) that occur outside of clouds (sometimes also known as the direct effect) and a component due to aerosol cloud interactions (ACIs, or indirect effects). The ACI ERF is often also broken down into two
further components. The first is due to an increase in cloud droplet concentration ($N_d$) alone at constant liquid water content (LWC) and constant cloud fraction ($f_c$), which causes a decrease in the cloud droplet effective radius ($r_e$). Here, this is termed $ERF_{Nd}$ (or often the Twomey effect; Twomey, 1977). The second ERF component concerns rapid adjustments of LWC (or the vertical integral of this, which is the Liquid Water Path, LWP) for only the cloudy parts of model grid boxes (termed in-cloud LWP, or $LWP_{ic}$ here) and/or adjustments in $f_c$ that occur in response to the initial decrease in droplet size associated with the
$N_d$ increase. These are termed $ERF_{LWPic}$ and $ERF_{fc}$, respectively.

The mechanisms that cause the adjustments involve several microphysical and thermodynamical processes (Albrecht, 1989; Stevens et al., 1998; Ackerman et al., 2004a; Bretherton et al., 2007; Hill et al., 2009; Berner et al., 2013; Feingold et al., 2015). Simulation of adjustments in GCMs therefore requires the parameterization of many sub-grid-scale processes, which are likely to be more difficult for GCMs to get right than $ERF_{Nd}$ where only the change in $r_e$ needs to be parameterized. It is therefore
desirable to separate $ERF_{Nd}$ and the adjustment effects within GCMs so that they can be evaluated (against observations and high resolution models) and improved individually. The observational constraint of $ERF_{Nd}$ (which is likely easier than the constraint of adjustments) might then reduce the overall forcing uncertainty and highlight issues with the adjustment part of the forcing. Separating the adjustments into $ERF_{LWPic}$ and $ERF_{fc}$ components will further facilitate more detailed process





level improvements. A further reason to separate the different components is that current models simulate the same forcing

with different combinations of $ERF_{Nd}$ and adjustment components (Gryspeerdt et al., 2020). Therefore one aim of this study is to separate and quantify the ERF contributions from $N_d$, $LWP_{ic}$ and $f_c$ changes in a GCM. A second aim is to also quantify the amount of aerosol forcing from a GCM that originates from different cloud types and changes between cloud types. This too will allow a more targeted model evaluation and improvement in future work.

The aerosol forcing of GCMs is also important regionally, for example in the North Atlantic (NA) region, which is the focus

of this paper. It has been suggested (Booth et al., 2012, hereafter B12) that surface radiative aerosol forcing is the dominant driver of the variability in multi-decadal NA sea surface temperatures (SSTs) for the ocean-atmosphere coupled GCM (the UK Met Office HadGEM2-ES model) that was used in the 3$^{rd}$ Coupled Model Intercomparison Project (CMIP). NA SST variability has been linked to impacts on important climate phenomena such as hurricane and tropical storm activity (Zhang and Delworth, 2006; Smith et al., 2010; Dunstone et al., 2013); rainfall anomalies in Europe and North America (Sutton and

Hodson, 2005; Sutton and Dong, 2012); droughts in the African Sahel and Amazonian regions (Hoerling et al., 2006; Knight et al., 2006; Ackerley et al., 2011); Greenland ice-sheet melt rates (Holland et al., 2008; Hanna et al., 2012); anomalies in sea-levels (McCarthy et al., 2015); and the mid-latitude jet strength (Woollings et al., 2015). For a review of changes in the North Atlantic climate system (with a focus on more recent changes) see Robson et al. (2018).

B12 showed that HADGEM2-ES, which represented aerosol-cloud interactions, reproduced the observed NA SSTs with

good fidelity in contrast to the other CMIP3 models that mostly did not include aerosol-cloud effects. Furthermore, tests using constant aerosols clearly showed the impact of aerosols upon NA SST variability in HADGEM2-ES. Aerosol ARI forcing was found to be negligible in the NA compared to aerosol indirect forcing. The spatial patterns of the indirect forcing, the downwelling surface shortwave (SW) radiation anomalies and the SST anomalies were all consistent, indicating a link between the three. Moreover, a simulation with fixed SSTs showed similar incoming surface SW to that in the coupled model.

This suggested that the simulated surface SW anomalies were not brought about by the modification of SSTs as a result of ocean dynamics, or other processes. Thus, the implication from the HADGEM2-ES model is that aerosol indirect forcing has a direct local impact on SSTs via the modification of surface downwelling SW radiation.

The claims made in B12 are based upon a GCM and not direct observations. As mentioned earlier, the aerosol forcing in GCMs is highly uncertain for many potential reasons. For example, B12 used a coarse model resolution and thus the model

relies upon parameterizations to represent sub-grid cloud formation and cloud aerosol interactions. It could be the case that HADGEM2-ES overestimates the magnitude of the aerosol forcing and thus overstates its role in driving NA SST variability. Zhang et al. (2013) suggest that the HADGEM2-ES model has some shortcomings in its representation of the ocean that may affect its ability to properly simulate the influence of the ocean. Other papers also argue for an important role for ocean processes in determining the NA SST variability (Ba et al., 2014; Knight, 2005; Menary et al., 2015; Robson et al., 2016;

Zhang et al., 2016a; Yan et al., 2018), which may indicate a lesser role for aerosol than simulated in B12.

The aim of this paper is to quantify the mechanisms by which a global climate model (an improved version of the model used in B12) produces aerosol forcing, with a focus on the NA region. Some work breaking down the aerosol ERF of a different GCM into that due to changes in $N_d$, $LWP_{ic}$ and $f_c$ has already been recently published (Mülmenstädt et al., 2019). They found



that in the ECHAM-HAMMOZ model $ERF_{LWPic}$ was quite similar to $ERF_{Nd}$ for most latitudes, except in the Southern

Ocean where there was a much larger $ERF_{LWPic}$ contribution. $ERF_{fc}$ contributions were mostly between around 50–75% of

the $ERF_{Nd}$ contribution, but again in the Southern Ocean there was a larger $ERF_{fc}$ contribution than $ERF_{Nd}$ contribution.

This was also true in the polar regions. Globally the overall contribution from adjustments ($ERF_{fc}$ and $ERF_{LWPic}$) was

-0.92 W m$^{-2}$ compared to an $ERF_{Nd}$ contribution of -0.52 W m$^{-2}$, so that in this model more aerosol forcing is coming

from the more complicated adjustment processes highlighting the importance of evaluating these processes in more detail.

Here we perform a similar analysis but with a different model. This will allow the two models to be compared in terms of how

they produce their aerosol forcing. A very different breakdown between the models would mean that one of the models was

incorrect and allow the basis for more detailed evaluation. We also focus on the NA region and on surface aerosol forcing rather

than top of the atmosphere (TOA) forcing due to the potential importance of aerosol forcing for the climate variability via sea

surface temperature changes there. The focus on the NA also allows a more detailed look at the processes than is possible

from a global study. We also go beyond the study of Mülmenstädt et al. (2019) by also characterizing the cloud regimes and

the changes in cloud regimes for which aerosol forcing predominantly occurs according to the model; this will then allow

(in future work) these regimes to be comprehensively evaluated against observations and also against high resolution models.

High resolution modelling needs to be targeted to a smaller selection of regimes given the high computational cost. In addition,

observations will be used to evaluate the general model cloud properties (spatial positioning, areal coverage, thickness, droplet

concentrations, etc.) since these will affect the resulting aerosol forcing.

## 2  Methods

### 2.1  Definition of Liquid Water Path

Throughout this paper the term LWP refers to the all-sky value (i.e., including both the cloudy and clear-sky portions of model

gridboxes or observed regions) and $LWP_{ic}$ refers to the in-cloud liquid water path, which is that from the cloudy regions only.

It is assumed here that $LWP = f_c LWP_{ic}$.

### 2.2  Model details

We examine the aerosol-cloud interactions in the UKESM1-A model, which is the atmosphere-only version of the coupled

UKESM1 (UK Earth System Model), which was submitted as part of the 6$^{th}$ Coupled Model Intercomparison Project (CMIP6).

UKESM1 is based on the HADGEM3-GC3.1 physical climate model (Williams et al., 2017; Kuhlbrodt et al., 2018), but in

addition couples several earth system processes (Sellar et al., 2019). These additional components include the MEDUSA

ocean biogeochemistry model (Yool et al., 2013), the TRIFFID dynamic vegetation model (Cox, 2001) and the stratospheric-

tropospheric version of the United Kingdom Chemistry and Aerosol (UKCA) chemistry model (Archibald et al., 2019). This

version of the UKCA allows a more complete description of atmospheric chemistry compared to HADGEM3-GC3.1. For

example, the latter uses an offline climatology for oxidants, whereas in the UKESM1 oxidants are treated explicitly.





The UKESM1-A atmosphere only version differs from the UKESM1 in that it does not include: the ocean and sea ice models; MEDUSA; and TRIFFID. Instead, the UKESM1-A configuration uses observed sea surface temperatures and sea ice concentrations provided by the Program for Climate Model Diagnosis and Intercomparison (Taylor et al., 2000, https://pcmdi.llnl.gov/mips/amip/). Vegetation (vegetation fractions, Leaf Area Index, canopy height) and surface ocean biology fields (dimethyl sulphide and chlorophyll ocean concentrations) are prescribed from a member of the UKESM1 CMIP6 historical

ensemble (Sellar et al., 2019). This ensures that the prescribed vegetation and ocean biological fields mirror those simulated by the TRIFFID dynamic vegetation and MEDUSA scheme in the coupled historical run. The vegetation fractions are prescribed from time-varying annual means; Leaf Area Index, DMS and chlorophyll concentrations are monthly values from the time-means of the 1979-2014 period; and the canopy heights are an overall time-mean for 1979-2014. All other emissions of gases and particles from sea and land surfaces are prescribed from the CMIP6 inventories; see Mulcahy (2020) for details of the

specific implementation for the UKESM1.

    The atmospheric component is the GA7.1 atmospheric configuration of the Unified Model (UM). Full details of this can be found in Walters et al. (2019) and Mulcahy (2020). Here we only describe in detail the features that are more relevant to aerosol-cloud interactions. We use an N96 horizontal resolution, which is $1.875 \times 1.25$ º ($208 \times 139$ km) at the equator. 85 vertical levels are used between the surface and 85 km altitude with a stretched grid such that the vertical resolution is 13 m

near the surface and around 150–200 m at the top of the boundary layer. We chose this resolution since it is the same as that used for long climate runs in the CMIP6 model intercomparison and is the same horizontal resolution as that used in the B12 study.

    Aerosol number concentrations are treated prognostically with the GLOMAP multi-modal scheme (Mann et al., 2010, 2012), which uses five log-normal aerosol size modes and includes sulfate, sea-salt, black carbon and organic carbon chemical com-

ponents. These aerosol components are treated as being internally mixed within each size mode. Mann et al. (2010) and Mann et al. (2012) provide further details with some small changes for the implementation within UKESM1-A described in Mulcahy (2020). Mineral dust is simulated separately using the CLASSIC dust scheme (Woodward, 2001; Mulcahy, 2020).

    Shallow, mid and deep convection are parameterized separately to other cloud types (see Walters et al., 2019, for details). The parameterizations do not take into account aerosol, or droplet concentrations and they use their own simplified microphysics

scheme. For cloud that is not shallow, mid, or deep convection (termed large-scale cloud), the UKCA-Activate scheme is used to calculate cloud droplet concentrations from the aerosol size distribution using the West et al. (2014) scheme based on the parameterisation of droplet activation in Abdul-Razzak and Ghan (2000). Cloud droplet concentrations at cloud base are replicated vertically throughout contiguous columns of cloud. The cloud droplet activation scheme is a diagnostic scheme since it is run on each model time step without consideration of how many cloud droplets were present before. The cloud droplet

number concentration is then passed to the radiation and microphysics schemes.

    The large-scale cloud microphysics is single-moment, such that the mass of liquid water, but not the cloud droplet number, is advected by the model and retained in memory between model time steps. It is based on Wilson and Ballard (1999), but with improvements to the warm rain parameterisations suggested by Boutle et al. (2014), which include bug fixes, a treatment of rain fraction that is consistent with the prognostic rain formulation, a switch to the Khairoutdinov and Kogan (2000) pa-





rameterisation for autoconversion and accretion, and a bias correction for the latter processes to deal with sub-grid variability of cloud and rain water. Relative to Wilson and Ballard (1999) there is also an improved treatment of drizzle rates (Abel and Shipway, 2007) and a prognostic treatment of rain that allows the 3-dimensional advection of precipitation. The introduction of the latter required modifications to be made to the aerosol wet scavenging processes as described in (Mulcahy, 2020). The bulk properties (cloud fraction, cloud liquid water content, vertical overlap, etc.) of large-scale cloud are parameterized using

the prognostic cloud fraction and prognostic condensate (PC2) scheme (Wilson et al., 2008a, b) with modifications described in Morcrette (2012). The atmospheric boundary layer is parameterizated using the turbulence closure scheme of Lock et al. (2000) with modifications described in Lock (2001) and Brown et al. (2008).

There are some some differences in the treatment of aerosols in UKESM-A and the physical climate model (HADGEM-GC3.1) primarily related to natural aerosols, aerosol chemistry and the prescription of anthropogenic $SO_2$ emissions (see

Mulcahy, 2020, for details).

## 2.3 Simulation details

The model is run from 1st March 2009 to 28th March 2010. The first 27 days were used to spin-up the aerosol and chemistry fields leaving one year of data for analysis. The time period was chosen to allow comparisons with relevant field campaigns that will be performed in future work. Two parallel simulations were performed; one using pre-industrial (PI) CMIP6 aerosol

emissions from the year 1850 and the other using present-day (PD) emissions (i.e., the CMIP6 emissions corresponding to the simulation period). Both simulations were nudged every 6 h to ERA-Interim horizontal wind fields between ~2277 m and ~47,251 m (applied between the 18th and 76th model level from the surface). The nudging ensures that the meteorology in the two runs is very similar, thereby allowing cloud radiative effects due solely to the aerosol perturbation to be calculated. Following the recommendations of Zhang et al. (2014), we do not nudge the temperature field in order to allow fast-acting

local responses to aerosol-induced temperature changes (such as those from precipitation suppression, ARI and semi-direct radiative effects). Instantaneous model fields are output every 27 hours, which allows the sampling of the diurnal cycle and more complex output analysis than is possible using time-averaged data.

## 2.4 Surface forcing calculations and forcing partitioning

In this paper we are concerned with the shortwave aerosol ERF at the surface since we are interested in the effects on SSTs and

N. Atlantic climate variability. To separate the aerosol ERF into ARI and ACI components we use output from the triple calls to the radiation scheme that are made by the model every timestep. One call calculates the surface SW fluxes taking into account both aerosols and clouds (designated here as $SW_{aerosol+cloudy}$), one call calculates the fluxes with the background aerosol removed (i.e., the aerosol outside of clouds and interstitial aerosol; aerosol Twomey and adjustment effects on the clouds are still included here; $SW_{clean+cloudy}$) and one call calculates the fluxes with both the background aerosol and clouds removed





($SW_{clean+clear}$). These are detailed in Table 1. The instantaneous changes in SW due to ARIs and ACIs for a given grid box are then calculated as :-

$$\Delta SW_{ARI} = SW_{aerosol+cloudy} - SW_{clean+cloudy}$$
$$\Delta SW_{ACI} = SW_{clean+cloudy} - SW_{clean+clear} \qquad (1)$$

The corresponding instantaneous radiative forcings due to anthropogenic aerosols ($ERF_{ARI}$ and $ERF_{ACI}$) can then be calculated from simulations with PI and PD aerosol emissions :-

$$ERF_{ARI} = \Delta SW_{ARI_{PD}} - \Delta SW_{ARI_{PI}}$$
$$ERF_{ACI} = \Delta SW_{ACI_{PD}} - \Delta SW_{ACI_{PI}} \qquad (2)$$

**Table 1.** Details about the aerosol and clouds that are used in the three calls to the radiation scheme performed on each timestep.

| SW radiation call | Description |
| --- | --- |
| $SW_{aerosol+cloudy}$ | Aerosols and clouds |
| $SW_{clean+cloudy}$ | Aerosols removed, but clouds included |
| $SW_{clean+clear}$ | Aerosols and clouds removed |

We also decompose the surface ACI aerosol forcing into contributions from changes in $N_d$, $LWP_{ic}$ and $f_c$ between the PI and PD. This is achieved using an offline calculation of the surface SW fluxes, which allows the magnitude of each term to be assessed individually. The approach is described in Grosvenor et al. (2017) and in Appendix E.

## 3 Results

### 3.1 Model evaluation against satellite observations


Here we evaluate the PD simulation against satellite observations. The motivation for the model evaluation is that without a good representation of cloud properties and spatial distribution it is likely that the model aerosol forcing will be incorrect. For example, if the simulated clouds have a cloud fraction that is too high then a cloud albedo perturbation due to aerosol would be larger than in reality. Placement of clouds in the wrong locations could affect forcing via the differing incoming
solar insolation, or could affect their chances of interacting with sources of aerosol. Where possible we use the COSP (Cloud Feedback Model Intercomparison Project Observation Simulator Package Bodas-Salcedo et al., 2011) satellite simulator model output for the relevant satellite in order to get a fairer comparison between the model and satellite. This is particularly important when comparing cloud fractions since the cloud detection threshold (e.g., in terms of optical depth) varies between the different





satellite instruments and needs to be consistent with the threshold LWC used to define a cloud in the model. Furthermore, COSP
accounts for the effect of overlying layers of cloud, which affects the observation of underlying clouds.

Following Gustafson and Yu (2012), model biases are quoted in terms of the normalized mean bias factor (NMBF) and
the normalized mean absolute error factor (NMAEF), which are both expressed as a percentage, in order to provide unbiased
metrics that are symmetric about zero. These quantities are similar to the mean percentage bias and the RMSE except that
account is taken of whether the model is greater or lower than the observations. In addition, the NMAEF does not involve
squaring the error, which can lead to an exaggerated influence from larger biases. Appendix B provides the definitions for
these metrics. Table 2 lists these bias metrics (along with the spatial correlation coefficients, r, between the model and the
observations) for the different regions considered (see next section) and for the different cloud variables. It should also be
borne in mind that as well being due to issues with the representation of clouds in the model, differences in cloud properties
between the model and satellite might also be due to cloud adjustments to aerosol that are too strong or weak, and that it is
difficult to distinguish between the two.

### 3.1.1   Cloud fraction evaluation

The distribution of time-mean low-altitude $f_c$ from the model is compared to CALIPSO (Cloud-Aerosol Lidar and Infrared
Pathfinder Satellite Observation) satellite LIDAR observations in Fig. 1. Version 3 of the CALIPSO-GOCCP (GCM Oriented
Cloud Calipso Product; Chepfer et al., 2010) is used, which is available at https://climserv.ipsl.polytechnique275.fr/cfmip-obs/
index.html.satelliteinstrument. The model has a good spatial correlation with the satellite (r=0.83) over the region shown with
both model and observations showing higher cloud fractions in the northern part of the NA where the time-mean values can
reach around 70% especially towards the west near the coast of Canada and north of Iceland and Scandinavia. More than
40–50% of the low-altitude clouds in these regions are stratocumulus (Wood, 2012). The model has a positive bias of around
30 to 40% in the region off the eastern Canadian coast and along the east coast of the USA. There is also a band of higher cloud
fractions (around 20–30%) down the west coast of Africa in both the model and observations; this region is also associated
with a high occurrence of stratocumulus (40–60% of the low clouds according to Wood, 2012). Model biases are minimal in
these regions. The model and observations both show lower cloud fractions in the southwest region of the domain.

The model captures the main pattern of low-altitude cloud fraction well giving us some confidence that it provides a realistic
representation of the broad cloud types and locations in the NA. However, the model uses observed SSTs and the winds
above the boundary layer are nudged to reanalyses, which will help the model to replicate the correct cloud patterns to some
degree. But even with the correct SSTs and wind fields, accurate simulation of the spatial pattern of $f_c$ is a strong test of
the model boundary layer and cloud schemes, in particular the correct vertical thermodyamical structure and entrainment, the
corresponding cloud fraction response, etc. Furthermore, the free-running ocean-coupled version of the model (the UKESM1;
Sellar et al., 2019), which has no wind nudging and predicts the SSTs itself, also reproduces the cloud pattern and amount well
(Robson et al., 2020, submitted to JAMES).

We define a sub-region (marked in Fig. 1 and denoted northern North Atlantic, or northern NA) to represent the region where
the stratocumulus cloud fraction is large. We also define a second region immediately to the south of the northern NA region





(southern North Atlantic, or southern NA, see Fig. 1) where the annual mean stratocumulus cloud cover is lower (Wood, 2012) indicating more broken stratocumulus, and/or a lower frequency of occurrence of stratocumulus. The overall low-altitude $f_c$

area-weighted biases for these regions were 5.1% and -23.9% for the northern and southern NA regions, respectively (see Table 2).

For mid-altitude clouds (Fig. 2) the spatial pattern is again captured well by the model (r=0.80). Percentage biases are generally low in the northern NA region with a mean model bias of -11.0%. There are larger negative biases in the southern part of the domain with a mean bias of -25.8% for the southern NA region; however, the observed cloud fractions are very

low in those regions making higher percentage biases more likely. There are also positive biases of up to around 30% in the stratocumulus region north of Scandinavia.

The modelled high-altitude cloud fraction is generally biased high over the ocean with mean biases of 7.3% for the northern NA and 4.5% for the southern NA, although the spatial pattern is again good (r=0.79 for the whole region). Positive biases are particularly evident in the region north of around 60°N where biases of up to 90 % occur. Since low-altitude clouds are likely to

be the most important for aerosol-cloud interactions we will leave an investigation into the biases in the mid and high altitude clouds to future research.

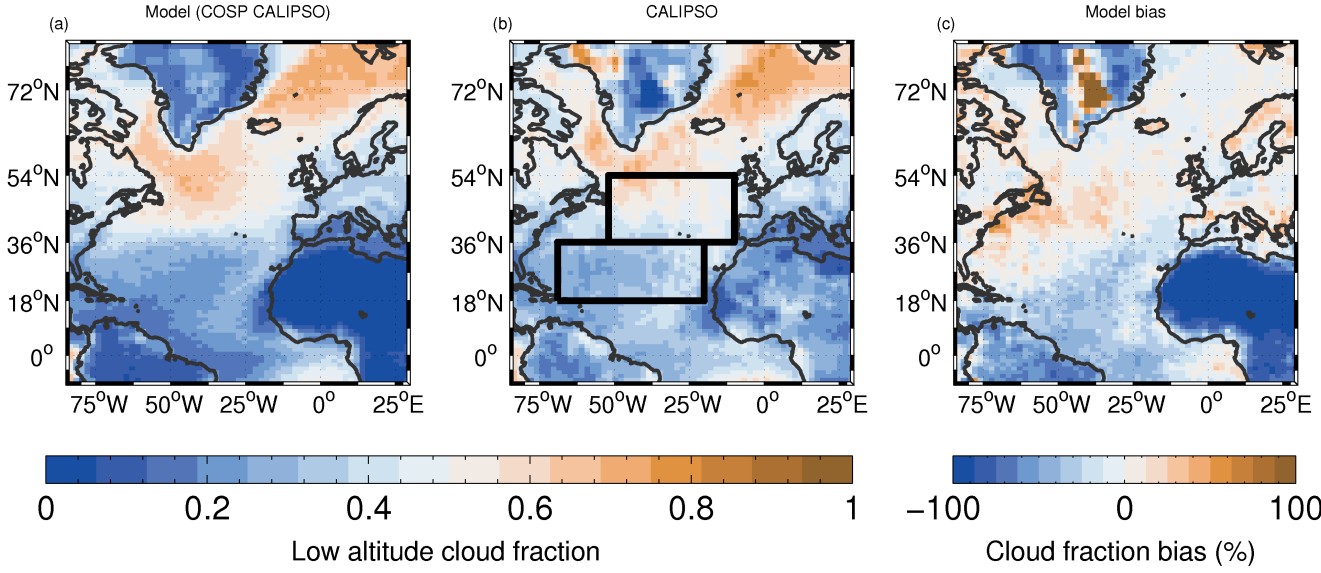

**Figure 1.** Low-altitude (CTP>=680 hPa) cloud fraction evaluation. Left: model; middle: CALIPSO satellite data;. right: model bias. For the model we are using the COSP simulator for CALIPSO to deal with the satellite cloud detectability threshold and overlying layers.





**Table 2.** Model evaluation statistics for the various sub-regions using time-averaged data. The normalized mean bias factor (NMBF) and the normalized mean absolute (NMAEF) error factor statistics are used following (Gustafson and Yu, 2012); see Appendix B for definitions. r is the spatial correlation coefficient between the model and observed time-averages. All values are area-weighted to account for the variation in area of the model grid-boxes.

| # | Region name | r | Model mean | Obs. mean | NMBF (%) | NMAEF (%) |
|---|---|---|---|---|---|---|
| | | | Low-altitude cloud fraction | | | |
| 1 | N Atlantic | 0.83 | 0.27 | 0.34 | -28.1 | 37.3 |
| 2 | NN Atlantic | 0.79 | 0.50 | 0.48 | 5.1 | 11.1 |
| 3 | SN Atlantic | 0.37 | 0.26 | 0.32 | -23.9 | 26.5 |
| | | | Mid altitude cloud fraction | | | |
| 1 | N Atlantic | 0.80 | 0.12 | 0.16 | -37.8 | 44.8 |
| 2 | NN Atlantic | 0.82 | 0.17 | 0.18 | -11.0 | 16.8 |
| 3 | SN Atlantic | 0.74 | 0.06 | 0.08 | -25.8 | 36.1 |
| | | | High altitude cloud fraction | | | |
| 1 | N Atlantic | 0.79 | 0.29 | 0.32 | -8.5 | 22.5 |
| 2 | NN Atlantic | 0.52 | 0.36 | 0.33 | 7.3 | 10.5 |
| 3 | SN Atlantic | 0.89 | 0.26 | 0.25 | 4.5 | 12.6 |
| | | | In-cloud Liquid Water Path ($LWP_{ic}$; g m$^{-2}$) | | | |
| 1 | N Atlantic | 0.68 | 164.4 | 197.3 | -20.0 | 30.9 |
| 2 | NN Atlantic | 0.70 | 175.3 | 201.5 | -14.9 | 16.2 |
| 3 | SN Atlantic | 0.80 | 168.2 | 199.1 | -18.4 | 24.6 |
| | | | In-cloud Liquid Water Path for $f_{LWP}> 0.99$ ($LWPic_{0.99}$; g m$^{-2}$) | | | |
| 1 | N Atlantic | 0.42 | 71.3 | 69.3 | 2.9 | 35.5 |
| 2 | NN Atlantic | 0.28 | 70.7 | 75.1 | -6.2 | 38.1 |
| 3 | SN Atlantic | 0.37 | 68.2 | 77.4 | -13.5 | 28.3 |
| | | | Droplet concentration ($N_d$; cm$^{-3}$) | | | |
| 1 | N Atlantic | 0.67 | 155.5 | 139.7 | 11.3 | 37.5 |
| 2 | NN Atlantic | 0.15 | 86.6 | 104.5 | -20.6 | 28.5 |
| 3 | SN Atlantic | 0.22 | 124.0 | 102.1 | 21.5 | 37.5 |
| | | | SW TOA flux (W m$^{-2}$) | | | |
| 1 | N Atlantic | 0.89 | 96.8 | 99.1 | -2.4 | 8.0 |
| 2 | NN Atlantic | 0.89 | 104.2 | 98.5 | 5.8 | 6.7 |
| 3 | SN Atlantic | 0.91 | 76.5 | 75.2 | 1.7 | 4.3 |



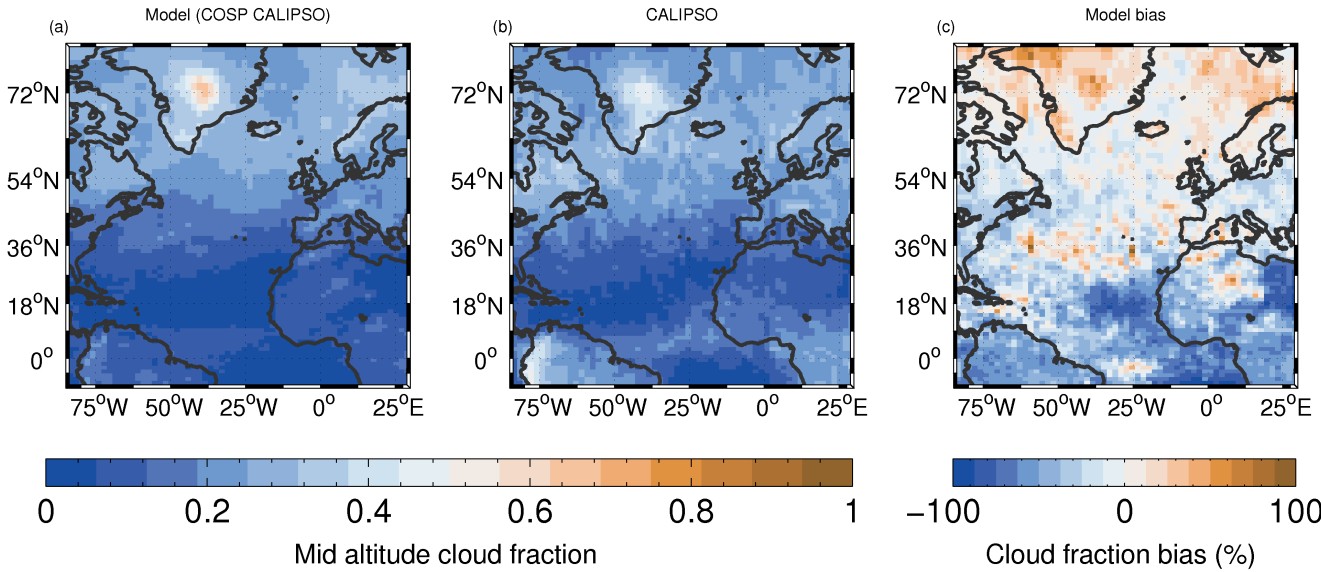

**Figure 2.** As for Fig. 1, except for mid-altitude cloud ($440 <= CTP < 680$ hPa).

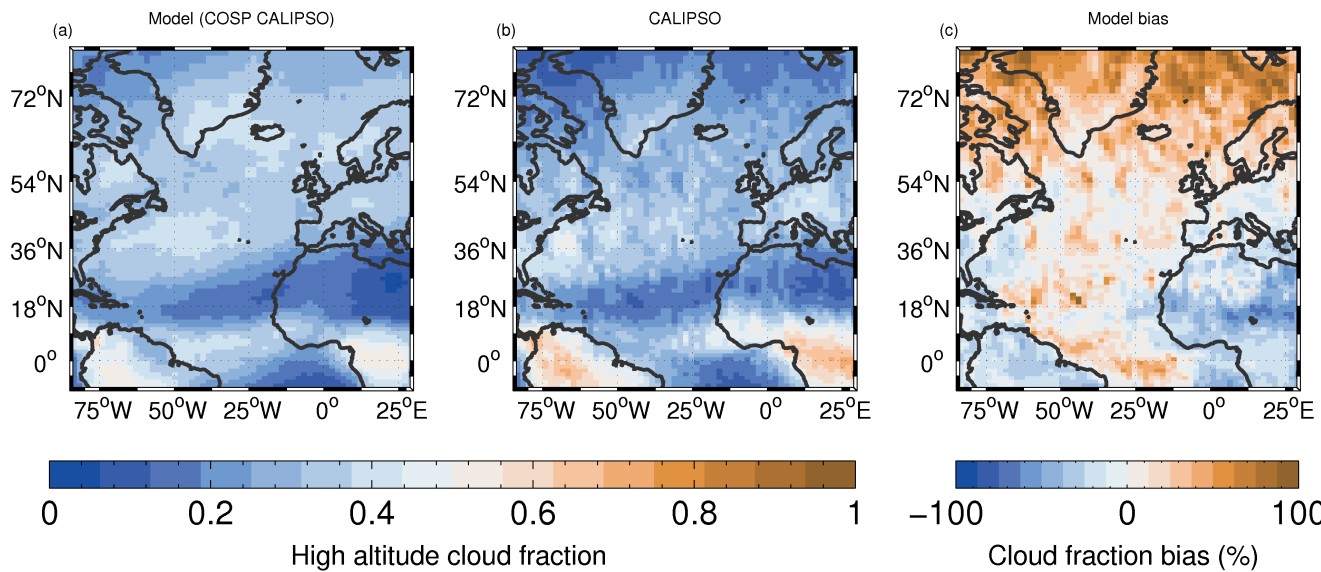

**Figure 3.** As for Fig. 1, except for high-altitude cloud ($CTP < 440$ hPa).





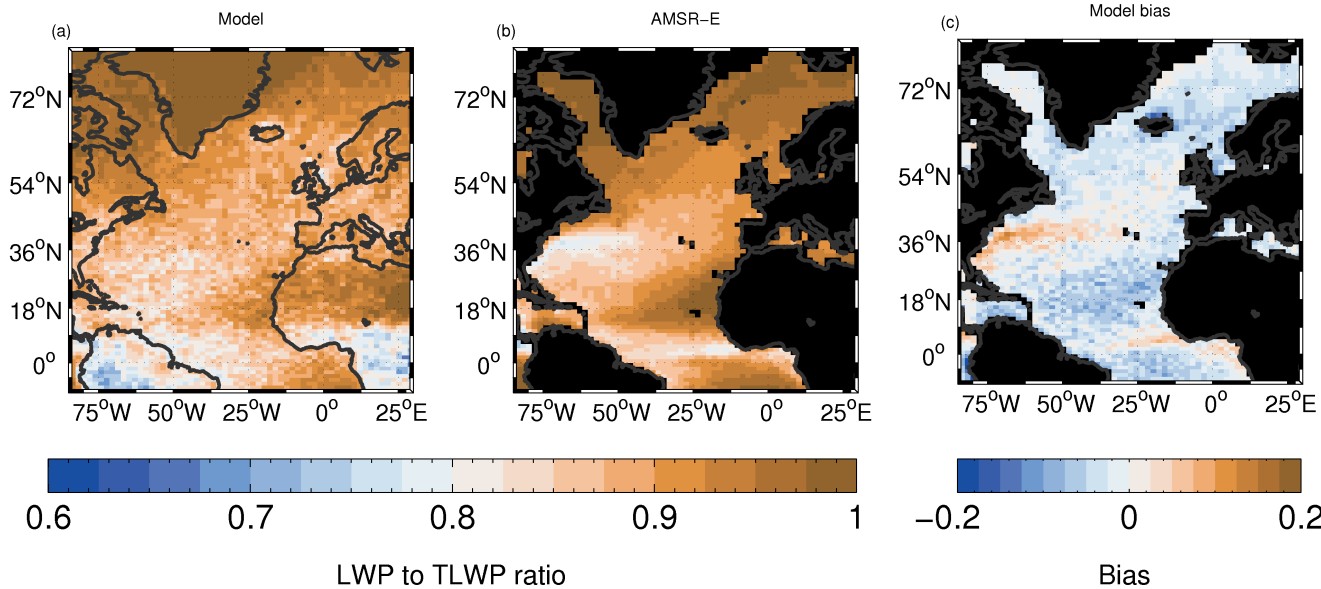

**Figure 4.** Time average of $f_{\text{LWP}}$ (ratio of LWP to LWP+RWP). For the model, LWP and RWP contributions from both the large-scale cloud scheme and the convective parameterization are included. The RWP for AMSR-E is calculated from the retrieved LWP and rain rate by inverting the retrieval algorithm, as described in Elsaesser et al. (2017). The same raindrop size distribution and fall-speed relationship as for the model is used. Both daytime and nighttime AMSR-E overpasses are used.

### 3.1.2 LWP evaluation

Liquid water path is important for cloud optical depth and hence cloud albedo. For example, for a fully overcast cloud with a liquid water content that increases with height in a manner consistent with adiabatic uplift, the optical depth ($\tau_{\text{c}}$) is proportional
to $LWP_{\text{ic}}^{5/6} N_{\text{d}}^{1/3}$. Therefore, a given relative change in $LWP_{\text{ic}}$ produces more than twice the relative change in $\tau_{\text{c}}$ that the same relative change in $N_{\text{d}}$ does. $LWP_{\text{ic}}$ is also important in terms of cloud physics since it helps to determine rain rates and droplet sizes.

Microwave satellite instruments such as AMSR-E (Advanced Microwave Scanning Radiometer for Earth Observing System; e.g., Wentz and Meissner, 2000), whilst only being able to retrieve over ocean surfaces, probably represent the most accurate of
the available retrievals of LWP since they are not subject to the biases associated with retrievals of LWP that use a combination of visible and shortwave-infrared wavelengths (e.g., MODIS, MODerate Imaging Spectroradiometer; Salomonson et al., 1998), they give a better representation of the total column LWP than from such retrievals, are not affected by the presence of ice and are available in both the daytime and nighttime (O'Dell et al., 2008; Elsaesser et al., 2017). Note that LWP from AMSR-E is the all-sky LWP and so includes the contributions from both clear and cloudy regions (as does the LWP output from the model),
whereas MODIS retrieves the in-cloud LWP, $LWP_{\text{ic}}$. However, AMSR-E retrievals are still subject to biases; Lebsock and Su





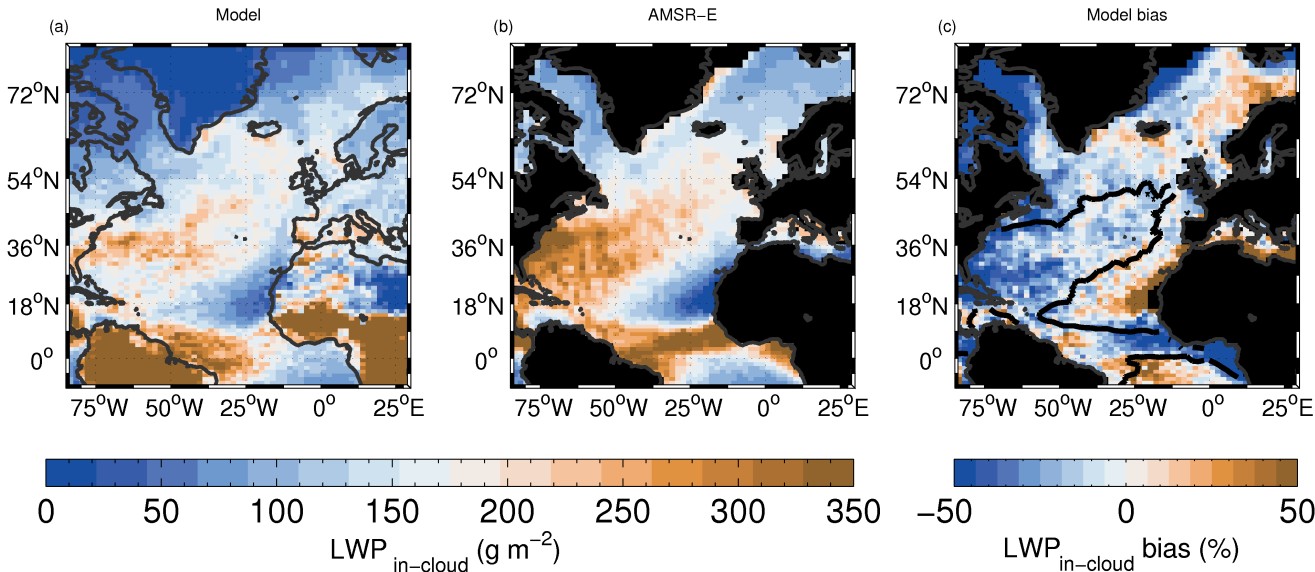

**Figure 5.** Time-mean in-cloud LWP ($LWP_{ic}$) model evaluation for both day and night overpasses. The bias plot on the right includes a contour of the 0.9 value of $f_{LWP}$; see Fig. 4 for the full map of $f_{LWP}$ for reference.

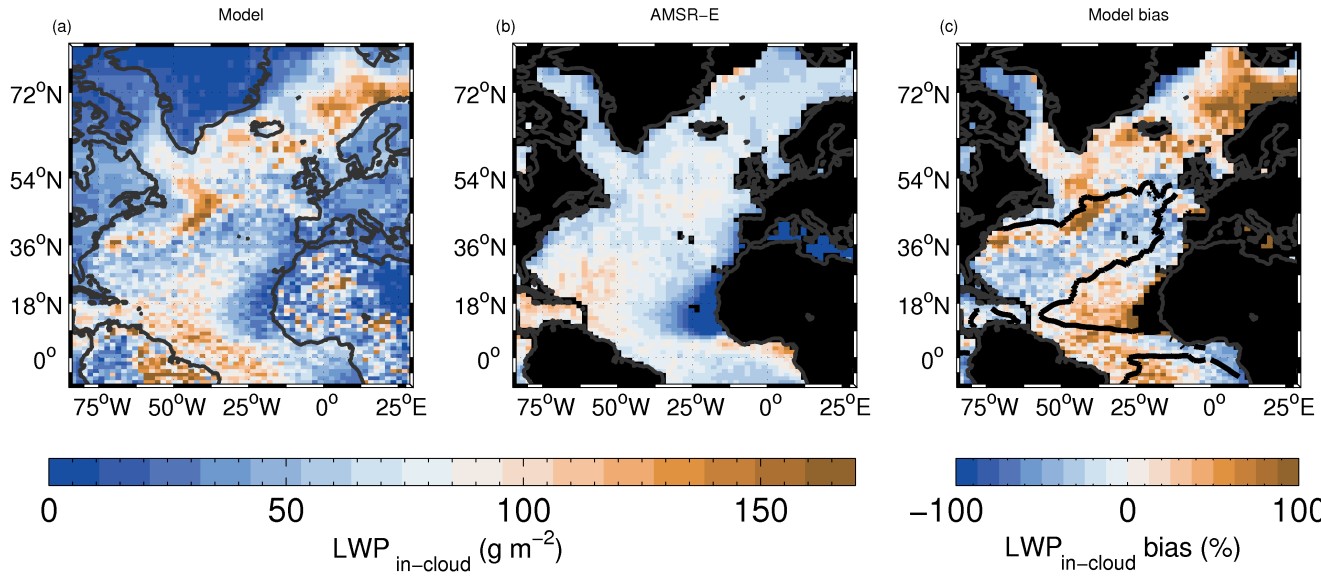

**Figure 6.** As for Fig. 5 except both the model and satellite data has been filtered before time averaging to only include datapoints for which $f_{LWP}$ is greater than 0.99. This quantity is denoted as $LWPic_{0.99}$.





(2014) suggest that the main cause of bias is the presence of rain (and the inability to directly detect whether rain is present). Therefore, as suggested in Elsaesser et al. (2017) we place more confidence in the microwave LWP observations when the ratio of the all-sky LWP to TLWP (LWP + RWP, where RWP is rain water path) is high; we denote the ratio as $f_{\mathrm{LWP}}$. A caveat here, though, is that the partitioning of LWP and RWP from microwave instruments, and hence the estimate of $f_{\mathrm{LWP}}$, is itself

uncertain.

Given the uncertainties when rain is present we also initially only use the model LWP not the RWP. Furthermore, we only consider the model LWP from the large-scale cloud scheme and not that from the convective parameterization. Convective LWP will be mostly associated with heavily raining environments and so the observations in such regions are again more uncertain. However, we examine and discuss how RWP and convective LWP and RWP change the model evaluation in Appendix C. The

AMSR-E instrument is onboard the Aqua satellite, which has local overhead overpass times of 01:30 and 13:30 and we use data from both overpasses. Therefore, when comparing to AMSR-E for LWP and RWP the model local times within 3 hours either side of both of these times are used. This is important for LWP and RWP because cloud water content can have a large diurnal cycle.

Fig. 4 shows $f_{\mathrm{LWP}}$ from the model and the AMSR-E satellite along with the model bias relative to the retrieval. $f_{\mathrm{LWP}}$ is quite

high throughout the region in both the model and the observations, with values over the ocean generally larger than around 0.8. The model bias is generally small ($<10\%$) except off the east coast of the USA where there is an overestimate, although with biases mostly less than 0.1 ($\sim 15\%$). This region corresponds to a region of high LWP (see Fig. 5) and might suggest that too much rain is produced by the model. Although, given that $f_{\mathrm{LWP}}$ is low, it is also a region where the satellite estimate of $f_{\mathrm{LWP}}$ is likely to be more uncertain. The largely good agreement between the model and satellite for $f_{\mathrm{LWP}}$ provides some confidence in

the ability of the model to represent cloud and rain formation processes, but insofar as we assume the model to be realistic, it also lends confidence to the satellite estimates of this quantity for this region, which, as explained above, is also uncertain.

$LWP_{\mathrm{ic}}$ values were estimated by dividing the time-mean all-sky LWP by the time-mean low-altitude cloud fraction from CALIPSO. The same was done for the model using the time-mean CALIPSO low cloud fraction from the COSP satellite simulator. For reference, the evaluation of the grid box mean LWP (including cloudy and clear regions) is shown in Appendix C.

Fig. 5 evaluates the time-mean $LWP_{\mathrm{ic}}$ with no filtering using $f_{\mathrm{LWP}}$. The bias plot in Fig. 5 shows the 0.9 value of the time-mean satellite $f_{\mathrm{LWP}}$ contour to highlight regions where the satellite data are likely to be more reliable. The spatial pattern of the model and satellite have a spatial correlation r value of 0.68. There is a negative model bias off the east coast of Florida, although there is generally a lot of rain present in this region as indicated by the $f_{\mathrm{LWP}}$ contour and Fig. 4, such that the satellite measurements are more uncertain. There is also a negative model bias between Greenland and Canada and on the east side

of Greenland where the $f_{\mathrm{LWP}}$ is high. The grid-box mean LWP is also biased low there (Fig. C1) indicating that the model clouds are too thin (rather than the cloud fraction too low; see also Fig. 1). Positive biases occur off the west coast of Africa at around 18°N, however, the $LWP_{\mathrm{ic}}$ is very low in this region. Further positive biases occur in the stratocumulus to the north of Scandinavia. Again, these are not associated with cloud fraction biases (see Figs. C1 and 1) suggesting model stratocumulus that are too thick.





Fig. 6 compares $LWP_{ic}$ where $f_{LWP} > 0.99$ for both the model and satellite (hereafter $LWPic_{0.99}$). This selects cloud scenes that are likely to be non-precipitating and non-convective clouds, which are likely to be either stratocumulus or shallow cumulus clouds. The regions of stratocumulus north of Scandinavia and around Iceland where there were positive biases with no filtering for $f_{LWP} > 0.99$ now have even larger $LWP_{ic}$ biases, again indicating that the model low clouds may to be too thick. The negative biases west of Greenland remain. Since there is generally little RWP in this region and the $f_{LWP}$ filtering has been performed, this indicates that the $LWP_{ic}$ evaluation in this region is likely to be robust. There are now negative

biases extending from the coast of Florida up to near the UK suggesting that clouds there are too thin, although instrumental uncertainties may still be high here given the uncertainty in the satellite estimate of $f_{LWP}$ and the larger amount of RWP that occurs here.

     The $LWP_{ic}$ values from the AMSR-E observations in Fig. 6 show remarkably little spatial variability, suggesting that

non-precipitating clouds tend to be fairly uniform across broad regions with an $LWP_{ic}$ value of around 60–100 g m$^{-2}$. Cloud fraction (Fig. 1) and $N_d$ (Fig. 7) vary quite widely across the same region potentially indicating a physical process that maintains a constant $LWP_{ic}$ despite a varying aerosol environment and cloud regime.

### 3.1.3    Cloud droplet concentration evaluation

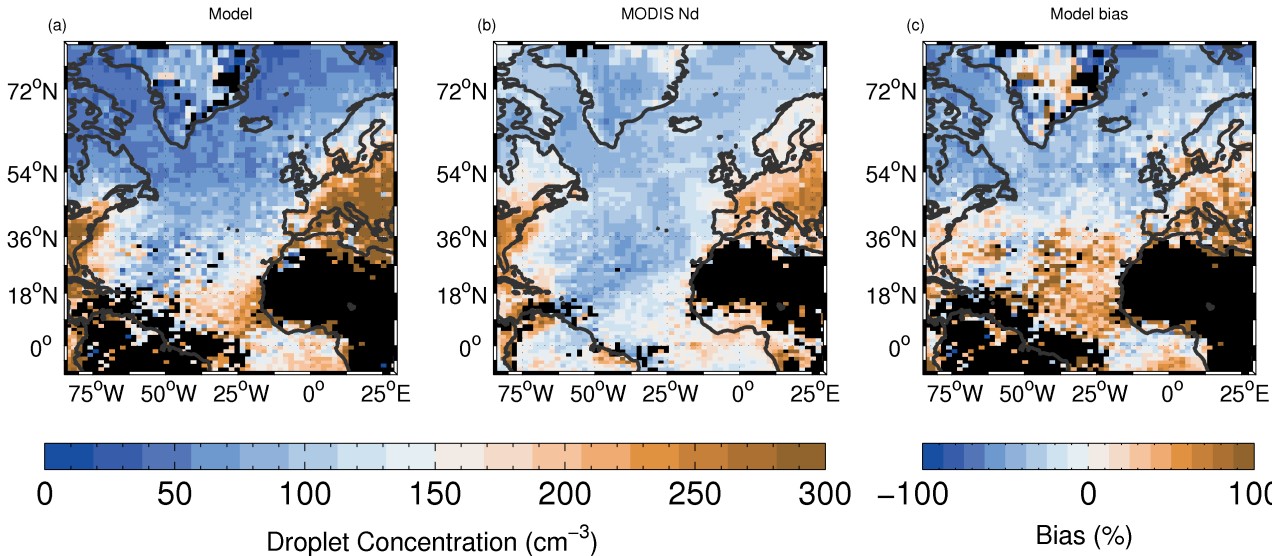

**Figure 7.** Time-mean model $N_d$ evaluation vs the MODIS satellite. The model and MODIS data are restricted to datapoints for which the grid-box mean cloud top height is $<= 3.2$ km and for which the liquid cloud fraction is $>= 80\%$. The COSP liquid cloud fraction is used for the model screening. The arrangement of the panels is as for Fig. 1.

     Cloud droplet concentration ($N_d$) is an important quantity because it generally represents the first step in the chain of pro-

cesses by which aerosols affect clouds. $N_d$ gives some indication of the number of CCN that were available to produce clouds





as well as other factors such as updraft speed, droplet collision coalesence, droplet scavenging by rain, cloud evaporation, etc. Thus, an evaluation of model $N_d$ can give an idea of how well the model captures a range of processes.

Here we evaluate $N_d$ using a $1\times1°$ resolution data set calculated from MODIS retrievals of $\tau_c$ and $r_e$. Two-dimensional fields of $N_d$ are derived by the retrieval since it is assumed that $N_d$ is constant throughout the depth of the cloud, which has

been shown to be a good approximation by aircraft observations of stratocumulus (Painemal and Zuidema, 2011). Details of the retrieval are given in Appendix A. For the model, two-dimensional $N_d$ fields were obtained from the instantaneous 3D $N_d$ fields by calculating a weighted vertical mean $N_d$, with the LWC on each level used for the weights. This ensures that the levels with the most LWC contribute most to the average $N_d$, which is similar to what is assumed in the MODIS retrieval since most of the $r_e$ signal comes from near cloud top where the LWC is assumed to be the largest and the $N_d$ calculation is a strong

function of $r_e$. It also avoids contributions from very thin clouds that would not be detected by MODIS. Only datapoints for which the liquid cloud fraction is larger than 80% and for which the mean cloud top height is below 3.2 km were included for the satellite $N_d$ calculation in order to help prevent satellite retrieval errors (see Grosvenor et al., 2018b). The same filtering was applied to the model to minimise sampling errors; the COSP MODIS liquid cloud fraction and a calculation of model cloud top height were used for this. The satellite data set was re-gridded to the model grid before comparison.

Figure 7 shows that the modelled and observed $N_d$ are largest near the continents and that $N_d$ decreases towards the middle of the Atlantic Ocean. This fits with the idea that CCN are scavenged during eastward transport (Wood et al., 2017). There are also likely to be some dilution effects as distance from the sources increases. Other factors may also influence the spatial $N_d$ pattern such as changes in boundary layer height, predominant cloud type, other meteorological factors, etc. CCN concentrations also increase close to the European and African source regions, consistent with the prevailing northerly wind, which

transports European and African pollution down the coast. Overall, the model produces a reasonable spatial pattern with a spatial correlation coefficient of 0.67 for the whole region. However, the model has a tendency to overestimate $N_d$ over the southern part of the North Atlantic region, off the east coast of the UK and over Europe. Conversely the model underestimates in the northern part of the domain. The NMBF is -20.6% for the northern NA region where stratocumulus dominates compared to 21.5% for the southern NA region where the clouds are more broken.

### 3.1.4    Shortwave top of the atmosphere flux evaluation

CERES-EBAF data was taken from https://ceres.larc.nasa.gov/order_data.php and is the monthly averaged product of observed TOA for which the TOA net flux is constrained to the ocean heat storage.

Figure 8 shows the evaluation of the time-mean top of the atmosphere SW radiative fluxes ($SW_{TOA}$) versus the CERES (Clouds and the Earth's Radiant Energy System) satellite. The spatial pattern of the model matches the observed pattern well

with a correlation coefficient of 0.89. The model biases are mostly small with positive biases that are $<\sim20\%$ over all of the oceanic regions except south of the equator where positive biases of up to $<\sim35\%$ occur. The NMBF bias for the whole region (including land) is -2.4% and it is 5.8% and 1.7% for the northern NA and southern NA regions, respectively.

The $SW_{TOA}$ biases can be caused by a combination of biases in $f_c$, $TLWP_{ic}$ and $N_d$. To estimate the relative contributions of these different biases individually to the $SW_{TOA}$ bias we used a similar calculation to that described in Section 2.4, but





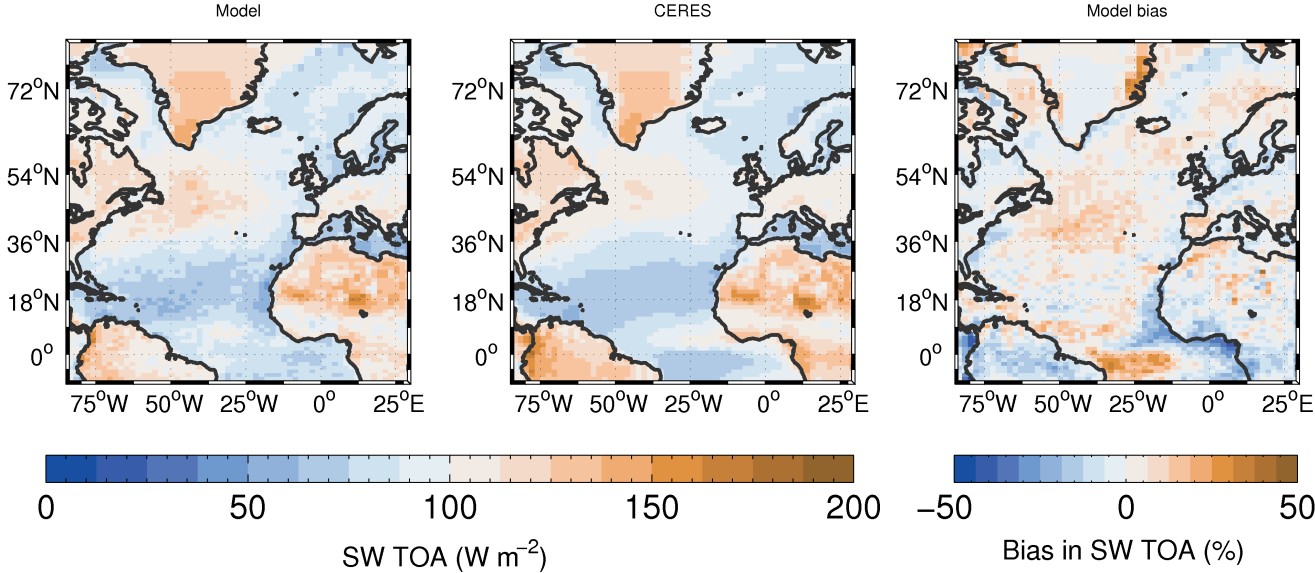

**Figure 8.** Time-mean model $SW_{\text{TOA}}$ evaluation vs the CERES satellite. The monthly mean CERES-EBAF data product is used here; this product uses data from both the Terra and Aqua satellites as well as geostationary satellites in order to approximate averaging across the diurnal cycle. The arrangement of the panels is as for Fig. 1.

applied to TOA fluxes, following the technique described in Grosvenor et al. (2017). Firstly, the modelled time-mean $SW_{\text{TOA}}$ field was reconstructed by using the modelled time-mean values of the COSP CALIPSO $f_{\text{c}}$, $LWP_{\text{ic}}$ (with no filtering using $f_{\text{LWP}}$) and $N_{\text{d}}$ (filtered as in Section 3.1.3) as inputs into the $SW_{\text{TOA}}$ offline calculation. This was also done using the time-mean observed fields from CALIPSO, AMSR-E and MODIS. The results are shown in Fig. 9 and can be compared to Fig. 8 a and b. Comparisons are not made over land since the higher land albedo was not taken into account in the calculations. In both cases,

the spatial pattern of the calculated $SW_{\text{TOA}}$ (Figs. 9 a and b) over the ocean correlates well with that of the actual $SW_{\text{TOA}}$ (Figs. 8 a and b).

The $SW_{\text{TOA}}$ values from the calculations are somewhat smaller than those actually modelled or observed. If considering only oceanic regions, the underestimates for the model off the east coast of Canada in the N. Atlantic stratocumulus region are perhaps of greatest concern since this is where the $SW_{\text{TOA}}$ values and their biases were highest. The underestimates are likely

due to the approximations and assumptions that have been made with this approach such as using the time-mean shortwave flux and cloud fields. However, the agreement is sufficient for the purposes of estimating the relative contributions from biases in individual cloud properties to the $SW_{\text{TOA}}$ bias.

Next, perturbations to $SW_{\text{TOA}}$ were calculated by applying the model biases in the individual cloud fields ($f_{\text{c}}$, $LWP_{\text{ic}}$ and $N_{\text{d}}$) to the observed cloud values on a one-at-a-time basis (Fig. 10). They are expressed as a percentage of the observed time-

mean $SW_{\text{TOA}}$ field. The sum of these perturbations, again expressed as a percentage, is shown in Fig. 9c and is intended for




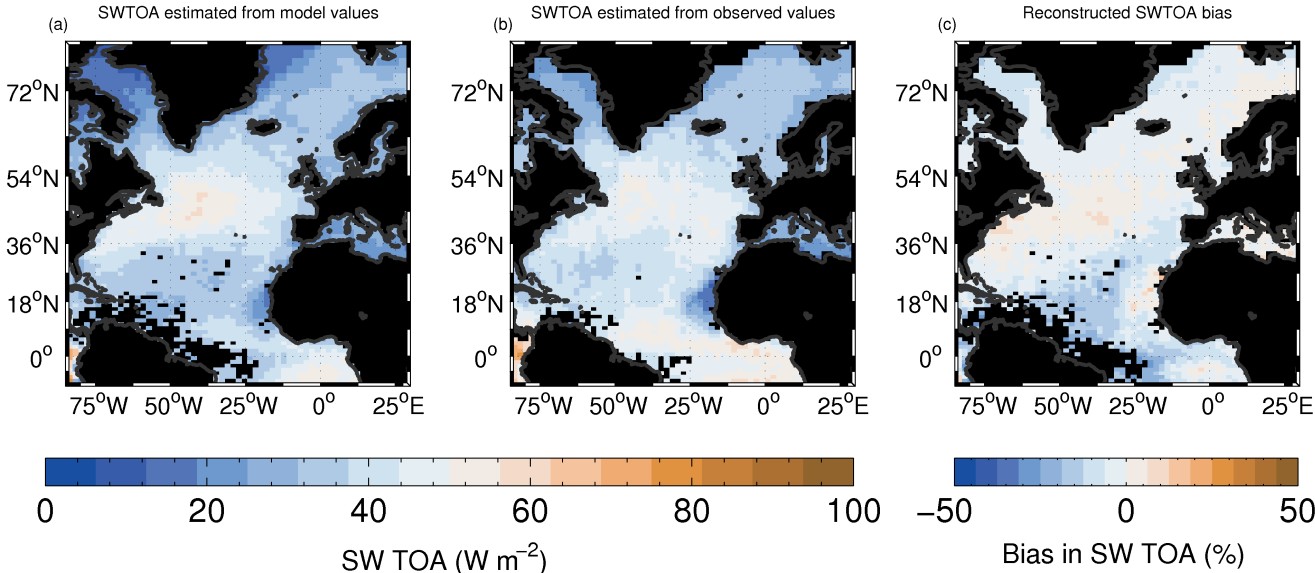

**Figure 9.** Estimates of the time-mean $SW_{TOA}$ flux (left and middle; $W m^{-2}$); left: calculated using the time-mean modelled cloud properties (for comparison with Fig. 8a); middle: calculated using the time-mean observed cloud properties (for comparison with Fig. 8b). The plot on the right shows the sum of the estimated contributions from the individual cloud property biases (see Fig. 10) expressed as a percentage of the observed CERES $SW_{TOA}$ flux (for comparison with Fig. 8c).

comparison to the actual model $SW_{TOA}$ bias shown in Fig. 8c. The spatial pattern of the combined bias estimate matches the actual bias well, although there are some regions of negative bias that are not present in the actual bias. These are in regions of low $SW_{TOA}$ and low $f_c$ and so are more prone to error and likely less important for the overall $SW_{TOA}$. In the regions of positive model $SW_{TOA}$ bias the estimate is a little lower than the true model bias, but again, the agreement should be sufficient
to compare the relative contributions from the individual cloud field biases.

From Fig. 10 it is clear that perturbations of the magnitude of the $f_c$ model biases have a very large impact on the $SW_{TOA}$ and are the main contributor to model $SW_{TOA}$ biases in most regions. Large negative contributions occur to the $SW_{TOA}$ bias in the southern part of the domain due to the negative cloud biases there (Fig. 1). Smaller positive contributions from $N_d$ and $LWP_{ic}$ biases also occur in this region to give less overall estimated $SW_{TOA}$ bias (Fig. 9c) in agreement with the small $SW_{TOA}$
model bias (i.e., that directly from the model output, Fig. 8c). This suggests that some cancellation of biases in $f_c$, $LWP_{ic}$ and $N_d$ is occurring here resulting in the observed low $SW_{TOA}$ bias. Large positive contributions to the $SW_{TOA}$ bias from $f_c$ biases occur in the stratocumulus region in the western part of the northern NA. These are also partially cancelled by negative $N_d$ and $LWP_{ic}$ biases, but the overall $SW_{TOA}$ is positive in this region (Fig. 8c). $LWP_{ic}$ (note the different colour scale) has only a minor effect north of Scandinavia (contributing to positive biases) and west and east of Greenland (negative contributions).





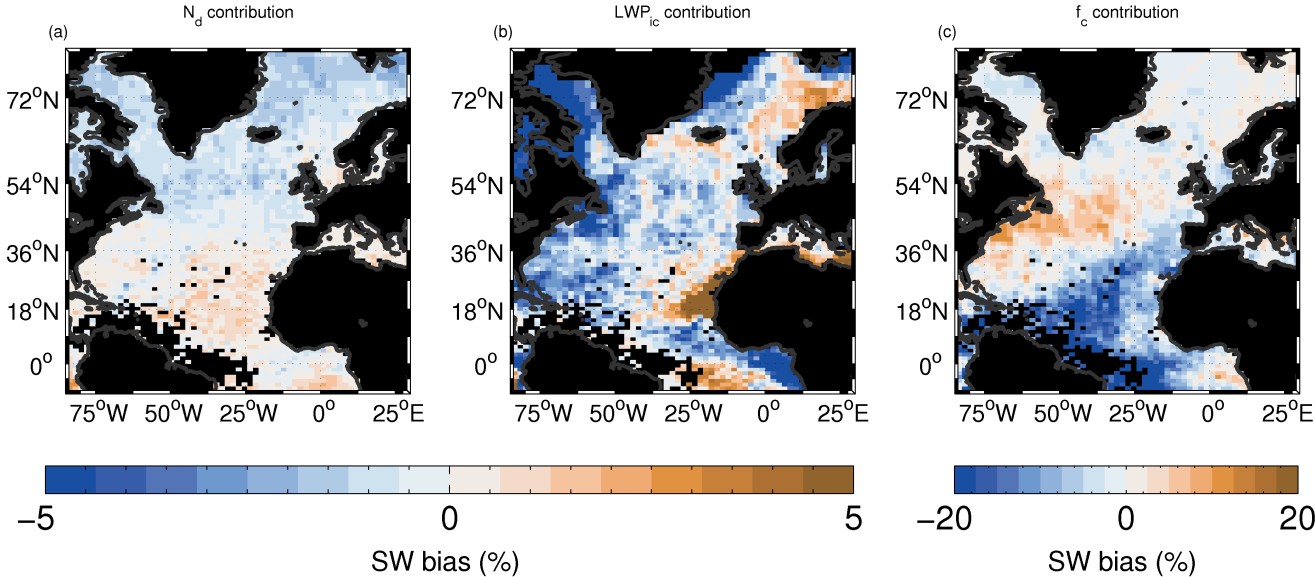

**Figure 10.** Estimated contributions from the model biases in $N_d$ (left; note the smaller colourbar range), $LWP_{ic}$ (middle) and $f_c$ (right) to the $SW_{TOA}$ model bias (Fig. 8c). Values are expressed as a percentage of the observed CERES $SW_{TOA}$ field for consistency with Fig. 8c. Fig. 9c shows the sum of these three contributions also as a percentage of the observed CERES $SW_{TOA}$.

The $N_d$ biases contribute even less to the $SW_{TOA}$ model bias than $LWP_{ic}$ biases; positive contributions from $N_d$ biases are seen south of 36ºN and negative contributions north of there.

Next, an attempt to quantify the relative percentage contributions ($P_x$) of the individual cloud field properties (denoted as x, where x is either $f_c$, $LWP_{ic}$, or $N_d$) to the total $SW_{TOA}$ bias is made using the following equation :-

$$P_x = \frac{100 \times \Delta SWTOA_x}{|\Delta SWTOA_{f_c}| + |\Delta SWTOA_{LWP_{ic}}| + |\Delta SW_{N_d}|} \tag{3}$$

where $\Delta SWTOA_x$ is the perturbation in $SW_{TOA}$ due to the bias in cloud field property x. Figure 11 shows that $f_c$ biases dominate over nearly all regions south of around 54ºN. However, there are some regions where $LWP_{ic}$ biases dominate; there are large positive relative contributions from $LWP_{ic}$ in the region north of Scandinavia and some large negative contributions surrounding Greenland. $N_d$ contributions are important in the diagonal band stretching from the southwest of Iceland up to Svalbard.

Overall, this analysis shows that positive biases in $f_c$ are important in explaining the positive $SW_{TOA}$ biases east of the USA and Canada, while positive $LWP_{ic}$ biases are important off the northern coast of Scandinavia.



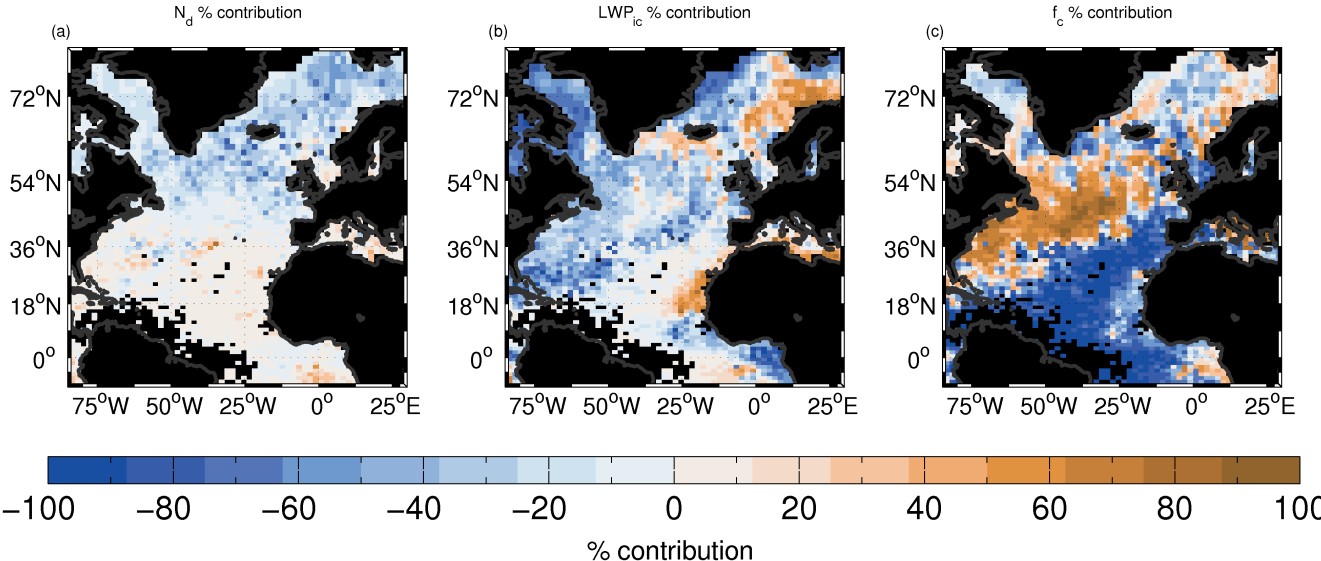

**Figure 11.** The percentage of the sum of the absolute biases contributed by the biases in $N_d$ (left), $LWP_{ic}$ (middle) and $f_c$ (right). See Eq. 3.
Note, that the absolute values of the percentages for each grid-box add up to 100%.

### 3.2 Aerosol Forcing

We now examine maps of the temporal mean aerosol forcings following Eqn. 2. Note that the calculations were performed using the instantaneous fields from the 27-hourly output, which ensures that the diurnal cycle is sampled evenly throughout the course of the simulation.

Figure 12 shows maps of $ERF_{ARI}$ and $ERF_{ACI}$. The magnitude of $ERF_{ACI}$ is generally larger than that of $ERF_{ARI}$ for oceanic regions. For example in the northern NA box the mean $ERF_{ARI}$ is -0.50 W m$^{-2}$ and $ERF_{ACI}$ is -3.1 W m$^{-2}$ (see Table 3). For the southern NA box $ERF_{ARI}$ and $ERF_{ACI}$ are -1.0 W m$^{-2}$ and -1.8 W m$^{-2}$, respectively. The dominance of $ERF_{ACI}$ in over the N. Atlantic ocean region is in agreement with B12 (Fig. 4b of that paper). The overall spatial pattern of $ERF_{ACI}$ is also similar to that in B12 with a band of large negative forcing running across the Atlantic from west to east between around 25 and 50º N, and another region of negative forcing following the west coast of Africa. Some differences are apparent, though; for example, the negative forcing is smaller in magnitude southwest of the UK in our simulations and larger in magnitude near Africa south of the equator. However, over the whole of the NA domain $ERF_{ARI}$ is larger in magnitude than $ERF_{ACI}$ (-1.9 vs -1.7 W m$^{-2}$) due to the fact that $ERF_{ACI}$ is usually larger over land. The dominance of $ERF_{ARI}$ in the southern regions of the domain will also have more influence on the area-weighted average.

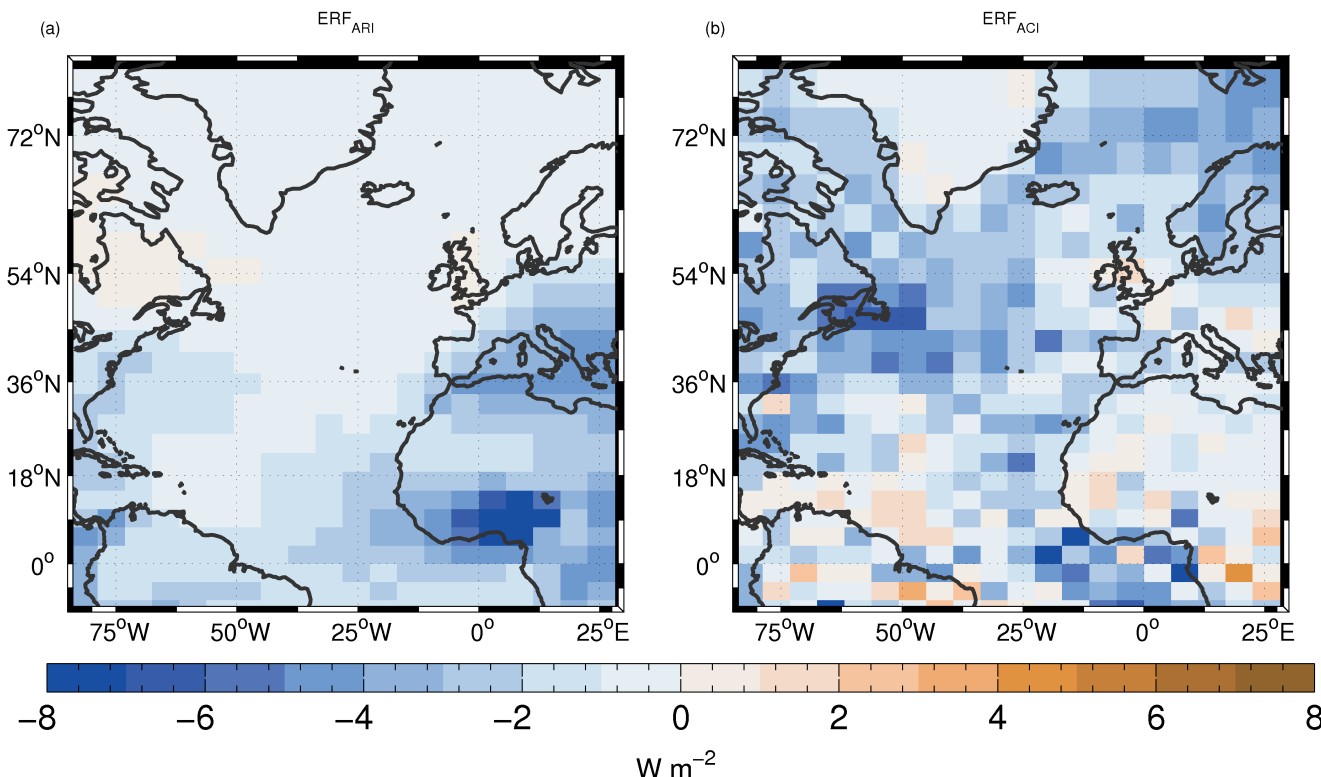

**Figure 12.** Time-mean $ERF_{ARI}$ (left) and $ERF_{ACI}$ (right) forcing. The maps have been averaged over 3x3 grid boxes to help reduce noise.

### 3.2.1 Changes in cloud properties from PI to PD and their contributions to the ACI forcing

Fig. 13 shows the time-mean changes in cloud properties ($N_d$, $LWP_{ic}$ and $f_c$) between the PI and PD simulations (PD minus PI as a percentage of PI). Considering oceanic regions, percentage increases in $N_d$ are greatest off the east coast of the USA and Canada (to the south of Newfoundland); off the SW coasts of Spain/Portugal and West Africa, and in the region north of

Scandinavia. There are increases across the whole Atlantic running from the USA to West Africa, but the magnitude decreases moving east (except close to the European/African west coast) likely reflecting the decreasing influence of pollution from the east coast of the USA.

$LWP_{ic}$ changes are generally quite noisy, but the largest changes over the ocean occur west of Greenland and in a diagonal band across the Atlantic in a similar way to the $N_d$ changes except further to the north. The changes in $f_c$ are largest in the

southern regions of the domain below around 35º S. The responses of $LWP_{ic}$ and $f_c$ can be termed macrophysical responses (or also adjustments) since they affect bulk cloud properties, as opposed to the response of $N_d$ that is termed a microphysical response. The macrophysical responses are mostly due to the suppression effects of aerosols on rain rates via the autoconversion





**Table 3.** Temporal and spatial means of various quantities for the whole NA domain and the various sub-regions. Values are shown for $ERF_{ARI}$ and $ERF_{ACI}$, and the percentage chagnes in $N_d$, $LWP_{ic}$ and $f_c$ between the PI and PD runs. Percentage $f_c$ changes are also shown for the runs with no aerosol effect on the rain autoconversion process. The other columns show results from the offline estimates of the ACI forcing. $ERF_{ACI,all}$ is the estimated forcing due to simulataneous PI to PD perturbations of $N_d$, $LWP_{ic}$ and $f_c$. $ERF_{Nd}$, $ERF_{LWPic}$ and $ERF_{fc}$ are the forcing estimates from one-at-a-time perturbations of $N_d$, $LWP_{ic}$ and $f_c$, respectively, and $ERF_{ACI,sum}$ is the sum of these three forcing estimates. $ERF_{ACI,macro}$ is the forcing contribution from the cloud macrophysical changes, i.e., the sum of $ERF_{LWPic}$ and $ERF_{fc}$. All values are area-weighted to account for the variation in area of the model grid-boxes.

| # | Region name | $ERF_{ARI}$ $(Wm^{-2})$ | $ERF_{ACI}$ $(Wm^{-2})$ | % change in $N_d$ | % change in $LWP_{ic}$ | % change in $f_c$ | % change in $LWP_{ic}$, no aerosol autocon. | % change in $f_c$, no aerosol autocon. | $ERF_{ACI,all}$ $(Wm^{-2})$ | $ERF_{ACI,sum}$ $(Wm^{-2})$ | $ERF_{Nd}$ $(Wm^{-2})$ | $ERF_{LWPic}$ $(Wm^{-2})$ | $ERF_{fc}$ $(Wm^{-2})$ | $ERF_{ACI,macro}$ $(Wm^{-2})$ |
|---|---|---|---|---|---|---|---|---|---|---|---|---|---|---|
| 1 | NA | -1.9 | -1.7 | 61.5 | 3.2 | 1.5 | -5.0 | 0.14 | -1.7 | -1.8 | -0.67 | -0.41 | -0.74 | -1.15 |
| 2 | Northern NA | -0.50 | -3.1 | 44.2 | 2.7 | 0.94 | -0.43 | 0.02 | -3.0 | -3.2 | -1.4 | -1.2 | -0.54 | -1.74 |
| 3 | Southern NA | -1.0 | -1.8 | 55.3 | -0.59 | 2.7 | -0.50 | -0.75 | -1.9 | -2.0 | -0.63 | -0.23 | -1.2 | -1.43 |

process. This is demonstrated by Figs. D1 and D2, and Table 3 that show the impact of preventing aerosol from affecting the autoconversion process (see Appendix D for details). We hypothesize that the changes in $LWP_{ic}$ over the northern NA regions
and lesser changes in $f_c$ reflects the presence of stratocumulus clouds with very large areal coverage in the north that can only respond to the suppression of rain by thickening (i.e., more $LWP_{ic}$) rather than by increasing $f_c$. In the southern NA the cloud coverage is much less due to the presence of broken stratocumulus and cumulus clouds, which respond in a different way to the rain suppression, namely by increasing their coverage ($f_c$) more than their thickness ($LWP_{ic}$).

Based on the maps of the cloud property changes alone it is difficult to judge the relative contributions of each type of
change to the forcing. Thus, we now make a quantitative estimate of this using offline radiative calculations as described in Section 2.4 and Appendix E. Fig. 14a shows the ACI forcing estimated using the offline method following Eq. E13 for the case where all of the cloud variables have been perturbed from their PI values to their PD values ($ERF_{ACI,all}$). Fig. 14b shows the sum of the contributions calculated by separately perturbing the individual cloud properties in one-at-a-time tests ($ERF_{ACI,sum}$). These can be compared to the forcing diagnosed from the full model outputs as generated by the online
radiation code (Fig. 12b). The general pattern and magnitude of the actual and estimated forcings are very similar. The fact that $ERF_{ACI,all}$ and $ERF_{ACI,sum}$ are similar indicates linearity in the effect of the perturbation of the individual cloud properties, such that they are almost directly additive. The mean forcing over the northern sub-region (northern NA) is -3.2 W m$^{-2}$ for $ERF_{ACI,sum}$, -3.0 W m$^{-2}$ for $ERF_{ACI,all}$ and -3.1 W m$^{-2}$ for the actual forcing. In the southern sub-region (southern NA) the mean forcings are -1.9, -2.0 and -1.8 W m$^{-2}$ for $ERF_{ACI,sum}$, $ERF_{ACI,all}$ and the actual forcing, respectively. Overall,





440 the match is very good suggesting that the estimation technique is sound and that it is likely to be useful for estimating the contributions from the different cloud property changes to the forcing.

Figure 15 shows the contribution to the surface ACI forcing from the changes in $N_d$, $LWP_{ic}$ and $f_c$ following Eqn. E15. Percentage contributions ($P_x$) from the individual cloud fields (denoted as x, where x is either $N_d$, $f_c$, or $LWP_{ic}$) to the sum of the absolute contributions are calculated in a similar way to Eqn. 3 :-

$$P_x = \frac{100 \times ERF_x}{|ERF_{Nd}| + |ERF_{LWPic}| + |ERF_{fc}|} \tag{4}$$

where $ERF_x$ is the ACI forcing contribution due to the change in cloud field x. These values are mostly negative since the ACI forcing is negative, but positive contributions and thus positive $ERF_x$ are possible. Figure 16 shows the results.

From the summary of the area means in Table 3 we see that for the northern NA region the area-mean contributions from changes in $N_d$ and $LWP_{ic}$ ($ERF_{Nd}$ and $ERF_{LWPic}$) are similar with values of -1.4 and -1.2 W m$^{-2}$, respectively. These are

450 larger than the $f_c$ contribution ($ERF_{fc}$) of -0.54 W m$^{-2}$. In contrast, for the southern NA region $ERF_{fc}$ is -1.2 W m$^{-2}$, which is considerably larger than the $N_d$ or $LWP_{ic}$ contributions (-0.63 and -0.23 W m$^{-2}$, respectively). Overall, for the whole NA region, $ERF_{fc}$ is -0.74 W m$^{-2}$, which is similar in magnitude to that from the $N_d$ changes (-0.67 W m$^{-2}$) with $ERF_{LWPic}$ a little smaller at -0.41 W m$^{-2}$.

The sum of the macrophysical contributions from $f_c$ and $LWP_{ic}$ (termed $ERF_{ACI,macro}$) is the dominant contribution to

455 aerosol forcing in the NA region providing 63.9 % of $ERF_{ACI,sum}$ ($=ERF_{Nd} + ERF_{LWPic} + ERF_{fc}$). This is important because these changes are likely to be associated with a higher model uncertainty than pure cloud albedo effects (due to $N_d$ changes alone). The macrophysical contribution to $ERF_{ACI,sum}$ is larger in the southern NA region than the northern NA region (71.5 % versus 54.4 %).

Figures 15 and 16 show the spatial patterns of the contributions and reveal that the contribution from $N_d$ is restricted to

460 the northern part of the domain (including the region of the northern NA sub-region) and is generally small elsewhere, except for a small contribution down the west coast of Africa. Somewhat surprisingly this contribution is not co-located with where the largest changes in $N_d$ are simulated in Fig. 13, but occurs to the north and east. This likely reflects the fact that the cloud fraction is large in this region due to the prevalence of stratocumulus, which results in a large radiative impact from even modest $N_d$ increases. A similar argument can be applied for $LWP_{ic}$ changes. The forcing contribution from $LWP_{ic}$ changes follows

465 a similar spatial pattern to that from $N_d$ changes and with similar magnitudes, although for $LWP_{ic}$ the largest $LWP_{ic}$ changes are co-located with the forcing contributions. The $N_d$ contribution is generally dominant in terms of Eqn. 4 north of 36ºN, but also provides a relatively large contribution down the west coast of Africa. The contribution from $f_c$ changes is largest in the southern NA region and around the coast of West Africa.





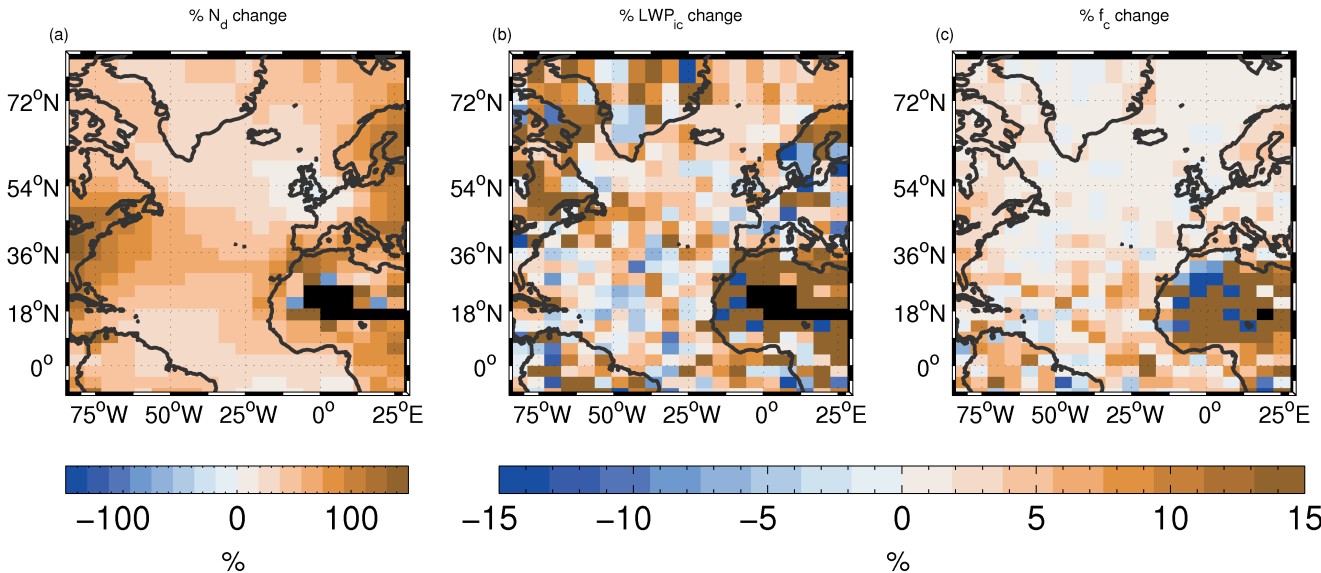

**Figure 13.** Mean percentage increase in $N_d$, $LWP_{ic}$ and $f_c$ between PI and PD runs.

### 3.2.2 Determining the most important meteorological situations for surface ACI forcing

We now attempt to determine the meteorological situations that are most important for aerosol forcing since this information can then be used to target particular situations for modelling at high resolution, or for more detailed comparisons to observations.

Initially we quantify the forcing as a function of "cloud state", which we define in terms of the 8 possible combinations of low-, mid- and high-altitude clouds (see Fig. 17), which are defined to be consistent with the ISCCP definitions (Rossow

et al., 1999): cloud top pressure (CTP) > 680 hPa for low-altitude cloud, 680<CTP<440 hPa for mid-altitude cloud and CTP<440 hPa for high altitude cloud. For this analysis cloud is considered present if the the cloud fraction is greater than 0.01. The results are presented in the form of 2D matrix plots showing the contribution to the overall $ERF_{ACI}$ for different combinations of PI and PD cloud states (Fig. 18) for both the northern NA and southern NA regions. The contributions take into account the frequency of occurrence of each pairing of PI and PD cloud states so that the values associated with the colours

in the plots add up to the overall regional forcing.

Generally for both regions the largest negative contributions to $ERF_{ACI}$ are associated with low-altitude cloud states (cloud states that have even numbers). To some extent this is expected since the convection scheme, which is likely one of the sources of higher altitude clouds, is not currently coded to respond to aerosol and because the lower-altitude clouds are closer to the surface aerosol sources. For the northern NA region, the largest terms are those with the same low cloud state in the PI and PD

(i.e., the diagonal terms for the even numbered states in the matrix). Although it should be noted that the diagonal terms could



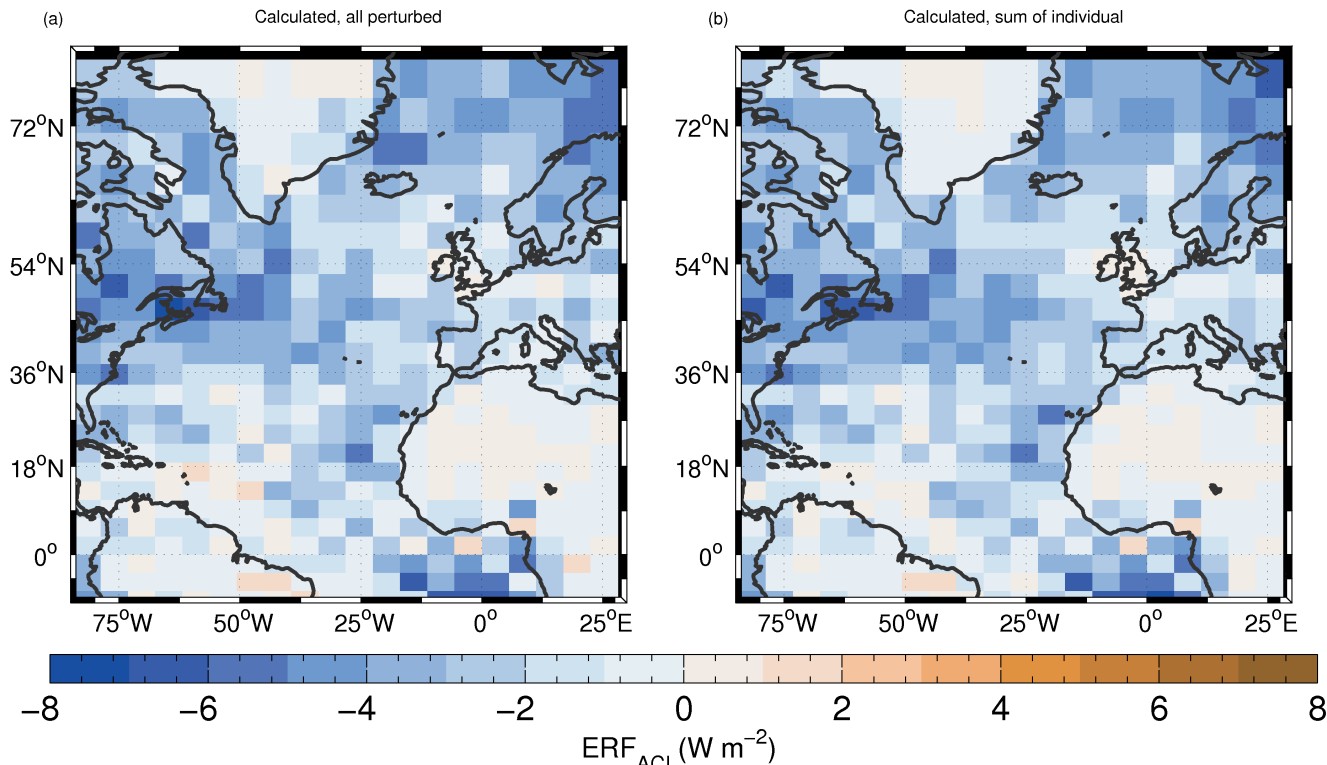

**Figure 14.** Time-mean surface ACI forcing ($Wm^{-2}$) calculated using different techniques. Left: estimated surface ACI forcing calculated using the changes in LWP$_{in-cloud}$, $N_d$ and cloud fraction between PD and PI simultaneously, according to Eqn. E13; right: the sum of the estimated contributions from the individual changes (Eqn. E15).

involve $f_c$ or $LWP_{ic}$ changes within these states. The largest overall term occurs when both the PI and PD are in cloud state 8 (i.e., no change in cloud state). This is when low, mid and high altitude clouds are present at the same time. However, if the mid and high altitude clouds are thin, it is still possible that the low-altitude cloud is having the largest impact here. The next two (joint) largest forcing contributions occur when both the PI and PD have only low clouds (state no. 2) and when low clouds
and high clouds are present together (state no. 6).

However, contributions from transitions between cloud states are not negligible for the northern region. For example, PI to PD transitions between state 1 and 2 (clear sky in the PI and low cloud in the PD) contribute -0.35 W m$^{-2}$. However, there is a reciprocal positive response of 0.30 W m$^{-2}$ for transitions between state 2 and 1, which represents the removal of low cloud that was present in the PI to create clear skies in the PD. There is a certain degree of randomness that is likely in the cloud state
changes due to all of the changes not necessarily being driven by aerosol; there is still some model freedom despite the nudging to meteorology since the model is only nudged above the boundary layer and only the winds are nudged. Thus, it is perhaps more useful to consider the net forcing contribution from both the PI to PD transitions and those from the reciprocal transitions

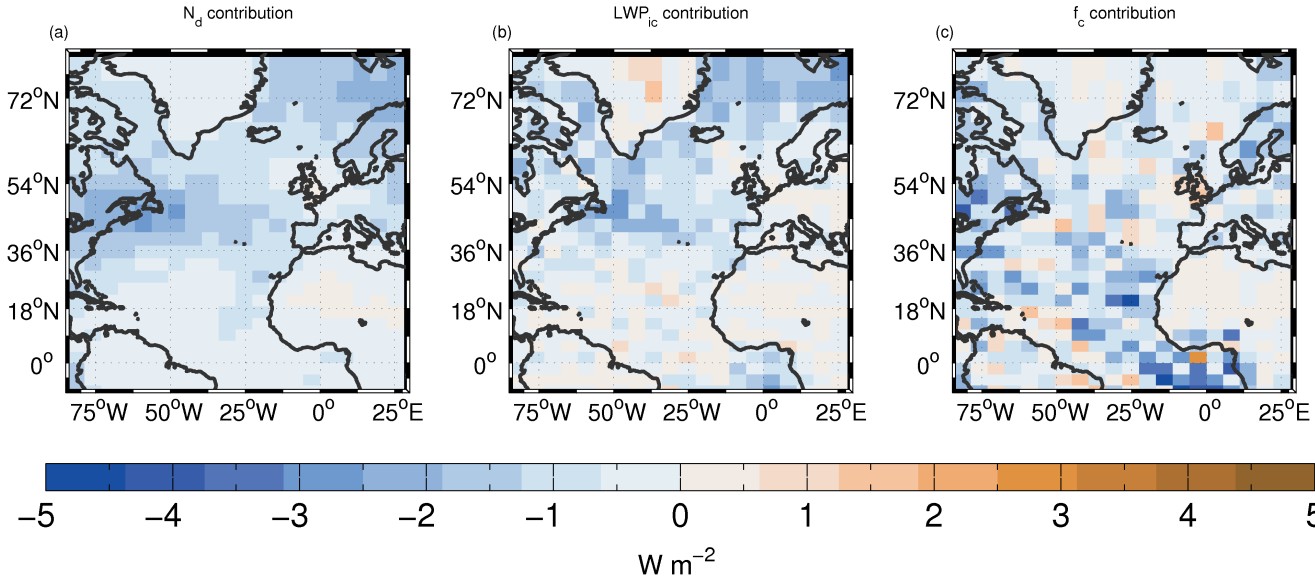

**Figure 15.** Estimated contributions to the surface ACI forcing from changes in $N_\mathrm{d}$ (left), $LWP_\mathrm{ic}$ (middle) and $f_\mathrm{c}$ (right). Fig. 14c shows the sum of these three terms.

as shown in the bottom row of Fig. 18. Appendix F explains the details of how these are produced. Other transitions have a larger net negative effect such as the transitions between states 6 and 8 (net effect of -0.18 W m$^{-2}$). This indicates transitions

from low+high altitude cloud to low+mid+high cloud, suggesting that the creation of mid-altitude clouds is also having an impact. Nevertheless, these transition terms are considerably smaller than the diagonal (no transition) terms described earlier for the northern NA region.

For the southern NA region, the low-altitude-only state (no. 2) in PI and PD again has a large impact (-0.53 W m$^{-2}$). There is also now a large negative contribution due to transitions from the clear state (no. 1) in the PI to the low-altitude only state

2 in the PD, but again there is a positive forcing due to the reverse situation, with a net contribution of -0.29 W m$^{-2}$. This transition indicates the creation of additional low-altitude cloud in the PD due to aerosol effects, which is consistent with the finding in the previous section that macrophysical changes to clouds (in particular changes in $f_\mathrm{c}$) are more important in this region compared to the northern NA region and suggest that low-altitude cloud is one of the most important cloud types for this. However, as in the northern NA region pairings with the same cloud states in the PI and PD (the diagonal terms) that involve

mid- and high-altitude clouds are also important for forcing, particularly for states 6 and 8, although these are relatively less important than in the northern NA region suggesting that higher altitude cloud has less effect on the forcing in the SSA region.





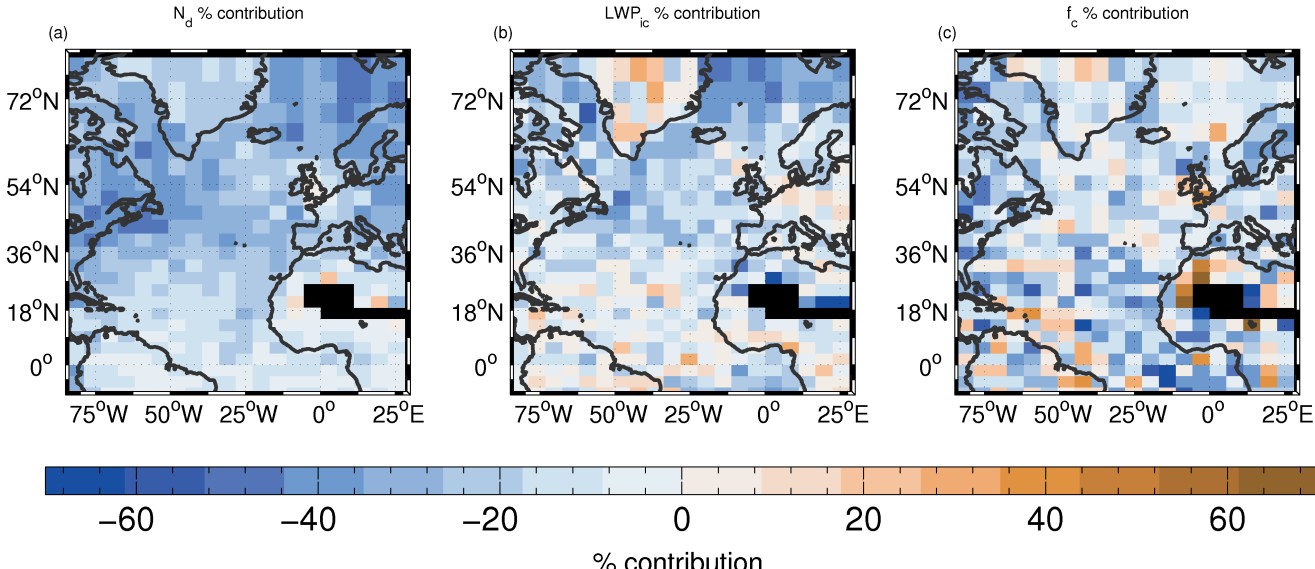

**Figure 16.** The percentage of the sum of the absolute contributions to the ACI forcing for the terms in Fig. 15. See Eqn. 4. Note, that the absolute values of the percentages for each grid-box add up to 100%.

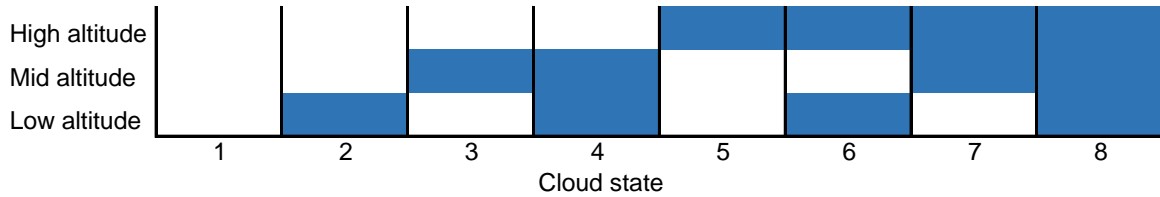

**Figure 17.** The cloud states used in the following figures. The shading indicates the presence of low, mid, or high altitude cloud (see text for the definitions of this) as determined by requiring the model cloud fraction to be larger than 0.01.

### 3.2.3 Determining the most important cloud types for surface ACI forcing

We now break down the forcing as a function of the PI and PD cloud fractions in order to determine the types of low cloud involved, i.e., whether they are stratocumulus (high cloud fraction) or trade cumulus (low cloud fraction). Given that the

contributions involving only cloud state 2 (low-altitude only clouds) were the single largest contribution in Fig. 18 for the southern NA region and provided the joint second-largest contribution for the northern NA (with low clouds also involved for the highest contributer) we examine the forcing contribution for this cloud state only, along with the clear sky state (no. 1).



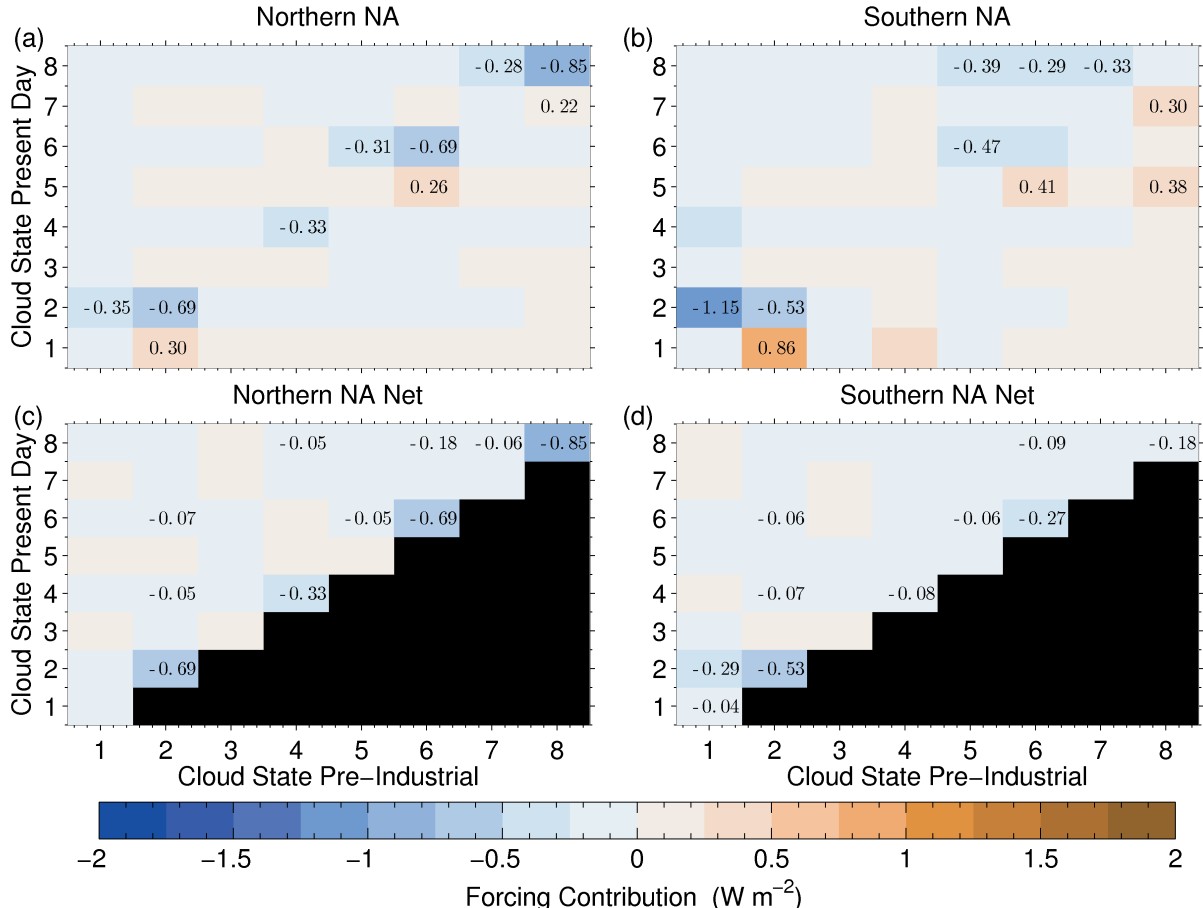

**Figure 18.** Contributions (colours; the highest 10 values in absolute magnitude are also labelled with text) to the surface ACI forcing for the different combinations of cloud state in the Pre-Industrial (PI) emission run (x-axis) and in the Present Day (PD) emission run (y-axis). The cloud states are the eight different possible combinations of low, mid and high altitude cloud (see Fig. 17). The overall contribution is shown, which includes the frequencies of occurrence such that the sum of all of the values gives the overall regional mean contribution. Left:the northern NA region; right: the southern NA region. The top row shows all of the combinations. In the bottom row plots the contributions due to PI to PD transitions between states are added to the same transition between PD and PI in order to get the net contribution; see Appendix F for details of this.

For the northern NA region Fig. 19 shows that by far the largest single term comes from the situation when both the PI and PD are fully overcast (-2.24 W m$^{-2}$). Consistent with the previous figures, this represents mainly the brightening of stratocumulus clouds due to increases in droplet concentrations combined with a smaller macrophysical effect from $LWP_{ic}$ increases. There are also some large negative contributions associated with increases in cloud fraction between PI and PD for this region too. The largest net contributions (ranging from -0.1 to -0.37 W m$^{-2}$) associated with cloud fraction changes



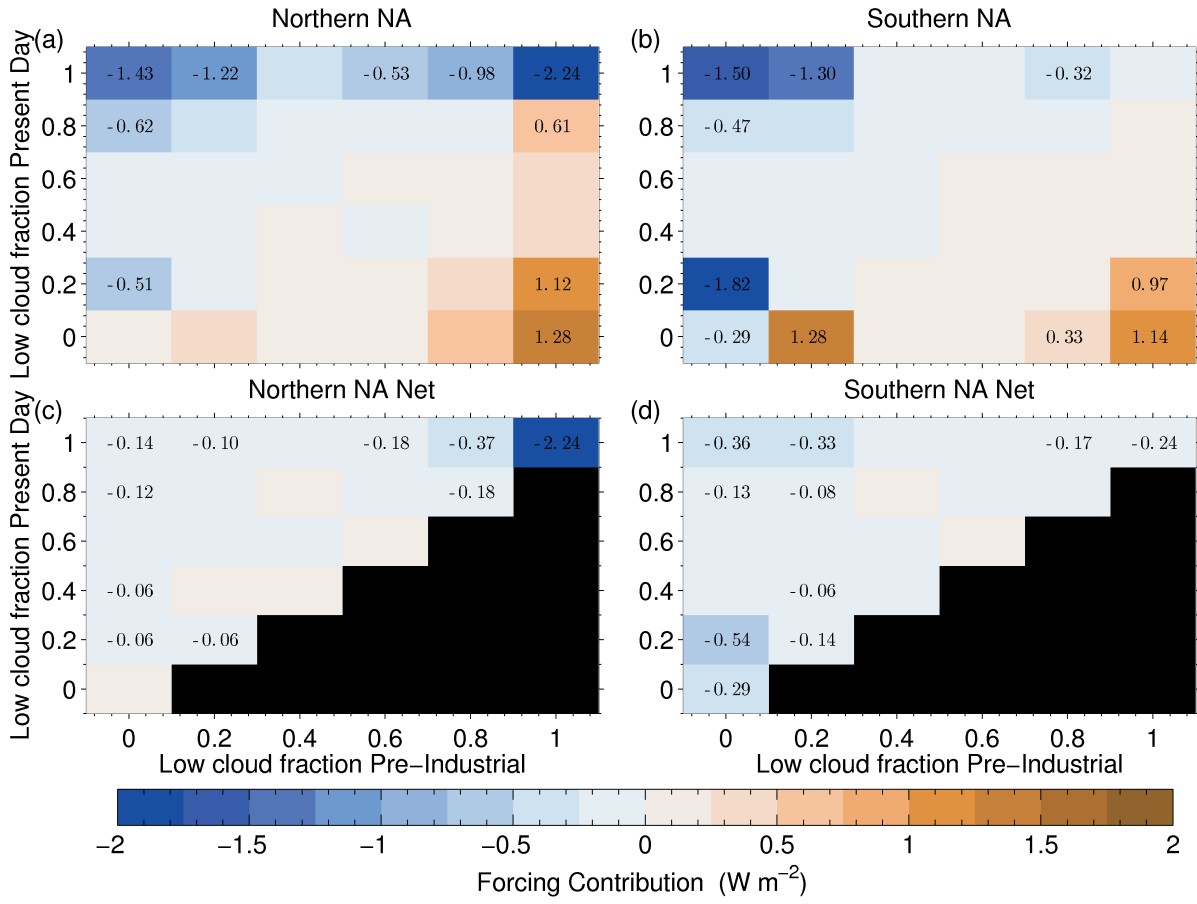

**Figure 19.** As for Fig. 18 except plotted as a function of the different combinations of areal low cloud fraction of the PI and PD runs.

come from the creation of fully overcast or $f_c$=0.8 stratocumulus from lower cloud fractions (ranging from 0 to 0.8) indicating that the creation of overcast clouds from more broken stratocumulus and cumulus is playing some role in this region. These
contributions sum to -0.91 W m$^{-2}$.

For the southern NA region the contribution from the overcast cloud state in both the PI and PD is much smaller (-0.24 W m$^{-2}$) compared to the northern NA indicating that pure microphysical (Twomey) changes are are less important. The net contribution from zero and overcast state combinations is (-1.50 + 1.14 = -0.36 W m$^{-2}$) and there are also large net contributions from PI $f_c$ values between 0 and 0.4 transitioning to PD values of 0.8-1.0. The creation of $f_c$=0.2 cloud from
clear skies is the most important single term associated with cloud fraction changes for the southern NA region, in contrast to the northern NA region. The Twomey effect for $f_c$<=0.2 clouds is also important. This indicates a relatively more important role for trade cumulus or open cell stratocumulus for the southern NA region. As also suggested from the previous results,





this indicates that cloud fraction changes are relatively more important than the Twomey effect for this region compared to the northern NA region.

## 4 Conclusions

Previous work (Booth et al., 2012) has suggested that cloud-aerosol forcing is the dominant driver of multi-decadal sea surface temperature (SST) variability in the North Atlantic region. In this paper we examined the cloud-aerosol response and surface aerosol forcing of a more modern version of the climate model used in Booth et al. (2012), namely the UKESM1-A, which is the atmosphere-only (i.e., non-ocean-coupled) version of the model used in the latest CMIP intercomparison (CMIP6). We

focused on the North Atlantic region and used one year of meteorologically nudged model output. The aims were to: i) examine the representation of clouds and cloud-aerosol interactions in order to identify potential biases in cloud properties compared to observations; ii) quantify the effect of cloud property biases on the shortwave (SW) top-of-the-atmosphere (TOA) fluxes; iii) determine the surface (downward) aerosol forcing response of the model; and iv) identify the most important meteorological situations/cloud types for the surface forcing. The latter two aims should allow a more targeted evaluation of the model forcing

in future work using observations and high resolution modelling in order to test the magnitude of the forcing from the low resolution climate model.

### 4.1 Model evaluation conclusions

The spatial pattern of low, mid and high altitude clouds in the model compare well against observations. However, in the southern North Atlantic (southern NA) the model underestimated the low and mid-level cloud fractions by -23.9 and -25.8%,

respectively. Low-altitude cloud biases in the northern NA sub-region were positive, but lower in magnitude (5.1%).

Low-altitude cloud fraction biases have the potential to significantly impact aerosol forcing since they are closest in altitude to the aerosol sources. If we assume that the PI cloud cover is biased by a similar amount to the PD cloud cover and assume no cloud adjustments to aerosol then the bias in forcing will be similar to the bias in $f_c$. The results from this paper suggest that the second assumption is reasonable for the northern NA region because the Twomey effect dominates. Thus we might expect

the 5.1% low-altitude $f_c$ bias there to make a small contribution to any error in forcing bias.

However, in the southern NA $f_c$ changes in response to aerosols (adjustments) were large, so the above assumptions are less valid and the effects on forcing less clear. The negative present-day $f_c$ biases could indicate a cloud fraction response to aerosol that is too low, which would cause subsequent negative forcing biases. On the other hand, a model $f_c$ that is too low may mean that the model is in a broken, precipitating cloud regime too often. Such regimes are thought to be more sensitive to aerosols

and more prone to produce cloud adjustments (Ackerman et al., 2004b). In this case the model forcing values would be too large. Further work would be needed to quantify the effect of the model $f_c$ biases on the aerosol forcing.

In-cloud Liquid Water Path ($LWP_{ic}$) was evaluated vs the AMSR-E microwave radiometer satellite instrument. The model percentage $LWP_{ic}$ biases are small in the northern part of the Atlantic where higher cloud fraction stratocumulus dominates and where the satellite retrievals are likely to be more reliable due to lower rain amounts. In the NE part of the domain (again





a region with likely reliable retrievals) there were some positive biases, particularly north of Scandinavia, which indicates that the stratocumulus in this region are too thick in the model. Elsewhere, off the east coast of Florida there tended to be negative $LWP_{ic}$ biases. When the raining data points were filtered out the biases tended to get worse, but the spatial pattern was preserved, suggesting that the biases are likely real rather than a result of artifacts in the observations or comparison method.

The overestimate of $LWP_{ic}$ in the model in the regions dominated by stratocumulus (northern part of the domain) has implications for its ability to simulate the correct cloud effective radius ($r_e$) given the correct cloud droplet concentration. This may confuse evaluation efforts using $r_e$. An incorrect simulation of cloud droplet size also has implications for the conversion of cloud water into rain (autoconversion), suggesting the model might convert cloud into rain too readily at a given cloud droplet concentration; this will affect rain rates, but may also make the response of cloud fraction more sensitive to aerosols and hence

enhance aerosol forcing. $LWP_{ic}$ is also likely to affect the sensitivity of the cloud albedo to cloud droplet concentration (and hence aerosol), which will also affect the magnitude of aerosol forcing.

Model cloud droplet number concentrations ($N_d$) were compared to satellite observations from MODIS. The model captures the observed spatial pattern well suggesting that the processes that govern the removal or dilution of aerosol as it travels eastwards from the American continent are broadly captured by the model. Such processes are likely to include: the scavenging

of CCN during precipitation (Wood et al., 2017); dilution effects as distance from the sources increases; variations in boundary layer height, which may affect aerosol scavenging due to cloud type and precipitation changes, and may cause the dilution of aerosol concentration over deeper boundary layers; as well as other unidentified meteorological effects. However, the model underestimates $N_d$ over most of the northern NA, but overestimates it over the southern NA, which could indicate that there are some issues with aerosol sources and/or scavenging. Cloud processes could also be involved, such as updraft speed distribution

errors, problems with the droplet activation scheme, or issues relating to droplet removal (evaporation, coagulation, etc.). A caveat to the conclusion that the model has biases in $N_d$ is that there is uncertainty in the MODIS $N_d$ observations that may be of the same magnitude, or more, as the model biases (Grosvenor et al., 2018b).

The positive model $N_d$ biases associated with the aerosol outflow regions of the USA and Europe have the potential to cause a positive bias in the aerosol forcing. In the northern parts of the Atlantic, which are less affected by anthropogenic

aerosol, the negative $N_d$ biases may indicate a lack of aerosols from natural sources (or too much scavenging). This would create pre-industrial background aerosol concentrations that are too low, leading to a positive forcing bias (e.g., Carslaw et al., 2013). Further investigation into the cause of the spatial $N_d$ pattern using model experiments; an examination of the realism of anthropogenic and natural aerosol sources; and quantification of the MODIS $N_d$ uncertainties (e.g., through comparison with aircraft data for this region) is therefore warranted.

Shortwave Top of the Atmosphere ($SW_{TOA}$) radiative fluxes from the model were compared to those from CERES. Again, the model reproduced the observed spatial pattern well indicating a general good fidelity of overall cloud positions and properties. However, there was also an overestimate in most regions, showing that the clouds were either too bright, or occur too frequently, or both.





The main cloud properties that determine $SW_{\text{TOA}}$ fluxes are $f_c$, $LWP_{\text{ic}}$ and $N_d$. Offline radiative analysis suggested that the $f_c$ bias is likely to contribute the most to the $SW_{\text{TOA}}$ biases, particularly in the statocumulus region in the northern Atlantic, suggesting that biases in $f_c$ are the most important to address. This result also shows that biases in the response of cloud fraction to aerosol are likely to cause large forcing biases. $LWP_{\text{ic}}$ biases were determined to be most important in causing the positive $SW_{\text{TOA}}$ bias north of Scandinavia and so improvements there are also needed. $N_d$ biases were deemed important in causing $SW_{\text{TOA}}$ biases in the northern N. Atlantic, but with only small regions of high impact in the southern N. Atlantic regions. However, it should be considered that a small impact from biases in a variable on the mean $SW_{\text{TOA}}$ flux bias does not preclude a large impact on forcing since the magnitude of the aerosol forcing is a lot smaller (of order 1-2%) than the mean $SW_{\text{TOA}}$ flux and so small biases can still have the potential to produce a significant forcing error.

### 4.2 Aerosol forcing conclusions

The ARI and ACI surface aerosol forcings were calculated using the instantaneous model output. In agreement with B12 the magnitude of the ARI forcing over ocean grid boxes in the NA region generally lower than the ACI forcing. The ACI forcing for the NA region (including land points) is negative with a mean value over the region of -1.7 W m$^{-2}$, showing that aerosol forcing is likely to be important in terms of the energy received at the ocean surface in this region. The spatial pattern agrees well with that from B12, however, the magnitude is somewhat lower. This likely reflects the steps taken to reduce the aerosol forcing in the UKESM model as described in Mulcahy et al. (2018).

The ACI forcing was decomposed into contributions from changes between PI and PD in $N_d$, $LWP_{\text{ic}}$ and $f_c$ ($\Delta N_d$, $\Delta LWP_{\text{ic}}$ and $\Delta f_c$, respectively) using an offline calculation of the net surface SW downwelling radiative fluxes. The results showed that $\Delta N_d$ and $\Delta LWP_{\text{ic}}$ contributed most in the northern part of the NA where overcast stratocumulus clouds are prevalent. $\Delta f_c$ had the highest contribution in the southern subtropical regions where cloud fractions are lower, indicating more broken clouds. We speculate that the higher cloud cover in the northern region allows the $\Delta N_d$ and $\Delta LWP_{\text{ic}}$ effect to be larger since the associated albedo perturbations can operate over a wider area. In the southern region it is likely that the broken cloud scenes are more susceptible to aerosol-induced cloud fraction increases. In contrast, in the northern NA region the overcast clouds cannot increase in cloud fraction much further.

Changes in $LWP_{\text{ic}}$ and $f_c$ can be considered to be cloud macrophysical responses because they are linked to the thermodynamics of liquid water production, whereas changes in $N_d$ can be considered more of a microphysical response since they are mainly caused by aerosol changes, although there is a link between the two. In the northern NA region the macrophysical changes contribute to approximately 54.4 % of the total ACI forcing, whereas in the southern North Atlantic region they contribute 71.5 %.

Our results are important in the context of Malavelle et al. (2017) who showed that, for an earlier version of this model, the macrophysical responses to volcanic sulphate aerosol perturbations are likely to be small in the impact region of the volcano (north of $\sim$50$^\text{o}$ N) . Note that the Malavelle et al. (2017) study quoted responses of the all-sky LWP, which is the product of $LWP_{\text{ic}}$ and $f_c$. Our Figures 15 and 16 show that with an updated model, using a year of data and considering PI to PD aerosol changes, macrophysical changes (dominated by $LWP_{\text{ic}}$ changes) have a similar radiative impact to $N_d$ changes for this





region. It would be interesting to discover whether the implied larger $LWP_{ic}$ response to aerosol in the newer model in the Holuhraun region is now larger than suggested by the observational constraint used in Malavelle et al. (2017). Furthermore,

our results show that macrophysical responses are even more important in other NA regions, especially the southern NA. This result highlights that cloud responses to aerosol perturbations in specific regions are not necessarily representative of those elsewhere.

In order to understand the cause of the aerosol-induced $LWP_{ic}$ and $f_c$ increases in the model (i.e., the macrophysical responses), simulations have been performed where $N_d$ has been prevented from affecting rain formation from cloud liquid

through the autoconversion parameterization. Instead a constant $N_d$ value is used. In these simulations the changes in $LWP_{ic}$ and $f_c$ between PI and PD in the N. Atlantic region drop to nearly zero, or even decrease between PI and PD, strongly indicating that the mechanism for the increases in macrophysical quantities in the standard simulations is the suppression of rain. This is consistent with previous studies which have shown that precipitation is a major factor in causing cloud breakup (Stevens et al., 1998; Berner et al., 2013) mainly due to the stabilization effect of rain evaporation in the lower boundary layer, but also

with some positive feedback due to the removal of aerosol by rain and the subsequent enhancement of rain due to the lower aerosol concentrations (larger droplets). In a global model the representation of the thermodynamics of this process will be reliant on the boundary layer scheme, along with input from the cloud microphysics scheme. Since both processes are highly parameterized in global models it is likely that there is some uncertainty in their representation and thus uncertainty in the response of the clouds to aerosol. Since the contribution to aerosol forcing from changes in macrophysical quantities is very

large in this model, this highlights the need for detailed and targeted model evaluation of these processes.

### 4.3    Important meteorological situations: conclusions

Contributions to the surface forcing were calculated for situations composed of each of the 8 different combinations (termed here as "cloud states") of cloudy or clear for low, mid and high cloud altitudes (see Fig. 17). The largest contributions (taking into account the frequency of occurrence as well as the forcing effect) always involved the presence of low-altitude clouds

in either the PI or PD suggesting that such clouds are driving most of the aerosol forcing in the model. There was some expectation of this apriori since the convection scheme in the model does not respond to aerosol and it is likely that this would be responsible for a lot of the creation of higher altitude cloud. However, it is also possible that such cloud is created by the convection scheme and then gets handed over to the large-scale cloud scheme and could then be affected by aerosol. It may be that low clouds are more affected simply because they are closer to the surface aerosol sources.

In both regions the largest forcing contributions came from situations with the same cloud states in the PI and PD. Of these the largest contributions for the northern NA region (and a considerable contribution for the southern NA region) was when low, mid and high altitude cloud were present at the same time suggesting that in these regions considerable forcing occurs when higher altitude cloud is present. Part of the reason for this is likely due to the prevalence of clouds from a range of altitudes in this region. Thus, it may be important to consider the potential radiative effects of mid and high altitude clouds such as the

shielding of low-altitude cloud, as also indicated in Malavelle et al. (2017). The neglecting of mid and high altitude clouds in the radiative calculations presented in this paper may therefore lead to some inaccuracy. On the other hand, the mid and high





altitude cloud may be sufficiently thin that it has negligible impact. Further work is needed to determine whether this is the case. However, the idea is supported by the fact that the offline calculations of ACI forcing that considered only low-altitude clouds matched the model calculated forcing (that included all cloud types) very closely.

Situations with only low-altitude cloud in the PI and PD were also very important for both the northern NA and southern NA regions, again highlighting the importance of low-altitude cloud for aerosol forcing. Net contributions involving the creation or destruction of a cloud type (low, mid or high cloud) were small for the northern NA region, but higher for the southern NA region. For the latter the creation of low-altitude cloud was implicated in each case. This fits with the result that $\Delta f_c$ was a large contributor to the surface forcing in the southern NA region.

Overall, the results suggest that low-altitude cloud is the largest contributor to aerosol forcing in the model and so improvements to the representation of this and its response to aerosol are likely to yield the biggest improvement in the aerosol forcing estimate in this model. On the other hand, the lack of aerosol awareness of the convection scheme is unrealistic and may lead to missing aerosol-cloud interaction processes in the model. There are some indications that cyclones respond to increased aerosol concentrations by increasing their LWP and hence radiative forcing Mccoy et al. (2018). The degree to which the con-

vection scheme is involved in representing the cloud associated with cyclones and hence the likelihood of such aerosol-cloud interactions being missed at climate model resolution could be explored with high resolution models (e.g., the nested version of the UM) in future work.

### 4.4   Important cloud types: conclusions

The forcing for gridboxes with low-altitude-only clouds and clear skies were further examined in terms of pairings of cloud

fraction from the PI and PD simulations. The aim was to try to understand the PI-to-PD cloud transitions that occur to produce the model aerosol forcing and therefore gain insight into the types of clouds involved. The results showed that gridboxes with overcast (100% cloud cover) in both the PI and the PD accounted for by far the largest contribution to forcing in the northern NA region, although there were some smaller contributions from transitions between 60–80 and 100% cloud cover. This agrees with the result from the forcing contribution calculations that changes in cloud fraction due to aerosol are not particularly

important for forcing in this region and suggests that cloud fraction transitions that do occur are mostly likely within the stratocumulus regime. The brightening of stratocumulus clouds by the Twomey effect seems to dominate the forcing in this region.

    For the southern NA region the largest contribution is from transitions from 0 to $<\sim$30% cloud cover indicating the creation of trade cumulus from clear skies. The creation of 80–100% cloud fraction states from 0–20% states also has an important

forcing contribution, suggesting the formation of stratocumulus clouds from clear skies and cumulus to stratocumulus transitions due to the anthropogenic aerosol input. The brightening of overcast clouds is also still very important here, but not as important as in the northern NA.

    A caveat is that with this analysis is that we only pair the same times in the PI and PD and don't allow for evolution over time. Therefore, the pairings may not be accurate since the Lagrangian trajectories associated with a given pairing may have

had different cloud fractions in the time before the snapshot. It may therefore be useful to examine the evolution of cloud





fraction, etc. over time using Lagrangian trajectories and examine how this varies with initial aerosol/droplet concentrations (e.g., see Eastman and Wood, 2016; Eastman et al., 2016; Eastman and Wood, 2018). This could be done in the PD simulation and compared to observations, but also comparisons between the evolution in the PI and PD runs could be made.

### 4.5 Recommendations

The results from this paper suggest that future studies should target the improvement of different cloud responses to aerosol depending on the location within the North Atlantic. In the northern NA the Twomey effect was a large contributor to the aerosol forcing. This suggests that constraint of the aerosol forcing using observations will be easier here than in the southern region where cloud adjustments/macrophyiscal effects were more dominant. Similar approaches to those performed in Malavelle et al. (2017) whereby the model $N_d$ and $r_e$ responses to known aerosol perturbations (the Holuhrahn volcanic eruption in the case of

Malavelle et al., 2017) were evaluated using satellite data may therefore be able to constrain aerosol forcing. Examination of modelled trends in $N_d$ and $r_e$ over time in response to known aerosol emission changes may also prove useful in this regard. The examination of $N_d$ and $r_e$ changes alone is advantageous since their natural variability is significantly lower than that of $LWP_{ic}$, $f_c$ and SW fluxes (Malavelle et al., 2017) making it easier to quantify aerosol induced signals from the observations. However, our model results also showed a fairly large influence on aerosol forcing from increases in $LWP_{ic}$ (i.e., cloud thickening) in

the northern NA region, which may complicate such efforts.

  In the southern NA, the accuracy of the cloud fraction response in the model should be evaluated, with a particular focus on low clouds and the creation of trade cumulus and stratocumulus. As mentioned earlier, this is likely to be more difficult than evaluating the Twomey effect due to the larger number of processes involved. Ideally, the individual processes would be targeted for model evaluation with the formation of rain via the autoconversion process a prime first target since this was

shown to be the cause of the $f_c$ and $LWP_{ic}$ response to aerosol in our model. Ways forward with this might include the use of observational data from the ground-based ARM site on the Azores and data from aircraft observations that have also taken place there. This is ideally located since it is near a location where the older version of the model predicts a large contribution to the aerosol forcing from cloud fraction changes, but the newer version less so. Thus, the observations might help to determine which is the most realistic. Combined with satellite data, the long time series observations of aerosols, $N_d$, $LWP_{ic}$, $f_c$ and rain

rates can be used to evaluate the relationships between these variables. Separating a causative signal due to the aerosol alone from that due to meteorology, etc. is difficult, but it may be possible to use the techniques described in McCoy et al. (2019). In that paper several meteorological drivers are controlled for via binning and multi linear regression, along with using the model to estimate and subtract the influence of non-aerosol-induced changes. However, it is possible that even an evaluation of the observed relationship between rain rates and other variables, without separating the causative effects of aerosol, will prove

useful in constraining the cloud adjustments.

  A complementary approach is to model this region using a high resolution nested version of the model, which would also allow for the use of the more sophisticated CASIM (Cloud AeroSol Interaction Microphysics) microphysics scheme (e.g., see Grosvenor et al., 2017; Stevens et al., 2018; Miltenberger et al., 2018a, b; Gordon et al., 2018). The assumption would be





that the cloud responses to aerosol of this would be more accurate than the global model resolution simulations, although its
performance should be tested using the observations.

Finally, work to evaluate and improve the accuracy of the satellite evaluation in this paper would be very useful. For example, to deal with the issues of evaluating the model LWP in situations with precipitation, a microwave radiometer simulator could be implemented into the model, which would be able to estimate the total attenuation of the 37 GHz channel (used by AMSR-E to retrieve LWP) taking into account the differing attenuation strengths of cloud water and rain water (Lebsock and Su, 2014).
This attenuation could then be directly compared to that from AMSR-E. Comparing the global model evaluation results with and without the convective contribution to those from a high resolution model (where the convective TLWP will be explicitly resolved) might help to assess the role of the convective parameterization on the evaluation of model TLWP. This could be combined with the microwave radiometer simulator mentioned above to help overcome the issue of retrieval problems in raining conditions too.

Much greater confidence in $N_d$ retrievals would gained through further validation using in-situ aircraft data. Cloud fraction could be evaluated with additional cloud fraction satellite datasets and ground based data to improve the confidence in the result. More sophisticated analyses such as 2-D histograms of cloud fraction and $LWP_{ic}$ would assess this important cloud fraction variable at different cloud thicknesses and may help to isolate issues in particular regimes. Model improvements to the sub-grid cloud scheme might be considered, such as a link between the boundary layer turbulence and the width of the sub-grid
relative humidity distribution. Comparisons to high resolution models may also help with this.

*Data availability.* Raw model data is kept on tape archive available through the JASMIN (http://www.jasmin.ac.uk/) service. Please see http://www.ceda.ac.uk/blog/access-to-the-met-office-mass-archive-on-jasmin-goes-live/ for details on how to arrange access to Met Office data via JASMIN.

## Appendix A:  Details on the satellite data sets

**A1   Droplet concentration**

$N_d$ can be estimated using satellite retrievals of $\tau_c$ and $r_e$ (Han et al., 1998; Brenguier et al., 2000; Boers et al., 2006; Bennartz, 2007a; Grosvenor et al., 2018b) made using observations from the visible and shortwave infrared wavelengths from instruments like MODIS (Nakajima and King, 1990). Here we use a dataset based on the methods from Grosvenor et al. (2018a), which represented some modifications to the methods described in Grosvenor and Wood (2014). The methodology used here differs
slightly from that used in Grosvenor et al. (2018a) in that data are not filtered for the presence of $\tau_c<5$ data points and the correction for the vertical penetration depth bias proposed in Grosvenor et al. (2018a) is not applied. The 3.7 $\mu$m $r_e$ is used for the $N_d$ calculations, which has been suggested to be less prone to errors due to cloud heteorogeneity (Zhang et al., 2012, 2016b; Grosvenor et al., 2018b). The data set excludes $1\times1^o$ data points with mean SZA greater than 65$^o$, mean cloud top heights greater than 3.2 km, liquid cloud fractions less than 80% and for which the maximum sea-ice areal coverage over a



765 moving 2-week window exceeded 0.001 %. The sea-ice data used were the daily $1 \times 1^{\circ}$ version of the "Sea Ice Concentrations from Nimbus-7 SMMR and DMSP SSM/I-SSMIS Passive Microwave Data, Version 1" data set (Cavalieri et al., 1996). The $N_{\mathrm{d}}$ dataset (without sea-ice screening) is available; see Grosvenor and Wood (2018).

**Appendix B: Bias metrics**

Following Gustafson and Yu (2012) , the NMBF is defined as

$$
\begin{cases}
100 \times \left( \left| \dfrac{\overline{M}}{\overline{O}} \right| - 1 \right), \text{ if } |\overline{M}| \geq |\overline{O}| \text{ and } \dfrac{\overline{M}}{|\overline{M}|} = \dfrac{\overline{O}}{|\overline{O}|} \\[2ex]
100 \times \left( 1 - \left| \dfrac{\overline{O}}{\overline{M}} \right| \right), \text{ if } |\overline{M}| < |\overline{O}| \text{ and } \dfrac{\overline{M}}{|\overline{M}|} = \dfrac{\overline{O}}{|\overline{O}|} \\[2ex]
\text{Undefined, if } \dfrac{\overline{M}}{|\overline{M}|} \neq \dfrac{\overline{O}}{|\overline{O}|} \text{ , i.e., the signs of } \overline{M} \text{ and } \overline{O} \text{ differ}
\end{cases}
\tag{B1}
$$

, where $\overline{M}$ is the mean of the model values and $\overline{O}$ is the mean of the observed values. In this paper this refers to the spatial mean time-averaged values. The NMAEF is defined as

$$
\begin{cases}
100 \times \left( \dfrac{\sum |M_i - O_i|}{|\sum O_i|} \right), \text{ if } |\overline{M}| \geq |\overline{O}| \text{ and } \dfrac{\overline{M}}{|\overline{M}|} = \dfrac{\overline{O}}{|\overline{O}|} \\[2ex]
100 \times \left( \dfrac{\sum |M_i - O_i|}{|\sum M_i|} \right), \text{ if } |\overline{M}| < |\overline{O}| \text{ and } \dfrac{\overline{M}}{|\overline{M}|} = \dfrac{\overline{O}}{|\overline{O}|} \\[2ex]
\text{Undefined, if } \dfrac{\overline{M}}{|\overline{M}|} \neq \dfrac{\overline{O}}{|\overline{O}|}
\end{cases}
\tag{B2}
$$

, where $M_i$ are the time-averaged model values and $O_i$ the time-averaged observed values for spatial location $i$.

775 **Appendix C: Evaluation of grid box mean LWP**

Here we show plots similar to Figures 5 and 6, but for the all-sky (cloudy and clear contributions) LWP; i.e., without applying the step of dividing by the cloud fraction. Table C1 summarizes the results for this and the other plots in this section. The bias pattern is very similar to that of the in-cloud values suggesting that the conversion between LWP and $LWP_{\mathrm{ic}}$ (by dividing by the low-altitude COSP-CALIPSO cloud fraction) does not greatly affect the evaluation, probably because the low cloud

780 fraction biases are generally very small (Fig. 1). However, for the regions of negative bias between the equator and 18ºN the biases are larger for LWP than for $LWP_{\mathrm{ic}}$ because there is a negative low cloud fraction bias here, which acts to increase the model $LWP_{\mathrm{ic}}$ values relative to the observed ones to give a lower $LWP_{\mathrm{ic}}$ bias. When filtering to include only datapoints with $f_{\mathrm{LWP}} > 0.99$ (Fig. C2) the bias pattern is again similar to that for $LWP_{\mathrm{ic}}$.

As discussed earlier, AMSR-E observations are potentially biased when it is raining. This is because assumptions are made

785 about the partitioning between LWP and RWP in order to facilitate the retrieval since rainwater attenuates the microwave signal more strongly than liquid droplets (Lebsock and Su, 2014). It is assumed that rain water does not occur for water



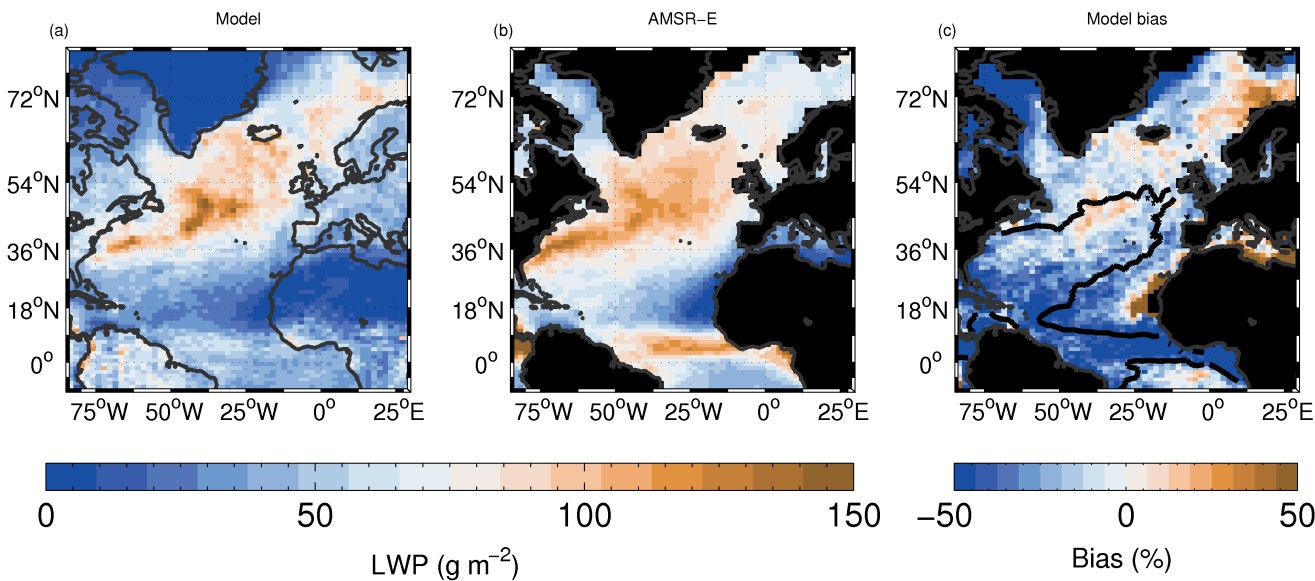

**Figure C1.** Time-mean all-sky (i.e., including cloudy and clear regions) LWP model evaluation for both day and night overpasses. c) includes a contour of the 0.9 value of $f_{\mathrm{LWP}}$; see Fig. 4 for the full map of $f_{\mathrm{LWP}}$ for reference.

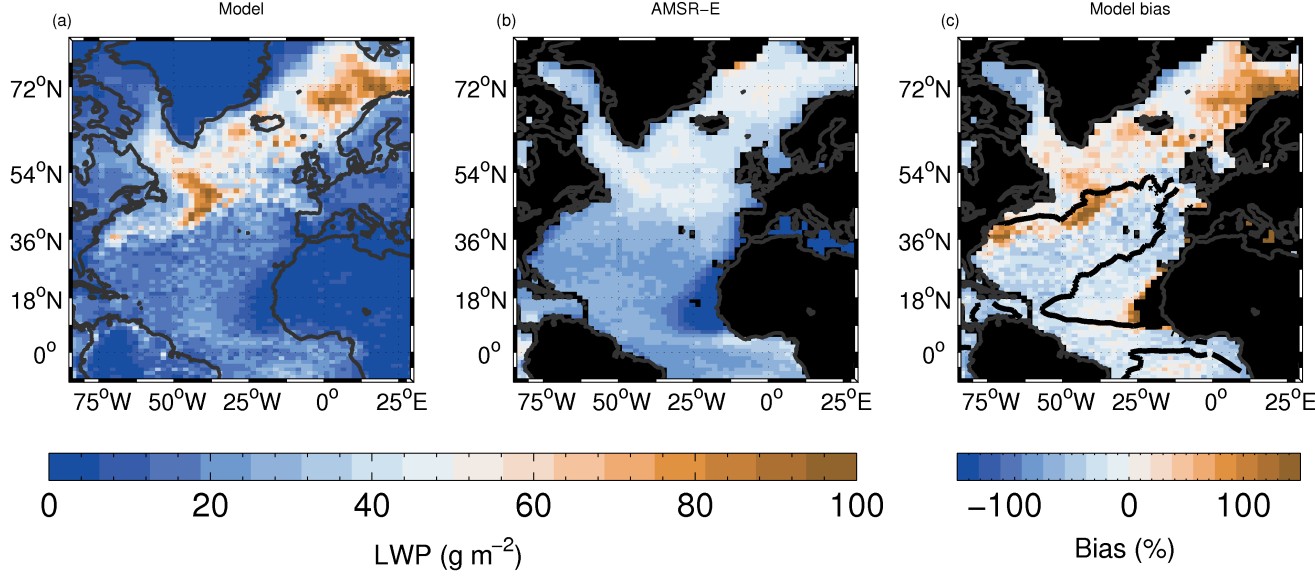

**Figure C2.** As for Fig. C1 except both the model and satellite data has been filtered before time averaging to only include datapoints for which $f_{\mathrm{LWP}}$ is greater than 0.99. This quantity is denoted as $LWP_{0.99}$.


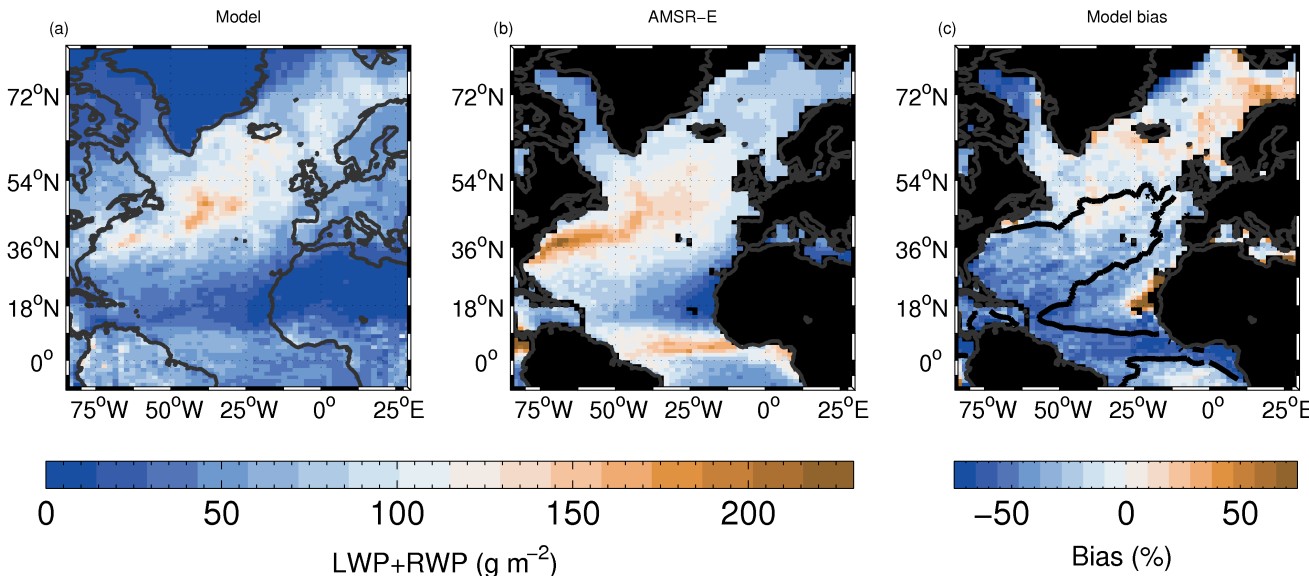

**Figure C3.** As for Fig. C1 except with the addition of the all-sky rain water path (RWP) for both the model and AMSR-E satellite data.No filtering for $f_{LWP}$ has been applied.

paths below 180 g m$^{-2}$, which may lead to inaccuracies since the true partitioning value is likely to vary. Because of these issues, some previous model evaluation studies (e.g., Furtado et al., 2016) have chosen to compare the total liquid water path (TLWP=LWP+RWP) to the TLWP provided in some microwave based products. We also do this in Fig. C3; for this plot we only use the large-scale RWP and not the convective RWP (or LWP). However, it should be borne in mind that if the assumed partitioning threshold is incorrect the TLWP value retrieved by the satellite will also be incorrect and so it is unclear whether this leads to a more accurate model-to-satellite evaluation.

The results show that the addition of RWP (compare Fig. C1 to Fig. C3) enhances both the model and the observations, so that the pattern of bias remains similar. This suggests that it does not matter greatly whether LWP or TLWP are used for model evaluation. The magnitudes of the negative model biases for TLWP are slightly enhanced and those of the positive biases reduced relative to those for LWP, though. The NMBFs for the northern NA region are -16.4% for TLWP and -8.3% for LWP. For the southern NA region the corresponding values are -65.7% and -39.5%.

Fig. C4 shows the ratio of the TLWP from the model's convection parameterization to that of the large-scale plus convective TLWP. In most of the southern part of the NA region the convective TLWP accounts for the majority of the TLWP. However, it is not straightforward to decide whether the convective TLWP from the model is physically meaningful and whether it should be included when comparing to the microwave instruments. On the one hand, LWP and RWP from convection in the real world will be detected by the instruments, but on the other hand it is unclear whether the condensed water from the convection parameterization in models properly represents this. Liquid water content and vapour are detrained to the





environment from the convection scheme and incorporated into the large-scale cloud scheme (see UM Documentation Paper
030; hereafter UMDP030; https://code.metoffice.gov.uk/doc/um/vn11.3/umdp.html#030). It may therefore be the case that
the large scale cloud amount is the more appropriate quantity even if there is condensate associated with the convection
scheme and that double counting would occur if also using the convective values (UMDP30). However, there is also some
convective condensate from the convective core that is not transferred to the large scale scheme. Another point of note is that
LWP associated with the convection scheme is only used by the radiation scheme for shallow convective clouds (clouds with
geometrical depths less than 500 m over land and 1500 m over ocean) meaning that the majority of liquid water in deep clouds
has no effect on radiation. Given the uncertainty, here we examine the effect of adding the convective LWP on the model
evaluation (Fig. C5). As might be expected from Fig. C4 its inclusion leads to much larger model values, particularly in the
southern parts of the region. The bias pattern also changes, so that large positive biases occur almost everywhere, compared to
negative biases in the south and positive biases in the north when using only large-scale TLWP (Fig. C3). Given this, it would
be useful for future studies to determine whether any of the convective TLWP should be included; ideas for how to make
progress on this are discussed in Section 4.5.

The effect of the further addition of convective RWP is shown in Fig. C6. This increases the model values somewhat in the
convective regions, but not by a large relative amount. As might be expected, the model biases are also therefore increased, but
not by the same extent as the addition of the convective LWP; the spatial pattern of model bias remains very similar to that in
Fig. C5.

We also test the effect of filtering using $f_{LWP} > 0.99$ for the model and satellite data when including the large-scale RWP
and the convective LWP and RWP (Fig. C7). The filtering reduces the large positive biases in the southern part of the domain
(which were mainly due to the addition of the convective LWP) considerably, presumably because most of the gridpoints with
convection are removed. The spatial pattern of the biases here are similar to those from Fig. C2 where only the LWP from the
large-scale cloud scheme was used, but the biases in the southern part of the domain have changed from being slightly negative
to slightly positive, while the biases in the northern part have become a little more positive. These relatively small changes
suggest that whether the convective LWP, which was the biggest contributor to the total water path, is used is not as critical
when filtering for non-precipitating situations. However, there are still large overall differences in the NMBF values between
Figs. C2 and C7 (see Table C1) such that whether RWP or convective LWP/RWP is used or not still has significant effects on
model evaluation attempts and therefore should be an avenue of future investigation.

## Appendix D: Removing aerosol impact on rain autoconversion

Figures D1 and D2 show the percentage increases between PI and PD for $LWP_{ic}$ and $f_c$ ($\Delta LWP_{ic}$ and $\Delta f_c$), respectively,
when aerosols are prevented from affecting the rain autoconversion process. This is done by setting $N_d$ in the autoconversion
process equation to a constant value of 300 cm$^{-3}$ over land and 100 cm$^{-3}$ over oceans. These are here termed the Constant-
NdAutoCon runs. Most oceanic regions showed positive $\Delta LWP_{ic}$ and $\Delta f_c$ in the full model and near-zero or negative changes
for ConstantNdAutoCon. Table 3 shows that between the standard and ConstantNdAutoCon runs $\Delta LWP_{ic}$ reduced from 3.2



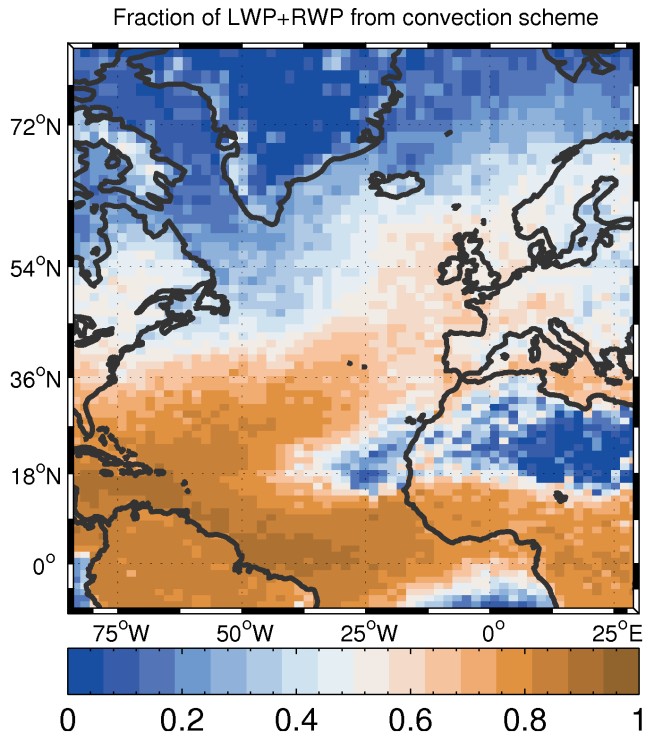

**Figure C4.** Ratio of the TLWP (where $TLWP = LWP + RWP$) from the model convection scheme to the total (from the convection + large-scale scheme) TLWP. The RWP from the convection scheme is calculated from the convective rain rate diagnostic by assuming a raindrop size distribution and fallspeed relationship (see Furtado et al., 2016, for details).

to -5.0 % for the NA region and from 2.7 to -0.43 % for the northern NA . For the southern NA region there was a small negative $\Delta LWP_{ic}$ in the standard runs (-0.59 %) and a very similar value for ConstantNdAutoCon (-0.50 %) consistent with the idea that aerosols have little impact on $LWP_{ic}$ in this region as discussed earlier. Respective $\Delta f_c$ values for the standard
and ConstantNdAutoCon runs were 1.5 and 0.14 % for the NA region; 0.94 and 0.02 % for the northern NA; and 2.7 and -0.75 % for the southern NA. These results suggest that most of the PI to PD increases in the macrophysical cloud properties ($LWP_{ic}$ and $f_c$) were due to the impact of aerosols on the rain autoconversion process, likely via the precipitation suppression effect of enhanced aerosol.

However, in the region of the Atlantic near the equator Fig. D2 suggests that the increase in $f_c$ between the PI and PD
runs was not due to the impact of aerosol on the autoconversion process since there are still positive $\Delta f_c$ values for the ConstantNdAutoCon runs. We hypothesize that, as far as allowed by the nudging (wind-only nudging applied above the boundary layer), the $f_c$ increases in this region may be related to other aerosol impacts such as the ARI, ACI or semi-direct effects, via thermodynamical or dynamical changes.





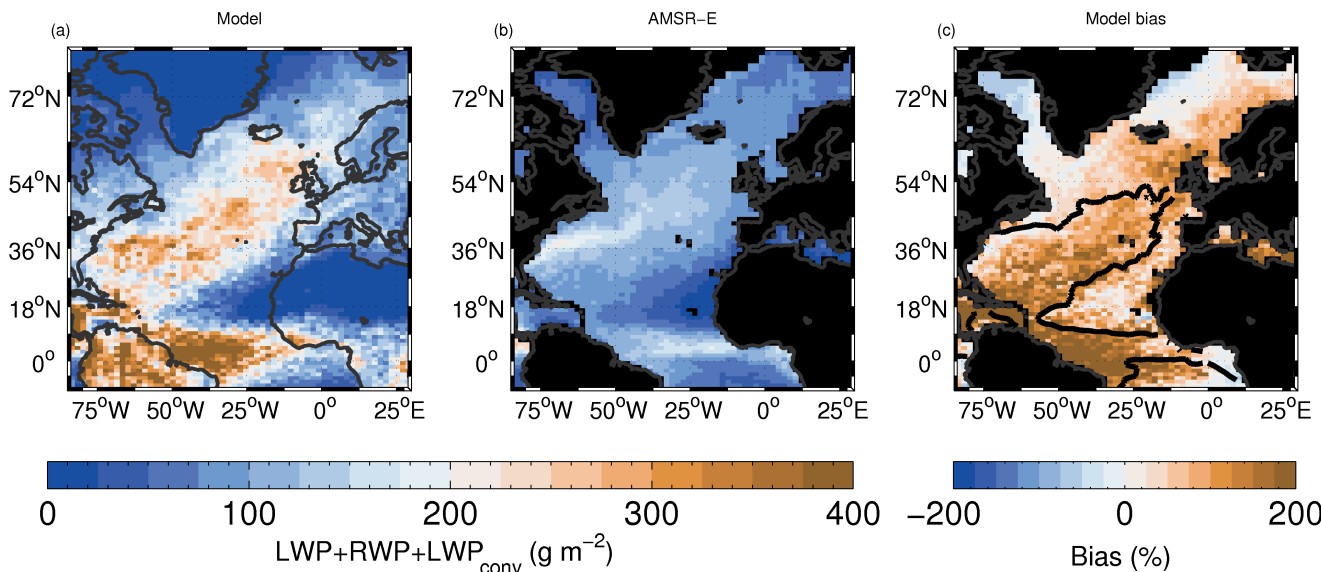

**Figure C5.** As for Fig. C3 except with the addition of the all-sky LWP from the convection parameterization for the model.

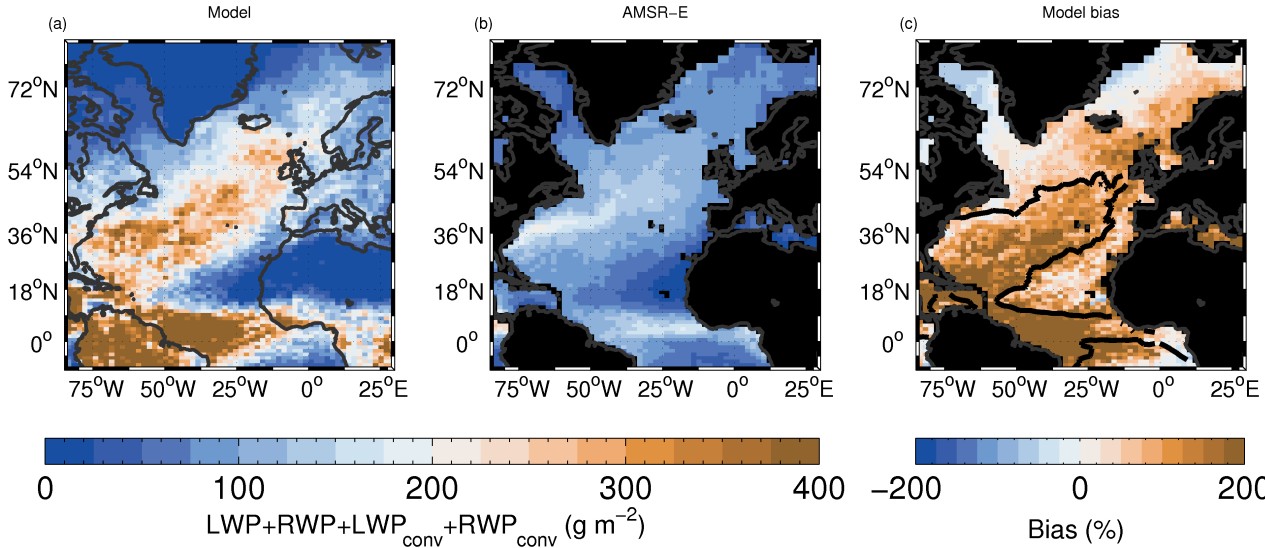

**Figure C6.** As for Fig. C5 except with the addition of the all-sky RWP from the convection parameterization of the model.





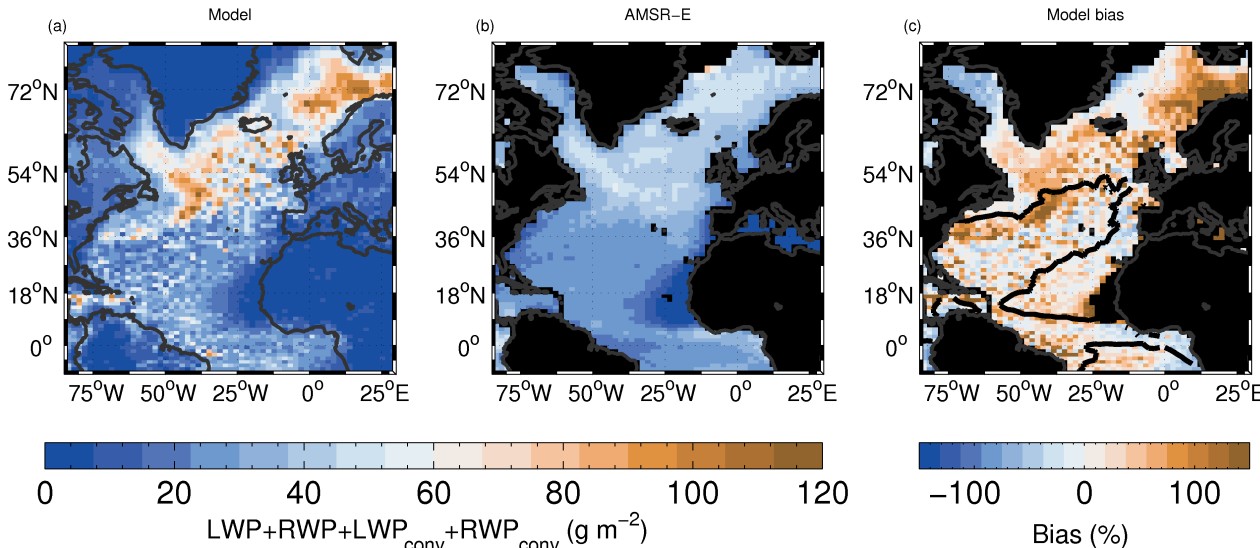

**Figure C7.** As for Fig. C6 except that the model and AMSR-E data have been filtered before time averaging to only include datapoints for which $f_{LWP}$ is greater than 0.99 (as in Fig. C2).

### Appendix E: Offline shortwave flux calculation and partitioning

Here we give details of the offline calculations used to estimate the SW fluxes, which are needed to estimate the contributions to the forcing from the individual changes in cloud properties between the PI and PD simulations.

An estimate of the cloud optical depth ($\tau_c$) can be made following Eqns. 1 and 5 of Grosvenor et al. (2018b) :-

$$\tau_c = \int_{z_{base}}^{z_{top}} \frac{3\,Q_{ext}}{4\,\rho_w} \frac{L(z)}{r_e(z)} dz \qquad (E1)$$

, where $Q_{ext}$ is the scattering efficiency, which we assume to have a constant value of 2; this has been shown to be the case
for droplet radii that are much larger than the wavelength of light concerned (Bennartz, 2007b). $\rho_w$ is the density of liquid water, $L(z)$ is the liquid water content, $r_e(z)$ is the effective radius, $z$ is height, $z_{base}$ is cloud base height and $z_{top}$ is cloud top height.

We assume that the clouds are adiabatic (or some constant fraction of adiabatic) so that their liquid water increases linearly with height, and it is assumed that $N_d$ is constant throughout their depth. Observations suggest that both are valid assumptions



**Table C1.** Model evaluation statistics for the various sub-regions using time-averaged data. See Gustafson and Yu (2012) for details of the normalized mean bias factor (NMBF) and the normalized mean absolute (NMAEF) error factor. r is the spatial correlation coefficient between the model and observed time-averages. All values are area-weighted to account for the variation in area of the model grid-boxes.

| # | Region name | r | Model mean | Obs. mean | NMBF (%) | NMAEF (%) |
|---|---|---|---|---|---|---|
| | | All-sky LWP ($gm^{-2}$) | | | | |
| 1 | N Atlantic | 0.71 | 51.9 | 70.5 | -35.9 | 40.0 |
| 2 | NN Atlantic | 0.89 | 88.2 | 95.5 | -8.3 | 12.9 |
| 3 | SN Atlantic | 0.86 | 43.3 | 60.4 | -39.5 | 40.4 |
| | | All-sky $LWP_{0.99}$ (LWP for $f_{LWP} > 0.99$; $gm^{-2}$) | | | | |
| 1 | N Atlantic | 0.66 | 24.3 | 27.2 | -11.9 | 40.0 |
| 2 | NN Atlantic | 0.56 | 37.5 | 36.1 | 3.9 | 41.1 |
| 3 | SN Atlantic | 0.47 | 17.2 | 23.9 | -39.1 | 42.5 |
| | | All-sky LWP+RWP ($gm^{-2}$) | | | | |
| 1 | N Atlantic | 0.68 | 60.4 | 91.5 | -51.4 | 54.9 |
| 2 | NN Atlantic | 0.80 | 104.5 | 121.6 | -16.4 | 18.6 |
| 3 | SN Atlantic | 0.87 | 49.0 | 81.2 | -65.7 | 66.4 |
| | | All-sky LWP+RWP+$LWP_{conv}$ ($gm^{-2}$) | | | | |
| 1 | N Atlantic | 0.64 | 185.7 | 91.5 | 103.0 | 105.8 |
| 2 | NN Atlantic | 0.52 | 217.2 | 121.6 | 78.6 | 78.6 |
| 3 | SN Atlantic | 0.84 | 178.3 | 81.2 | 119.6 | 119.8 |
| | | All-sky LWP+RWP+$LWP_{conv}$ + $RWP_{conv}$ ($gm^{-2}$) | | | | |
| 1 | N Atlantic | 0.63 | 211.4 | 91.5 | 131.1 | 133.2 |
| 2 | NN Atlantic | 0.53 | 235.4 | 121.6 | 93.6 | 93.6 |
| 3 | SN Atlantic | 0.85 | 202.1 | 81.2 | 148.9 | 149.0 |
| | | All-sky LWP+RWP+$LWP_{conv}$ + $RWP_{conv}$ for $f_{LWP} > 0.99$ ($gm^{-2}$) | | | | |
| 1 | N Atlantic | 0.61 | 33.9 | 27.2 | 24.6 | 44.3 |
| 2 | NN Atlantic | 0.63 | 48.0 | 36.1 | 32.8 | 48.6 |
| 3 | SN Atlantic | 0.43 | 27.9 | 23.9 | 16.7 | 31.1 |

for stratocumulus clouds (Albrecht et al., 1990; Zuidema et al., 2005; Painemal and Zuidema, 2011; Miles et al., 2000; Wood, 2005). The adiabatic assumption means that :-

$$L(z) = f_{ad} c_w z \tag{E2}$$

where $f_{ad}$ is the adiabatic fraction, which is assumed constant with height and with a value of 0.8. $c_w(T,P)$ is the rate of increase of liquid water content with height ($dL/dz$, with units $\mathrm{kg\,m^{-4}}$) and is referred to as the "condensation rate" in

Bennartz (2007b), or the "water content lapse rate" in Painemal and Zuidema (2011). See Ahmad et al. (2013) for a definition. A constant temperature of 278 K is used for the temperature ($T$) in the calculation of $c_w$, along with a constant pressure ($P$) of 850 hPa. This is an approximation since these values would vary depending upon cloud height and location. However, since we only consider low clouds here and because in Eqn. E1 the dependence of $\tau_c$ upon $c_w$ is very weak ($\tau \propto 1/c_w^{1/6}$), this introduces very little error.





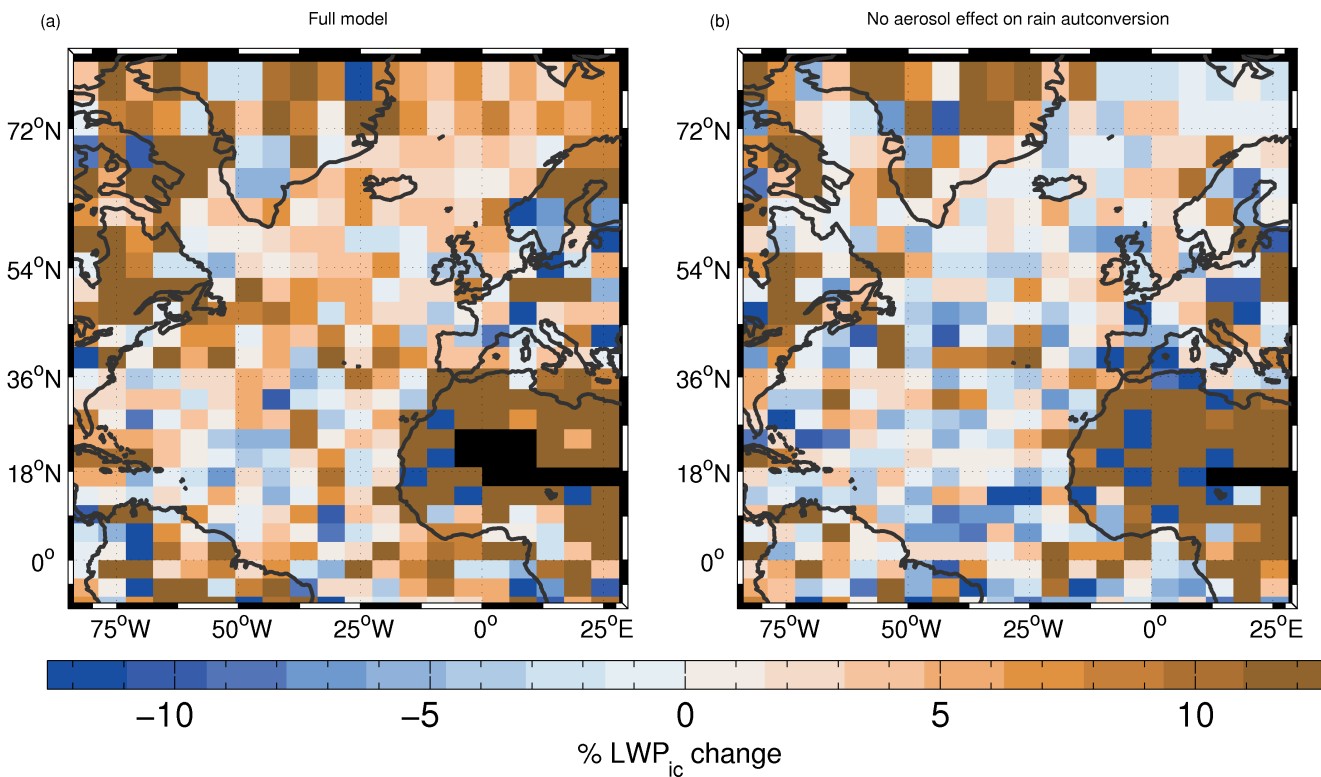

**Figure D1.** Mean percentage increase in $LWP_{ic}$ between PI and PD runs for left: the full run; right: the run where aerosol has been prevented from affecting the rain autoconversion.

In the radiative scheme of the UKESM model the parameterisation of Liu et al. (2008) is used, which makes the width of the droplet size distribution (assumed to be represented by a gamma function) a function of $N_d$ and $L$. For consistency with the model we also apply this parameterisation to our offline radiative calculations. From Mulcahy et al. (2018) :-

$$r_e(z) = \beta_m(z) \left( \frac{3L(z)}{4\pi\rho_w N_d} \right)^{\frac{1}{3}}, \tag{E3}$$

where $\beta_m$ is a parameter related to the droplet distribution width, which is parameterized in Liu et al. (2008) as :-

$$\beta_m(z) = x \left( \frac{N_d}{L(z)} \right)^y, \tag{E4}$$

$x$ and $y$ are constants set at 0.0266 (with units $kg^y$, which has been converted from the value of 0.07 $g^y$ from Liu et al. (2008)) and 0.14, respectively, following Liu et al. (2008) and Mulcahy et al. (2018). N.B., $\beta_m$ is related to the k parameter, which is often used to describe the droplet distribution width (e.g., Martin et al., 1994), as $\beta_m = 1/k^{(1/3)}$.





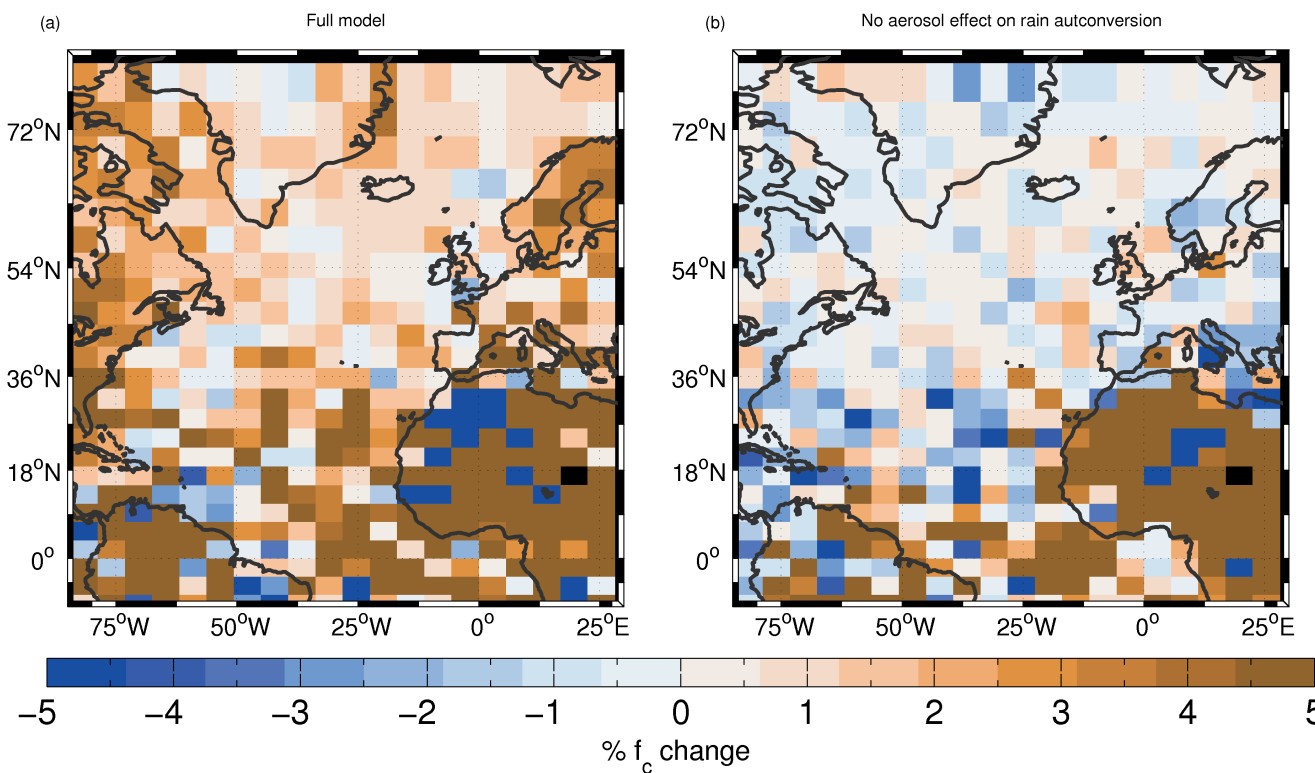

**Figure D2.** As for Fig. D1 except for $f_c$ changes.

Using Eqn. E3 to substitute for $r_e$ in Eqn. E1 and including Eqn. E2 and Eqn. E4 gives :-

$$\tau_c = B \int_{z_{\text{base}}}^{z_{\text{top}}} z^{(y+\frac{2}{3})} \mathrm{d}z \tag{E5}$$

, where

$$B = \frac{3 Q_{\text{ext}} (f_{ad} c_w)^{(y+\frac{2}{3})}}{4 \rho_{\text{w}} x \left(\frac{3}{4\pi\rho_{\text{w}}}\right)^{1/3} N_d^{(y-\frac{1}{3})}} \tag{E6}$$

Integrating over height gives :-

$$\tau_c = \frac{B H^{(y+\frac{5}{3})}}{(y+\frac{5}{3})} \tag{E7}$$



where $H$ is the depth of the cloud ($z_{top} - z_{base}$). $H$ can be determined from the in-cloud LWP under the adiabatic assumption :-

$$H = \left( \frac{2LWP_{ic}}{f_{ad}c_w} \right)^{\frac{1}{2}} \tag{E8}$$

, which follows from integrating Eqn. E2 over the depth of the cloud. This now allows $\tau_c$ to be calculated from the cloud $LWP_{ic}$ and $N_d$.

The cloud albedo ($A_c$) is then estimated using Eqn. 24.38 of Seinfeld and Pandis (2006), which is based on the two-stream approximation for a non-absorbing, horizontally homogeneous cloud :-

$$A_c = \frac{\tau_c}{\tau_c + 7.7} \tag{E9}$$

The shortwave downwards flux at the surface ($SW_{downSURF}$) for a given cloud fraction ($f_c$) can then be estimated from the cloudy and clear-sky fluxes ($SW_{downSURFcloudy}$ and $SW_{downSURFclear}$, respectively) using :-

$$SW_{downSURF} = f_c SW_{downSURFcloudy} + (1 - f_c) SW_{downSURFclear} \tag{E10}$$

$SW_{downSURFclear}$ is estimated from the incoming TOA (top of atmosphere) SW flux ($SW_{downTOA}$) using :-

$$SW_{downSURFclear} = T_{atmos} SW_{downTOA} \tag{E11}$$

where we have assumed a constant clear-sky transmissivity ($T_{atmos}$). The cloudy sky surface flux is calculated by assuming that the flux reaching the cloud top is equal to $SW_{downSURFclear}$ :-

$$SW_{downSURFcloudy} = T_{atmos} SW_{downTOA} (1 - A_c) \tag{E12}$$

Thus we now have a function $SW_{downSURF}(N_d, f_c, LWP_{ic})$ that estimates the surface downwelling SW flux as a function of the three cloud variables of interest. This allows us to estimate the surface $ERF_{ACI}$ following Eqns. 2 and 1 as :-

$$ERF_{ACI,all} = ERF_{PD} - ERF_{PI}$$

$$= SW_{PDclean+cloudy} - SW_{PDclean+clear} - SW_{PIclean+cloudy} + SW_{PIclean+clear}$$

$$= SW_{PDclean+cloudy} - SW_{PIclean+cloudy}$$

$$= SW_{downSURF}(N_{dPD}, f_{cPD}, LWP_{icPD}) - SW_{downSURF}(N_{dPI}, f_{cPI}, LWP_{icPI}) \tag{E13}$$





, where we have assumed that the PI and PD $SW_{clean+clear}$ values are the same.

The forcing contributions from the changes to the individual cloud parameters are then estimated using :-

$$ERF_{Nd} = 0.5\,(ERF_{Nd,PIbase} + ERF_{Nd,PDbase})$$
$$ERF_{fc} = 0.5\,(ERF_{fc,PIbase} + ERF_{fc,PDbase})$$
$$ERF_{LWPic} = 0.5\,(ERF_{LWPic,PIbase} + ERF_{LWPic,PDbase})$$

(E14)

, where :-

$$ERF_{Nd,PIbase} = SW_{downSURF}(N_{dPD}, f_{cPI}, LWP_{icPI}) - SW_{downSURF}(N_{dPI}, f_{cPI}, LWP_{icPI})$$
$$ERF_{fc,PIbase} = SW_{downSURF}(N_{dPI}, f_{cPD}, LWP_{icPI}) - SW_{downSURF}(N_{dPI}, f_{cPI}, LWP_{icPI})$$
$$ERF_{LWPic,PIbase} = SW_{downSURF}(N_{dPI}, f_{cPI}, LWP_{icPD}) - SW_{downSURF}(N_{dPI}, f_{cPI}, LWP_{icPI})$$

(E15)

Here, all of the PI cloud property values have been used as a baseline value, but with the PI value for one of either $N_d$, $f_c$ or

$LWP_{ic}$ replaced with the PD value. A similar calculation is done using the PD values as baselines and replacing with one of the PI values :-

$$ERF_{Nd,PDbase} = SW_{downSURF}(N_{dPD}, f_{cPD}, LWP_{icPD}) - SW_{downSURF}(N_{dPI}, f_{cPD}, LWP_{icPD})$$
$$ERF_{fc,PDbase} = SW_{downSURF}(N_{dPD}, f_{cPD}, LWP_{icPD}) - SW_{downSURF}(N_{dPD}, f_{cPI}, LWP_{icPD})$$
$$ERF_{LWPic,PDbase} = SW_{downSURF}(N_{dPD}, f_{cPD}, LWP_{icPD}) - SW_{downSURF}(N_{dPD}, f_{cPD}, LWP_{icPI})$$

(E16)

This follows the work of Mülmenstädt et al. (2019) who found that an average of values obtained from using both the PI and PD values as baselines was more accurate than only using say the PI as a baseline. If $f_c$ is zero in the baseline state (i.e., PI for

Eqn. E15, PD for Eqn. E16) then $N_d$ and $LWP_{ic}$ are undefined. Therefore, in order to calculate the effect of an increased $f_c$ between the baseline and perturbed state (i.e., PD for Eqn. E15, PI for Eqn. E16), the $N_d$ and $LWP_{ic}$ values from the perturbed state are used in such cases.

## Appendix F:  Net contributions to forcing from cloud state and cloud fraction state transitions

Sections 3.2.2 and 3.2.3 showed the contributions to the surface forcing from different combinations of PI and PD cloud states

(low, mid, high altitude cloud combinations) and cloud fraction. We explained that it is useful to consider the net forcing contribution from both the PI to PD transitions and those from the reciprocal transitions. I.e., if $g_i$ and $g_j$ are two different





cloud states (where the possible states run from 1 to 8, see Fig. 17) the net contribution ($ERFcont_{NET}$) for that pairing of cloud states is given by :-

$$ERFcont_{NET}(g_i, g_j) = ERFcont_{g_iPI, g_jPD} + ERFcont_{g_jPI, g_iPD}$$

$$g_j > g_i \tag{F1}$$

925       , where $ERFcont_{g_iPI, g_jPD}$ is the forcing contribution from pairings of $g_i$ in the PI simulation and $g_j$ in the PD simulation, and $ERFcont_{g_jPI, g_iPD}$ is the contribution from pairings of $g_j$ in the PI simulation and $g_i$ in the PD simulation.

A similar equation can be formulated using the 5 different cloud fraction bins instead of cloud states.

*Author contributions.* D. P. Grosvenor designed and ran the simulations; analysed the output; processed the satellite data; and produced the text and figures. K. S. Carslaw helped to analyse the model output and provided feedback and edits to manuscript drafts.

*Competing interests.* Author K. S. Carslaw is an executive editor of ACP.

*Acknowledgements.* DPG was funded from the National Environment Research Council (NERC) funded "North Atlantic Climate System: integrated Study (ACSIS)" project via NCAS. We acknowledge use of the MONSooN system, a collaborative facility supplied under the Joint Weather and Climate Research Programme, a strategic partnership between the Met Office and the Natural Environment Research Council. Version 3 of the CALIPSO-GOCCP (GCM Oriented Cloud Calipso Product; Chepfer et al., 2010) was obtained from https://
climserv.ipsl.polytechnique275.fr/cfmip-obs/index.html.satelliteinstrument. AMSR-E LWP data are produced by Remote Sensing Systems and sponsored by the NASA Earth Science MEaSUREs DISCOVER Project and the NASA AMSR-E Science Team. Data are available at www.remss.com. The MODIS data were obtained from NASA's Level 1 and Atmosphere Archive and Distribution System(LAADS http://ladsweb.nascom.nasa.gov/). CERES-EBAF data was taken from https://ceres.larc.nasa.gov/order_data.php and is the monthly averaged product of observed TOA for which the TOA net flux is constrained to the ocean heat storage.



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
