# Peer review of "How does the UKESM1 climate model produce its cloud-aerosol forcing in the North Atlantic?"

_Atmospheric Chemistry and Physics, 2020_

## Referee Comment (RC1) · Anonymous Referee #1 · 23 Jul 2020

In this study, the authors discuss three related analyses: (1) quantification of biases in cloud properties and radiative fluxes simulated by the UKESM1 climate model in the North Atlantic against diverse observations, with good performance overall but regional-dependent biases; (2) decomposition of aerosol effective radiative forcing into microphysical response in cloud droplet number and rapid adjustments in liquid water path and cloud fraction, finding that adjustments contribute strongly albeit again in regional way; (3) identification of cloud types and cloud regime transition that contribute to forcing, finding that forcing is mostly exerted in regions of low clouds with no regime transition.

This is a good paper, which covers a lot of material. The analysis of biases is very careful and informative. I also commend the authors for clearly discussing the impact of biases on aerosol forcing in the conclusion section 4.1 – this is often not done but is important. The authors make an interesting use of offline radiative transfer calculations to quantify contributions of adjustments and give an interesting analysis of cloud regime transitions.

The paper is long, but there are no clear candidates for shortening. The conclusion repeats many points made in the body of the paper but is well structured and many readers will only read that anyway. The many figures illustrate the discussion well. The appendices provide useful information.

I have three major comments, regarding the method used to isolate the impact of aerosols on autoconversion; the impact of temporal variability on the conclusions; and improving the discussion of links between model biases and aerosol forcing. Because answering those comments may involve additional analyses, I recommend major revisions. I also recommend clarifying the discussion in places.

**1  Main comments**

- According to Appendix D, the impact of aerosols on autoconversion rates is isolated by ignoring modelled cloud droplet numbers and using fixed values instead. Doing so measures the impact of switching to a different set of cloud droplet numbers but does not isolate the impact of aerosols on autoconversion. Indeed, the reason why the prescribed numbers differ over land and ocean is to crudely represent the more polluted conditions over land. The easy option would be to take the prescribed values from the land/ocean averages of the PI simulation. A much more accurate configuration would be to use the distribution of cloud droplet number simulated in the PI simulation.

- Clouds are a very variable component of the atmosphere, as evidenced by the noisy aspect of many figures, so I was surprised that the authors base their analysis on a single, 1-year simulation. There is year-to-year variability even in stratocumulus regions, and as acknowledged by the authors nudging will not suppress that variability, which is a good thing if one wants to isolate adjustments. So readers need to be told which results can be safely generalised. This is especially true of the decomposition of aerosol forcing and the analysis of aerosol-driven cloud regime transitions. I acknowledge that extending the simulated period represents a lot of work, so a discussion of relevant literature in the conclusion section may be a good alternative, although I could not identify specific papers. Perhaps some AeroCom papers looked at interannual variability in aci?

- As I said above I very much like that the authors try to evaluate (or at least speculate) the potential impact of model biases on simulated aerosol forcing. However, that discussion could be more complete. Could biases in $f_c$ affect adjustments in that variable – for example allowing $f_c$ adjustments in sky that should be overcast, or vice-versa? Could that be significant? Also, ari is easily masked by even moderately thick clouds, so biases in $f_c$ or LWP could translate in the wrong masking of ari. Is that important? A similar comment could be made about high-cloud biases, as those biases would affect the amount of aerosol forcing that is masked by clouds above. Finally, the authors dutifully restrict their analysis to ocean regions, but land-based biases (which seem much larger than over ocean according to section 3.1.1 figures) likely matter for aerosol transport to ocean regions and their biases.

**2 Other comments**

- Line 5: Caution is of course also needed when interpreting high-resolution, process-resolving models!

- Line 14: "further large increases in $f_c$" implies that aerosols can only increase, and not decrease, cloud fraction, at least on average over the regions studied. This is true in the model used in the present study, but not in the real world, so I would suggest rephrasing here.

- Lines 28-29: Why the sudden focus on the north of Scandinavia?

- Line 75: HadGEM2-ES was a CMIP5 model, wasn't it?

- Line 84: Could give the resolution here, as "coarse resolution" for a given model may be medium or even high resolution for another.

- Line 113, lines 148-150 and lines 164-165: The paper should say early (and in the abstract) that it only considers aci with a subset of liquid (not ice) clouds, but determining what that subset is seems complicated by the distinction between convective and large-scale clouds. Does the model carry two sets of cloud variables (especially water content and cloud fraction)? Are those two sets considered separately for cloud fraction and radiative purposes? Or is $f_c$ based on both types of clouds? Would it then follow that aerosols only affect an unknown fraction of the cloud field? What would that mean for linking cloud biases to forcing?

- Lines 201-205: Should say here that the analysis of biases is limited to ocean surfaces.

- Line 220: To evaluate whether aerosol forcing contributes to biases, one could probably identify regions where aerosol impacts on $f_c$ go in the same direction of

the bias. Or you could repeat your bias analysis with the PI simulations. If it looks better against observations than PI, then aerosols must be to blame.

- Lines 238-240: This is an important observation if one wants to link the present paper to Booth et al. (2012). Because of the definition of ERF, one needs to assume that forcing decomposition and cloud regime transitions are not affected by the coupling with the ocean.

- Line 269: Clear regions do not really contribute to LWP. I suggest rephrasing.

- Line 409: "usually larger" – should be "smaller" I think.

- Lines 424-429: It would be useful to clarify here that the hypothesis is reasonable because in the model aerosols only affect autoconversion rates. Alternative mechanisms that could potentially decrease liquid water content and/or cloud fraction, for example easier evaporation of smaller droplets or changes in above-cloud air entrainment, are not represented in the model.

**3  Technical comments**

- Line 332: "exclude" rather than "prevent"?

- Figures 9 and 10: It would help to apply the same colour scale for panels (c) of both figures.

- Caption of Figure 17 could note that category 1 is clear sky.

---

## Referee Comment (RC2) · Anonymous Referee #2 · 7 Aug 2020

This paper examined the aerosol-cloud interaction performance of UK Earth System Model over the North Atlantic (NA) region. Different components of surface ERF were separated, and contributions from changes of cloud fraction, in-cloud liquid water path and droplet number concentration were evaluated. It was found the dominant forcing component of northern and southern NA region are different, which is associated with climatological cloud amount of the region. A creation of trade cumulus clouds due to the aerosols was found over the southern NA region, where the climatological cloud fraction is smaller. The paper provided a comprehensive analysis on the main topic and the structure is also well organized. Following questions should be replied before it could be published.

[Figure]

Major comments

In this study, the aerosol radiative forcing is estimated with one-year model simulation. However, the surface ACI forcing is quite noisy over some regions, even a smooth is applied. The estimated ACI forcing could be from model internal variability other than aerosol cloud interaction. With the large internal noise, it is difficult to tell whether further findings of the manuscript are correct. Ensemble simulations could be a useful and simple way to estimate the uncertainty from the internal noise (Liu et al. 2018). Only the point where the estimated ACI forcing is statistically significant could be analyzed.

The authors made detailed evaluation of the model simulated cloud properties against the observation. However, the simulated aerosol properties were barely mentioned in the manuscript. Please compare PD AOD with the observation. The changes in AOD from PI to PD should be also shown. More details could be found in the comments below.

Other comments

Line 100: Does the UKESM1 has the similar performance on global scale? Please make a comparison with the results of Mülmenstädt et al. 2019.

Line 115: Is it done in any previous studies? Please provide references here.

Line 185: Similar methods were applied in previous studies (e.g., Ghan et al. 2012; Jiang et al. 2016), which should be mentioned here.

Line 197: The surface forcing could be decomposed. How about the TOA forcing?

Line 200: There are two many figures for this part. Please consider move some figures (e.g. middle and high cloud fraction) to the supplement.

Line 400: Please show PD AOD values and make a comparison with the observation.

Line 400: Please show changes in AOD from PI to PD. The contribution from different aerosol types (sulfate, BC, dust and POM) should be also shown.

[Figure]

Line 415: Please show the cloud condensation nuclei (CCN) change together with other cloud properties.

Line 465: The surface ACI forcing due to LWP and fc is very noisy over the southern NA region. It implies the change could be from model internal variability other than aerosol-cloud-interaction.

Line 490: Are the different states classified with the annual mean value or instantaneous value? Are the estimated forcing values statically significantïij§

References

Liu, Y., Zhang, K., Qian, Y., Wang, Y., Zou, Y., Song, Y., Wan, H., Liu, X., and Yang, X.-Q.: Investigation of short-term effective radiative forcing of fire aerosols over North America using nudged hindcast ensembles, Atmos. Chem. Phys., 18, 31–47, https://doi.org/10.5194/acp-18-31-2018, 2018.

Ghan, S. J.: Technical Note: Estimating aerosol effects on cloud radiative forcing, Atmos. Chem. Phys., 13, 9971–9974, https://doi.org/10.5194/acp-13-9971-2013, 2013.

Jiang, Y., Z. Lu, X. Liu, Y. Qian, K. Zhang, Y. Wang, and X. Q. Yang (2016), Impacts of global open-fire aerosols on direct radiative, cloud and surface-albedo effects simulated with CAM5, Atmos. Chem. Phys., 16(23), 14805-14824, doi:10.5194/acp-16-14805-2016.

---

## Author Response (AR1)

We thank the Referees for taking the time to review our paper and for making helpful comments. We hope we have addressed all of the concerns below.

**Referee #1.**

According to Appendix D, the impact of aerosols on autoconversion rates is isolated by ignoring modelled cloud droplet numbers and using fixed values instead. Doing so measures the impact of switching to a different set of cloud droplet numbers but does not isolate the impact of aerosols on autoconversion. Indeed, the reason why the prescribed numbers differ over land and ocean is to crudely represent the more polluted conditions over land. The easy option would be to take the prescribed values from the land/ocean averages of the PI simulation. A much more accurate configuration would be to use the distribution of cloud droplet number simulated in the PI simulation.

*In the simulations mentioned in Appendix D we set the droplet concentration (Nd) terms as used by the auto-conversion equation to the constant values in both the PI and PD simulations. Since both the PI and PD runs both use the same Nd values there would be no impact of the aerosol changes between PI and PD on the autoconversion rates except via any changes in liquid water content. The latter is possible via changes in droplet scattering, semi-direct aerosol effects, etc. However, the main aim was to test the effect of the aerosols on the auto-conversion rate via changes in Nd to test the hypothesis that this was the main cause of the LWP and cloud fraction changes. Since the results showed that the increases in these two quantities between PI and PD was reduced dramatically in the constant auto-conversion runs we feel that this hypothesis was sufficiently proved with this test. The differences in Nd between land and ocean regions for the constant values used in the autoconversion equation should have little impact on this conclusion since the contrast is the same in both the PI and PD runs. The description given in Appendix D did not get across the fact that we used the constant Nd values in both the PI and PD and so we have changed it to :-*

**Appendix E: Removing aerosol impact on rain autoconversion**

900 Figures E1 and E2 show the percentage increases between PI and PD for $LWP_\text{ic}$ and $f_\text{c}$ ($\Delta LWP_\text{ic}$ and $\Delta f_\text{c}$), respectively, when aerosols are prevented from affecting the rain autoconversion process. This is done by setting $N_\text{d}$ in the autoconversion process equation to a constant value of $300 \text{ cm}^{-3}$ over land and $100 \text{ cm}^{-3}$ over oceans in both the PI and PD runs. These are here termed the ConstantNdAutoCon runs. Most oceanic regions showed positive $\Delta LWP_\text{ic}$ and $\Delta f_\text{c}$ in the full model and near-zero

Clouds are a very variable component of the atmosphere, as evidenced by the noisy aspect of many figures, so I was surprised that the authors base their analysis on a single, 1-year simulation. There is year-to-year variability even in stratocumulus regions, and as acknowledged by the authors nudging will not suppress that variability, which is a good thing if one wants to isolate adjustments. So readers need to be told which results can be safely generalised. This is especially true of the decomposition of aerosol forcing and the analysis of aerosol-driven cloud regime transitions. I acknowledge that extending the simulated period represents a lot of work, so a discussion of relevant literature in the conclusion section may be a good alternative, although I could not identify specific papers. Perhaps some AeroCom papers looked at interannual variability in aci?

*We decided to run repeat the simulations for an additional year to determine how sensitive the results were to a different meteorological year. We found that the decomposition of the aerosol forcing was very similar to that obtained using the original meteorology with only small differences apparent. We added the following text and figures as a new appendix :-*

Fig. H1 shows that the pattern and magnitude of both $ERF_{ARI}$ and $ERF_{ACI}$ are very similar for the alternate year with only slight differences (compare to Fig. 12): the region of $ERF_{ARI}$ forcing off the coast of the USA extends further east across the Atlantic; the main region of $ERF_{ACI}$ around Newfoundland is still present, but extends further south and less to the southeast; the region of $ERF_{ACI}$ north of Scandinavia has a similar spatial pattern, but is enhanced somewhat in the alternate year.

[Figure]

**Figure H1.** As for Fig. 12 except for the 2010-2011 period.

Fig. H2 (compare to Fig 15) shows the contributions to $ERF_{ACI}$ from the changes in $N_d$, $LWP_{ic}$ and $f_c$ between the PI and PD for the alternate year. It shows that there are some small differences in the spatial patterns, but the overall patterns of the contributions remain very similar with $N_d$ and $LWP_{ic}$ changes dominating over the northern NA region and $f_c$ changes dominating over the southern NA as was the case for the original chosen time period.

[Figure]

**Figure H2.** As for Fig. 15 except for the 2010-2011 period.

A figure equivalent to Fig. 18 for the alternative year showing the contributions from changes between different cloud states (not shown) reveals a very similar pattern with very similar magnitudes of contribution. Likewise, the alternative version of Fig. 19 (contributions from changes between cloud fractions; not shown) is again very similar to original version in terms of both the pattern and magnitude of the contributions.

*We have also added the following to the main body of the text :-*

The results in the main body of the paper are based on meteorology and emissions from the period 28th March 2009 to 28th March 2010. It is possible that the results presented vary depending on the chosen year since meteorology, cloud fields, etc. vary from year to year. To address this we have also run the PI and PD simulations for an additional year for the period 28th March 2010 to 28th March 2011. We have compared selected key results from Section 3.2, which are shown in Appendix G. Very similar results are found using the alternative year, which demonstrates that our results are robust and not sensitive to the chosen year of meteorology.

As I said above I very much like that the authors try to evaluate (or at least speculate) the potential impact of model biases on simulated aerosol forcing. However, that discussion could be more complete. Could biases in $f_c$ affect adjustments in that variable – for example allowing $f_c$ adjustments in sky that should be overcast, or vice-versa? Could that be significant? Also, ari is easily masked by even moderately thick clouds, so biases in $f_c$ or LWP could translate in the wrong masking of ari. Is that important? A similar comment could be made about high cloud biases, as those biases would affect the amount of aerosol forcing that is masked by clouds above. Finally, the authors dutifully restrict their analysis to ocean regions, but land-based biases (which seem much larger than over ocean according to section 3.1.1 figures) likely matter for aerosol transport to ocean regions and their biases.

*We have added further discussion on this in Section 4.1. However, it is difficult to quantify these effects, so we leave this for future more detailed work on this. As for biases over land we decided not to focus on these since satellite retrievals are less accurate over land (or not possible for AMSRE), which would make interpretation more difficult. Also, the focus of our study is the North Atlantic oceanic region. Here is the revised part of Section 4.1 :-*

570      Low-altitude cloud fraction biases have the potential to significantly impact aerosol forcing since they are closest in altitude to the aerosol sources. If we assume that the PI cloud cover is biased by a similar amount to the PD cloud cover and assume no cloud adjustments to aerosol then the bias in forcing will be similar to the bias in $f_c$. The results from this paper suggest that the second assumption is reasonable for the northern NA region because the Twomey effect dominates. Thus we might expect

the 5.1% low-altitude $f_c$ bias there to make a small contribution to any error in forcing bias. Of course it is also possible that
575     the $f_c$ increase between PI and PD is underestimated by the model for this region, but the PI low-altitude $f_c$ is too high in order to give an overall PD bias. As such it is difficult to make firm conclusions.

    However, in In the southern NA $f_c$ changes in response to aerosols (adjustments) were large, so the above assumptions are less valid and the effects on forcing even less clear. The negative present-day $f_c$ biases could indicate a cloud fraction response to aerosol that is too low, which would cause subsequent negative forcing biases. On the other hand, a model the too-low $f_c$
580     that is too low may mean that the model PI era is in a broken, precipitating cloud regime too often. Such regimes are thought to be more sensitive to aerosols and more prone to produce cloud adjustments (Ackerman et al., 2004b)(Ackerman et al., 2004a). In this case the model forcing values would be too large. Further The effect of this may be significant given the large bias here (-23.9%). Likewise, the too-large $f_c$ values in the northern NA (if also occurring in the PI) might prevent some instances of $f_c$ increase between PI and PD and lead to a forcing that is too small, although the overall bias for the northern NA region
585     was only 5.1%. The presence of clouds can also mask ARI forcing and hence the $f_c$ and $LWP_{ic}$ biases might therefore affect the predicted $ERF_{ARI}$ magnitude. The larger low- and mid-altitude biases (which are likely to be thicker and hence have a stronger masking effect than high-altitude clouds) in the southern NA combined with the larger $ERF_{ARI}$ suggest that this effect would likely be more pronounced there. Here the $f_c$ biases are negative, which would produce a positive $ERF_{ARI}$ bias by this mechanism. Mid- and high-altitude clouds can also mask $ERF_{ACI}$ forcing from low-altitude cloud (which is likely to
590     be the biggest contributor to forcing). For both the northern NA and southern NA mid-altitude clouds tend to have negative biases that are larger in magnitude than the positive biases of the high-altitude clouds suggesting the potential for an overall negative $ERF_{ACI}$ forcing from this mechanism. However, further work would be needed to quantify the effect of the model $f_c$ biases on the aerosol forcing.

**Line 5: Caution is of course also needed when interpreting high-resolution, process-resolving models!**

*Agreed, but it is difficult to mention this here in the abstract. However, we note that we include this sentence in the Recommendations section :-*
*"The assumption would be that the cloud responses to aerosol of this would be more accurate than the global model resolution simulations, although its performance should be tested using the observations."*

**Line 14: "further large increases in fc" implies that aerosols can only increase, and not decrease, cloud fraction, at least on average over the regions studied. This is true in the model used in the present study, but not in the real world, so I would suggest rephrasing here.**

*We feel that it is clear that we are talking about the "world according to the model" throughout this part of the abstract and so would like to keep this sentence as it is to avoid adding lots of caveats and making it less succinct.*

**Lines 28-29: Why the sudden focus on the north of Scandinavia?**

*This was the main region where LWPic biases were seen and it is still part of the North Atlantic region, so we thought it should be mentioned.*

Line 75: HadGEM2-ES was a CMIP5 model, wasn't it?

*Yes, thanks, this has been fixed :-*

> of this paper. It has been suggested (Booth et al., 2012, hereafter B12) that surface radiative aerosol forcing is the dominant driver of the variability in multi-decadal NA sea surface temperatures (SSTs) for the ocean-atmosphere coupled GCM (the UK
> 70 Met Office HadGEM2-ES model) that was used in the  Coupled Model Intercomparison Project (CMIP5). NA SST variability has been linked to impacts on important climate phenomena such as hurricane and tropical storm activity (Zhang and Delworth, 2006; Smith et al., 2010; Dunstone et al., 2013); rainfall anomalies in Europe and North America (Sutton and Hodson, 2005; Sutton and Dong, 2012); droughts in the African Sahel and Amazonian regions (Hoerling et al., 2006; Knight et al., 2006; Ackerley et al., 2011); Greenland ice-sheet melt rates (Holland et al., 2008; Hanna et al., 2012); anomalies in
> 75 sea-levels (McCarthy et al., 2015); and the mid-latitude jet strength (Woollings et al., 2015). For a review of changes in the North Atlantic climate system (with a focus on more recent changes) see Robson et al. (2018).
>     B12 showed that HADGEM2-ES, which represented aerosol-cloud interactions, reproduced the observed NA SSTs with good fidelity in contrast to the  CMIP3 models that mostly did not include aerosol-cloud effects. Furthermore, tests

Line 84: Could give the resolution here, as "coarse resolution" for a given model may be medium or even high resolution for another.

*This has been added:-*

> The claims made in B12 are based upon a GCM and not direct observations. As mentioned earlier, the aerosol forcing in GCMs is highly uncertain for many potential reasons. For example, B12 used a coarse N96 model resolution and thus the model

Line 113, lines 148-150 and lines 164-165: The paper should say early (and in the abstract) that it only considers aci with a subset of liquid (not ice) clouds, but determining what that subset is seems complicated by the distinction between convective and large-scale clouds. Does the model carry two sets of cloud variables (especially water content and cloud fraction)? Are those two sets considered separately for cloud fraction and radiative purposes? Or is $f_c$ based on both types of clouds? Would it then follow that aerosols only affect an unknown fraction of the cloud field? What would that mean for linking cloud biases to forcing?

*The model does carry two sets of liquid water content and cloud fraction values – one for large-scale cloud and one for convective cores. As mentioned in Appendix C :-*

*"LWP associated with the convection scheme is only used by the radiation scheme for shallow convective clouds (clouds with geometrical depths less than 500 m over land and 1500 m over ocean) meaning that the majority of liquid water in deep clouds has no effect on radiation"*

*When doing the forcing calculations and partitioning we use the model cloud fraction from the large-scale cloud scheme. When comparing with observations and doing the model bias analysis the cloud fractions used in this paper come from the COSP satellite simulator, which only uses the cloud that is considered by the radiation scheme (i.e., mostly from the large-scale cloud scheme, see above). Hence the model analysis does not apply to the deeper convective core regions. However, convective cores make up only a very small fraction of the total cloud area and so would have little radiative impact anyway. Also, as also mentioned in Appendix C :-*

*"Liquid water content and vapour are detrained to the environment from the convection scheme and incorporated into the large-scale cloud scheme (see UM Documentation Paper 030; hereafter UMDP030; \url{https://code.metoffice.gov.uk/doc/um/vn11.3/umdp.html#030})."*

*This means that cloud associated with the convection scheme (that directly detrained or created due to the convective detrainment of humidity) does count towards the large-scale cloud fraction and has a radiative impact, and so will be included in both the COSP and model cloud fractions used here.*

*However, aerosols are not considered by the convective parameterization and for convection can only affect the cloud after it has been detrained. This is a limitation in the representation of aerosol forcing in this model (and many others) and is mentioned in the paper when talking about convective parameterizations :-*

*"The parameterizations do not take into account aerosol, or droplet concentrations and they use their own simplified microphysics scheme."*

*We have also added the following (in blue) to the abstract :-*

nudging to analysis; one simulation has pre-industrial (PI) and one has present-day (PD) aerosol emissions. This model does not include aerosol effects within the convective parameterization (but aerosol does affect the cloud associated with detrainment) and so it should be noted that the representation of aerosol forcing for convection is incomplete.

*We also add this line (in blue) to the main text in Section 2.2 :-*

Shallow, mid and deep convection are parameterized separately to other cloud types (see Walters et al., 2019, for details). The parameterizations do not take into account aerosol, or droplet concentrations and they use their own simplified microphysics scheme. Therefore the representation of aerosol forcing is incomplete. For cloud that is not shallow, mid, or deep convection

Lines 201-205: Should say here that the analysis of biases is limited to ocean surfaces.

*Done. :-*

**3.1 Model evaluation against satellite observations**

Here we evaluate the PD simulation against satellite observations, but limit our analysis to ocean regions due to the lesser reliability of satellite products over land. The motivation for the model evaluation is that without a good representation of

Line 220: To evaluate whether aerosol forcing contributes to biases, one could probably identify regions where aerosol impacts on $f_c$ go in the same direction of the bias. Or you could repeat your bias analysis with the PI simulations. If it looks better against observations than PI, then aerosols must be to blame.

*We feel that we have addressed this point above and in Section 4.1.*

Lines 238-240: This is an important observation if one wants to link the present paper to Booth et al. (2012). Because of the definition of ERF, one needs to assume that forcing decomposition and cloud regime transitions are not affected by the coupling with the ocean.

*We have added this sentence :-*

*"This also implies that the coupled model and the nudged model used here exhibit similar cloud regimes and gives more confidence that the results in this paper apply to coupled models."*

Line 269: Clear regions do not really contribute to LWP. I suggest rephrasing.

*Thanks. This has been changed to :-*

*"Note that LWP from AMSR-E is the all-sky LWP and so includes the zero values from clear regions and the LWP contributions from cloudy regions"*

Line 409: "usually larger" – should be "smaller" I think.

*Thanks – we meant ERF_ARI here.*

in magnitude near Africa south of the equator. However, over the whole of the NA domain $ERF_{ARI}$ is larger in magnitude than $ERF_{ACI}$ (-1.9 vs -1.7 W m$^{-2}$) due to the fact that $ERF_{ARI}$ is usually larger over land. The dominance of

Lines 424-429: It would be useful to clarify here that the hypothesis is reasonable because in the model aerosols only affect autoconversion rates. Alternative mechanisms that could potentially decrease liquid water content and/or cloud fraction, for example easier evaporation of smaller droplets or changes in above cloud air entrainment, are not represented in the model.

*We added a sentence here to make this point.*

*"We note that in reality clouds can also respond to enhanced aerosol by increasing cloud top entrainment, which can reduce LWPic and fc (Ackerman et al., 2004; Bretherton et al., 2007; Hill et al., 2009). However, this mechanism is currently not included in the model."*

Line 332: "exclude" rather than "prevent"?
*Changed.*

Figures 9 and 10: It would help to apply the same colour scale for panels (c) of both figures.

*We have applied the same colourscale for panel c of Figs. 8 and 9 since they are directly comparable. However, Fig. 10c is the contribution from just the cloud fraction bias and we would prefer to keep the colour scale as it is to show some of the detail.*

Caption of Figure 17 could note that category 1 is clear sky.

*Fixed.*

[Figure]

**Figure 17.** The cloud states used in the following figures. The shading indicates the presence of low, mid, or high altitude cloud (see text for the definitions of this) as determined by requiring the model cloud fraction to be larger than 0.01. Cloud state #1 is clear-sky.

**Referee #2**

**Major comments**

In this study, the aerosol radiative forcing is estimated with one-year model simulation. However, the surface ACI forcing is quite noisy over some regions, even a smooth is applied. The estimated ACI forcing could be from model internal variability other than aerosol cloud interaction. With the large internal noise, it is difficult to tell whether further findings of the manuscript are correct. Ensemble simulations could be a useful and simple way to estimate the uncertainty from the internal noise (Liu et al. 2018). Only the point where the estimated ACI forcing is statistically significant could be analyzed. The authors made detailed evaluation of the model simulated cloud properties against the observation. However, the simulated aerosol properties were barely mentioned in the manuscript. Please compare PD AOD with the observation. The changes in AOD from PI to PD should be also shown. More details could be found in the comments below.

*The point about only using one year of data and the results potentially being due to noise was also mentioned by Referee #1. We have run an additional year of simulation (with a different year of meteorology) and found almost identical results. Please see our response above for more details. We have added a reference to Liu (2018) also. We have included an AOD evaluation and PI to PD changes in the revised manuscript.*

**Other comments**

Line 100: Does the UKESM1 has the similar performance on global scale? Please make a comparison with the results of Mülmenstädt et al. 2019.
*We have added some text to describe the comparison to Mulmenstadt 2019. However, we restricted the comparison to the North Atlantic since this was the focus of our paper :-*

> 680    Our results are somewhat different from those obtained in Mülmenstädt et al. (2019) using a similar technique for the ECHAM-HAMMOZ model. For the North Atlantic region their model produced $ERF_{ACI}$ contributions from $N_{\mathrm{d}}$, $LWP_{\mathrm{ic}}$ and $f_{\mathrm{c}}$ that were located in approximately the same locations in contrast to the $f_{\mathrm{c}}$ contribution from our model being predominantly in the southern NA and the $N_{\mathrm{d}}$ and $LWP_{\mathrm{ic}}$ contributions being further north. This suggests that the lifetime effect (and the associated aerosol induced precipitation suppression process described above) operates differently between the two models, or
> 685    that the types and locations of clouds differs between the models. Their model also has very little aerosol forcing in the region north of Scandinavia, whereas our model had quite a large forcing due to $N_{\mathrm{d}}$ and $LWP_{\mathrm{ic}}$ changes there. It is possible that some of these differences are due to the use of different years or different decomposition techniques. Nevertheless, understanding these model differences may lead to a method of model evaluation to determine which is correct that might help bring down aerosol forcing uncertainty.

Line 115: Is it done in any previous studies? Please provide references here.
*Yes, this method has been used before. We added a reference to Seethala, JGR, 2020 (doi:10.1029/2009JD012662).*

Line 185: Similar methods were applied in previous studies (e.g., Ghan et al. 2012; Jiang et al. 2016), which should be mentioned here.
*Thanks, we added references to these papers.*

Line 197: The surface forcing could be decomposed. How about the TOA forcing?

*TOA forcing can also be decomposed using this method. We now mention this in the paper. We focus on surface forcing because of the interest in ocean forcing in the North Atlantic. We added this :-*
*"We also note that TOA fluxes could also be decomposed using this technique, but we focus on the surface forcing here."*

Line 200: There are two many figures for this part. Please consider move some figures (e.g. middle and high cloud fraction) to the supplement.

*We would prefer to keep these figures in we feel that clouds of different altitudes should be considered. Referee #1 has also referred to them, for example indicating their importance for shielding lower clouds and masking ARI and ACI effects.*

Line 400: Please show PD AOD values and make a comparison with the observation.

*This is now included in the new manuscript :-*

820 **Appendix C: Aerosol properties**

Fig. C1 shows an evaluation of the 550 nm model aerosol optical depth (AOD) using data from MODIS. The MODIS data is a combination of the 550 nm Dark Target and Deep Blue product "Dark_Target_Deep_Blue_Optical_Depth_550_Combined" from the Level-3 product (Levy et al., 2013). The model shows positive biases in equatorial Africa, but negative biases in northern Africa and in the ocean to the west of there. This indicates some issues with dust in the model. Further north there

825 are small positive biases in the aerosol outflow region to the east of the USA. Model values are also too high in the region to the north and west of the UK, and over the Mediterranean region. This may indicate an overestimate of aerosol mass from the land sources of pollution, or a lack of scavenging. However, there are potential issues with the comparison between MODIS and the model in these regions since MODIS retrievals are only possible in cloud-free regions and model values are taken in both cloudy and cloud-free regions. Since the cloud coverage of this region is very high (see Figs. 1, 2 and 3) this may cause

830 sampling biases and further work is needed to examine the effects of this. Overall model NMBF error values are -8.5, 5.6 and -18.3% over the NA, Northern NA and Southern NA regions, respectively.

[Figure]

**Figure C1.** Time-mean 550 nm aerosol optical depth (AOD) model evaluation. The MODIS data is from daytime overpasses of the Aqua satellite using the combined Dark Target and Deep Blue Level-3 product. Model local times within 3 hours either side of 13:30 are used to approximately match the satellite overpass times.

Line 400: Please show changes in AOD from PI to PD. The contribution from different aerosol types (sulfate, BC, dust and POM) should be also shown.

*We now show the PI to PD changes. However, we don't have information on the direct contributions from the different aerosol types to AOD. Instead we show the changes in the column integrated aerosol mass mixing ratios for the different types :-*

We now examine the changes between PI and PD for various aerosol-related model fields. Figures C2 and 13 shows that the spatial patterns of the changes in AOD, CCN at 0.2% supersaturation ($CCN_{0.2\%}$) and $N_d$ are very similar suggesting that changes in both AOD and $CCN_{0.2\%}$ are a good proxy for $N_d$ changes in this region. In terms of aerosol composition (Fig. C3),

835 the largest changes in column aerosol mass (the vertically integrated aerosol mass concentrations for all size modes) occur for sulphate and black carbon (BC). Sulphate changes occur further north than BC changes with the latter occurring mostly over southern Europe and northern Africa. Sulphate changes dominate over the ocean except at around 18° N where there is a band of larger BC change likely associated with outflow from Africa. Similarly to BC, organic matter (OM) increases also occur over Africa and over the Atlantic ocean at 18° N with both BC and OM potentially contributing to the increase in $CCN_{0.2\%}$

840 and $N_d$ over the Atlantic that region. Sulphate changes likely dominate the $N_d$ changes further north in the Atlantic. Organic matter (OM) aerosol mass decreases in the northern part of the Atlantic. Dust and sea-salt column mass changes are very small in comparison to the other aerosol types for this region.

[Figure]

**Figure C2.** Mean percentage increase (between PI and PD model runs) in: a) 550 nm AOD, b) CCN at 0.2% supersaturation.

[Figure]

**Figure C3.** As for Fig. C2 except for changes in : a) column sulphate aerosol mass, b) column black carbon (BC) aerosol mass, c) column organic matter (OM) aerosol mass, d) column dust mass, e) column sea-salt aerosol mass.

*We also add this to the main text :-*

**3.2.1 Changes in cloud properties from PI to PD and their contributions to the ACI forcing**

Fig. 13 shows the time-mean changes in cloud properties ($N_d$, $LWP_{ic}$ and $f_c$) between the PI and PD simulations (PD minus PI as a percentage of PI). Considering oceanic regions, percentage increases in $N_d$ are greatest off the east coast of the USA and Canada (to the south of Newfoundland); off the SW coasts of Spain/Portugal and West Africa, and in the region north of Scandinavia. There are increases across the whole Atlantic running from the USA to West Africa, but the magnitude decreases moving east (except close to the European/African west coast) likely reflecting the decreasing influence of pollution from the east coast of the USA. The spatial pattern of $N_d$ change matches the spatial pattern of the change in column sulphate aerosol mass fairly well (Fig. C3) suggesting that changes in sulphate aerosol are the main cause of the $N_d$ changes. The exception is the region stretching from southern Portugal down the west coast of North Africa and also across the Atlantic south of around $20^\circ$ N where the $N_d$ changes coincide with changes in black carbon (BC) and organic matter (OM) column mass.

Line 415: Please show the cloud condensation nuclei (CCN) change together with other cloud properties.
*This is now shown (see the response above).*

Line 465: The surface ACI forcing due to LWP and fc is very noisy over the southern NA region. It implies the change could be from model internal variability other than aerosol-cloud-interaction.

*We feel that the addition of the results from the extra year of data (see above) shows that this is not due to noise since a similar result is found.*

Line 490: Are the different states classified with the annual mean value or instantaneous value? Are the estimated forcing values statically significant?

*Yes, the states are classified with the instantaneous data. We have added a sentence to make this clearer in the revised manuscript. The fact that the same result is obtained using the alternative year of meteorology indicates that the result is robust.*

**Other changes**

Title changed to :-

[revised manuscript text omitted]